# EFFICIENT DIFFUSION MODELS UNDER NONCON-VEX EQUALITY AND INEQUALITY CONSTRAINTS VIA LANDING

## ABSTRACT

The generative modeling of data in constrained sets is central to scientific and engineering applications with physical, geometric, or safety constraints (e.g., molecular generation, robotics). This article constructs constrained diffusion models on a generic nonconvex feasible sets $\Sigma$, by introducing a unified framework that simultaneously enforces both equality and inequality constraints throughout the diffusion process. Our theory and implementations encompass both overdamped and underdamped dynamics for the forward and backward sampling. The key algorithmic ingredient is a computationally efficient landing mechanism that replaces costly and not-always-well-defined projections onto $\Sigma$, maintaining feasibility without Newton solves and avoiding projection failures. Leveraging underdamped dynamics whose faster mixing reduces the steps needed to reach the prior distribution, the commonly-believed unavoidable heavy forward simulation cost in the constrained diffusion is alleviated. Empirically, this reduces function evaluations, enabling more efficient inference and training while preserving sample quality and substantially lowering memory usage. On equality-only and mixed (equality and inequality) benchmarks, our method shows reasonable sample quality, while substantially reducing computational cost and function evaluations. These results indicate that landing-based enforcement combined with underdamped dynamics provides a practical and scalable recipe for constrained diffusion on nonconvex feasible sets.

## 1 INTRODUCTION

Generative modeling is a fundamental machine learning task. In recent years, denoising diffusion models [1, 2] have become the state-of-the-art in image, audio, and video generation, matching and surpassing earlier approaches such as GAN [3, 4]. Key advantages include ease of training, high-fidelity sampling, and their flexibility in conditional generation.

The majority of these impressive results have been achieved for data in an unconstrained space, particularly $\mathbb{R}^d$. While this suffices for digital content creation, there are many emerging applications in science and engineering, where the generation of high-fidelity samples that adhere to constraints is crucial. Examples include molecular generation [5–7], where atoms must satisfy distance or chirality constraints and respect physical laws, robotics [8–10], where trajectories are subject to dynamics, actuation limits, safety margins, and collision-avoidance, and shape optimization for engineering design [11–13], where specifications, symmetries, and manufacturing impose constraints. In these settings, a valid sample needs to remain inside a non-trivial feasible set $\Sigma \subset \mathbb{R}^d$, as constraint violations would render the generation physically meaningless or unsafe. Unfortunately, enforcing constraints in diffusion models is challenging: (i) Projection-based methods demand repeated Newton-iterations, which scales poorly with dimension and may fail when $\Sigma$ is nonconvex and projections are undefined [14]; (ii) Reparametrization encodes the constraints implicitly, which requires domain knowledge, may alter the conditioning of the score matching, and the fidelity of the sampling. Reparametrizations might be hard to find in applications with complex mixed equality-inequality constraints [15]; (iii) Penalty and barrier methods [16, 17] incorporate constraints through additional terms in the objective function, which may introduce bias (barrier functions), lead to constraint violations (penalty), and result in additional hyperparameters that are difficult to tune. Conse-

quently, constrained diffusion is computationally demanding as both training and inference rely on the simulation of diffusion processes. For example, this poses a significant bottleneck for applications including robotics - where trajectories have to be generated in real time on resource-constrained hardware.

This article addresses the need for computationally-efficient constrained diffusion. Our contributions are threefold:

- **(Landing based diffusion process)**    We extend the recently developed constrained optimization technique of landing [18–21] to generative modeling, enabling inexpensive first-order updates that steer every step of the diffusion process toward $\Sigma$ without requiring exact projections or Newton iterations. Landing incurs a negligible computational overhead and provably maintains feasibility along the path.

- **(Unified framework for constraints)**    Our method works for general nonconvex feasible sets as we develop an SDE-based framework that unifies both equality and inequality constraints.

- **(Fast constrained diffusion via underdamped dynamics)**    The framework encompasses constrained versions of both underdamped and overdamped Langevin dynamics. The fast mixing of the underdamped version is leveraged to substantially reduce the length of forward trajectory $N$ to reach the prior distribution, thereby cutting the dominant sampling (up to $47\times$) and training cost (up to $5\times$) in constrained diffusion models.

## 2    RELATED WORKS

Recent works have extended score-based diffusion models from Euclidean spaces to non-Euclidean domains. Our setup considers a domain specified by equality and inequality constraints, and closely related is a rich collection of successful generative models for manifold data [22–27]. However, while many manifolds considered (such as $S^d$ and $SO(n)$) can also be specified using equality constraints, the geometry of the constrained set can easily become too complicated to handle when there are a large number of constraints, or when the constraints introduce manifolds with boundaries or even lower dimensional structures.

Approaches directly targeting constrained generation also exist, particularly when data lie in a bounded subset $\Sigma \subset \mathbb{R}^d$. Several strategies have been explored. For example, classical reflected Brownian motion [28, 29] was recently leveraged to create constrained diffusion models [17, 15, 30] , and the recent development of mirror Langevin dynamics and algorithms (the version of [31, 32]) was also employed for constrained generation [33]. However, the former approach is difficult to be made simulation-free and/or tractable in the conditional score, which is essential for efficiency, while the latter only works for convex constraints. Even more recently, Riemannian Denoising Diffusion Probabilistic Models (RDDPM) [34] extends score-based models to general manifolds via per-step Newton's projections, ensuring feasibility but incurring sizable computational cost and occasional projection failures on nonconvex sets. In parallel, Riemannian Flow Matching (RFM) [26] learns manifold flows without projections for simple manifolds, yet typically needs a long integration horizon or still projections on nontrivial geometries.

In the light of these advances, we construct an efficient diffusion process that remains on feasible sets described by equality and inequality constraints. The key is to incorporate landing, a technique developed in constrained optimization [18–21], which handles non-convex constraints and guarantees feasibility (without requiring projections, retractions, or evaluations of the exponential map that tend to be expensive).

## 3    PRELIMINARIES & NOTATIONS

**Constrained set and geometry.**    We implement the diffusion model on a constrained set $\Sigma := \left\{ x \in \mathbb{R}^d \mid h(x) = 0, g(x) \leq 0 \right\}$ defined by smooth equality constraints $h : \mathbb{R}^d \to \mathbb{R}^m$ and inequality constraints $g : \mathbb{R}^d \to \mathbb{R}^l$. For theoretical analysis, we assume that $\Sigma$ is a stratified manifold and constraints $h, g$ satisfy the relaxed Constant Rank Constraint Qualification (rCRCQ) [35] on a neighborhood of $\Sigma$. Specifically, for each $x \in \mathbb{R}^d$ with the index set of active inequalities $I_x := \{ j \in [l] \mid g_j(x) \geq 0 \}$, we say that rCRCQ holds if, for every index set $J \subset I_x$, the

set $\{\nabla h_i(y)\}_{i=1}^m \cup \{\nabla g_j(y)\}_{j \in J}$ has the same rank for any $y$ in the neighborhood of $x$ (see Remark 2 for further discussion on rCRCQ). We note that the rCRCQ implies that the stacked Jacobian $\nabla J(x) \in \mathbb{R}^{(m+|I_x|) \times d}$ (with $J(x) := [h(x), g_{I_x}(x) + \epsilon]^T$) has a constant rank in the neighborhood of $x$, where the boundary repulsion rate $\epsilon > 0$ is a hyperparameter to be introduced later. Due to this result, the tangent space of $\Sigma$ is characterized by the kernel of the Jacobian, given by

$$T_x\Sigma := \left\{ p \in \mathbb{R}^d \mid \nabla J(x)p = 0 \right\}.$$

Accordingly, the orthogonal projector $\Pi(x) := I - \nabla J(x)^T G(x)^\dagger \nabla J(x)$ onto $T_x\Sigma$ is well-defined, where $G(x)^\dagger$ is the Moore-Penrose pseudo-inverse of $G(x) := \nabla J(x) \nabla J(x)^T$.

For the reference measure on $\Sigma$, we use the induced surface (Hausdorff) measure on $\Sigma$, denoted as $d\sigma_\Sigma$. In the underdamped setting, the natural phase space is the cotangent bundle given by

$$T^*\Sigma := \left\{ (x, p) \in \mathbb{R}^d \times \mathbb{R}^d \mid x \in \Sigma, \nabla J(x)p = 0 \right\}$$

In this manifold, the natural reference measure is the Liouville measure $d\sigma_{T^*\Sigma}(x, p) = d\sigma_\Sigma(x) \otimes dp(x)$ where $dp(x)$ is Lebesgue measure on $T_x^*\Sigma := \left\{ p \in \mathbb{R}^d \mid \nabla J(x)p = 0 \right\}$. For the detailed background and notations, see subsection B.1 for the overdamped, subsection B.2 for the underdamped, and the table of key notations (Table 5).

**Time grid and schedule.**    In our paper, we use continuous time $t \in [0, T]$ and a uniform grid $t_k := k\Delta t$ for $k \in \{0, ..., N\}$ with $T = N\Delta t$ for the implementation of the diffusion model with step size $\Delta t > 0$. Also, the noise magnitudes used in implemented diffusion models are specified by a scheduler $\sigma : [0, T] \to \mathbb{R}_+$ and, in our case, we use the linear scheduler given by $\sigma(t) := \sigma_{\min} + \frac{t}{T}(\sigma_{\max} - \sigma_{\min})$ with $\sigma_k := \sigma(t_k)$.

## 4 MAIN RESULTS

### 4.1 CONSTRAINED LANGEVIN DYNAMICS VIA LANDING

**From projections to intrinsic landing mechanisms.**    Classical constrained samplers take an unconstrained (in $\mathbb{R}^d$) or tangential step (in $T_x\Sigma$) and then project back to $\Sigma$. This is often problematic: (i) on nonconvex manifolds, nearest-point projection can be multi-valued or not globally defined; (ii) per-step projection solves are costly and may fail (e.g. Newton's method failure); and (iii) behavior near $\partial\Sigma$ is delicate as the active set $I_x$ changes frequently. We therefore seek a projection-free scheme that remains well-posed when local projections struggle. Our approach embeds robustness directly in the SDE via a landing term that enforces exponential decay of constraint violation $J$:

$$dJ(X_t) = -\alpha\sigma(t)^2 J(X_t)dt \qquad \text{(Target landing property)}$$

so discretization-induced infeasibility self-corrects without explicit projection. The landing modifies only the normal component of the drift, leaving tangential drift and diffusion unchanged, so trajectories evolve intrinsically on $\Sigma$ (or $T^*\Sigma$ in the underdamped case). Formally, any diffusion process $X_t$ with this property enjoys the guarantees stated next. For the detailed proofs of mathematical claims below, see Appendix B.

**Lemma 1** (Exponential decay of constraint functions). *Under the target property $dJ(X_t) = -\alpha\sigma(t)^2 J(X_t)dt$, the diffusion process $X_t$ satisfies the following constraint satisfaction property almost surely:*

$$h_i(X_t) = h_i(X_0)e^{-\alpha S(t)}, \quad t \geq 0$$

*and*

$$\begin{cases} g_j(X_t) = -\epsilon + (g_j(X_0) + \epsilon)e^{-\alpha S(t)}, & t \leq \tau_{j,\epsilon} \\ g_j(X_t) \leq 0, & t \geq \tau_{j,\epsilon}, \end{cases}$$

*where $S(t) := \int_0^t \sigma(s)^2 ds$ and $\tau_{j,\epsilon}$ are defined to be*

$$\tau_{j,\epsilon} := \inf\left\{ t \geq 0 \mid S(t) \geq \frac{1}{\alpha}\ln\left(\frac{g_j(X_0) + \epsilon}{\epsilon}\right) \right\}, \quad \forall j \in I_{X_0}.$$

**Constrained Overdamped Langevin dynamics via Landing (OLLA).** We first derive such landing-based constrained Langevin dynamics for the overdamped case. By viewing constrained Langevin dynamics in Lagrangian form [36], we can pick a Lagrangian process $d\lambda_t$ so that it can impose the target property $dJ(X_t) = -\alpha\sigma(t)^2 J(X_t)dt$ and have a closed-form SDE as follows:

**Proposition 1** (Construction, stationarity and backward process of OLLA). *Consider the following Lagrangian form constrained overdamped Langevin dynamics of $X_t \sim q_t$:*

$$dX_t = -\frac{1}{2}\sigma(t)^2 \nabla f(X_t)dt + \sigma(t) \circ dW_t + \nabla J(X_t)^T d\lambda_t \tag{1}$$

*where $d\lambda_t$ is the adapted process such that $dJ(X_t) = -\alpha\sigma(t)^2 J(X_t)$. The explicit solution of $d\lambda_t$ provides the closed form SDE of (1) as follows:*

$$dX_t = -\left[\frac{\sigma(t)^2}{2}\Pi(X_t)\nabla f(X_t) + \underbrace{\alpha\sigma(t)^2 \nabla J(X_t)^T G^\dagger(X_t)J(X_t)}_{Landing\ term}\right]dt + \frac{\sigma(t)^2}{2}\mathcal{H}(X_t)dt$$
$$+ \sigma(t)\Pi(X_t)dW_t,$$

*where $\mathcal{H}$ is the mean curvature correction term defined as*

$$\mathcal{H}(x) := -\nabla J(x)^T G^\dagger(x)\left[\mathsf{Tr}\left(\nabla^2 J_1(x)\Pi(x)\right), ..., \mathsf{Tr}\left(\nabla^2 J_{m+|I_x|}(x)\Pi(x)\right)\right]^T.$$

*Furthermore, the backward process $\overleftarrow{X}_t$ of OLLA is given as:*

$$d\overleftarrow{X}_t = \frac{1}{2}\sigma(T-t)^2\Pi(\overleftarrow{X}_t)\left[\nabla f(\overleftarrow{X}_t) + 2\nabla\ln q_{T-t}(\overleftarrow{X}_t)\right]dt + \frac{1}{2}\sigma(T-t)^2\mathcal{H}(\overleftarrow{X}_t)dt$$
$$- \underbrace{\alpha\sigma(T-t)^2\nabla J(\overleftarrow{X}_t)^T G^\dagger(\overleftarrow{X}_t)J(\overleftarrow{X}_t)}_{Landing\ term}dt + \sigma(T-t)\Pi(\overleftarrow{X}_t)\circ d\bar{W}_t.$$

*By assuming $\sigma(t)$ constant and $X_0 \in \Sigma$, OLLA has the stationary distribution $q_\Sigma \propto \exp(-f(x))$ with respect to $d\sigma_\Sigma$.*

**Constrained Underdamped Langevin Dynamics via Landing (ULLA).** In the underdamped case, we are required to satisfy both the target property $dJ(X_t) = -\alpha\sigma(t)^2 J(X_t)dt$ and also the momentum tangency constraint $\nabla J(X_t)P_t = 0$ so that $(X_t, P_t) \in T^*\Sigma$ for $t \geq 0$. Therefore, we control the two Lagrangian processes $d\lambda_t, d\mu_t$ to impose such constraints, and the resulting solution of Lagrangian processes produces the following closed SDE for ULLA:

**Proposition 2** (Construction, stationarity, and backward process of ULLA). *Consider the following Lagrangian form constrained underdamped Langevin of $(X_t, P_t) \sim q_t$:*

$$\begin{cases} dX_t = \sigma(t)^2 P_t dt + \nabla J(X_t)^T d\lambda_t \\ dP_t = -\sigma(t)^2 \nabla f(X_t)dt - \sigma(t)^2\gamma P_t dt + \sigma(t)\sqrt{2\gamma}\circ dW_t + \nabla J(X_t)^T d\mu_t, \end{cases} \tag{2}$$

*where $d\lambda_t, d\mu_t$ are the adapted processes such that $dJ(X_t) = -\alpha\sigma(t)^2 J(X_t)$ (position constraint) and $\nabla J(X_t)P_t = 0$ (momentum tangency constraint), respectively.*

*Assuming $\nabla J(X_0)P_0 = 0$, the explicit solution of $d\lambda_t, d\mu_t$ provides the closed form SDE of (2) as follows:*

$$\begin{cases} dX_t = \sigma(t)^2 P_t dt - \alpha\sigma(t)^2\nabla J(X_t)^T G^\dagger(X_t)J(X_t)dt \\ dP_t = \Pi(X_t)\left[-\sigma(t)^2\nabla f(X_t) - \sigma(t)^2\gamma P_t dt + \sigma(t)\sqrt{2\gamma}dW_t\right] \\ \qquad - \sigma(t)^2\nabla J(X_t)^T G^\dagger(X_t)\left[\mathcal{H}_1(X_t, P_t) - \alpha\mathcal{H}_2(X_t, P_t)\right]dt, \end{cases}$$

*where $\mathcal{H}_1 \in \mathbb{R}^{m+|I_x|}, \mathcal{H}_2 \in \mathbb{R}^{(m+|I_x|)\times(m+|I_x|)}$ are the curvature correction terms defined as*

$$[\mathcal{H}_1(x,p)]_i := p^T \nabla^2 J_i(x)p, \quad [\mathcal{H}_2(x,p)]_i := p^T \nabla^2 J_i(x)(\nabla J(x)^T G^\dagger(x)J(x))$$

*with $[\mathcal{H}_1(x,p)]_i, [\mathcal{H}_2(x,p)]_i$ being the $i$th entry and column of $\mathcal{H}_1(x,p), \mathcal{H}_2(x,p)$ respectively. Furthermore, the backward process $\overleftarrow{X}_t$ of ULLA is given as:*

$$\begin{cases} d\overleftarrow{X}_t = -\sigma(T-t)^2 \overleftarrow{P}_t dt \underbrace{-\alpha\sigma(T-t)^2 \nabla J(\overleftarrow{X}_t)G^\dagger(\overleftarrow{X}_t)J(\overleftarrow{X}_t)dt}_{\text{Landing term}} \\ \\ d\overleftarrow{P}_t = \Pi(\overleftarrow{X}_t)\left[\sigma(T-t)^2\left[\nabla f(\overleftarrow{X}_t)dt + \gamma\overleftarrow{P}_t dt + 2\gamma\nabla_p \ln q_{T-t}(\overleftarrow{X}_t, \overleftarrow{P}_t)\right]dt\right] + \\ \\ \sigma(T-t)^2 \nabla J(\overleftarrow{X}_t)^T G^\dagger(\overleftarrow{X}_t)\left[\mathcal{H}_1(\overleftarrow{X}_t, \overleftarrow{P}_t) + \alpha\mathcal{H}_2(\overleftarrow{X}_t, \overleftarrow{P}_t)\right]dt + \sigma(T-t)\sqrt{2\gamma}\Pi(\overleftarrow{X}_t)d\bar{W}_t. \end{cases}$$

*By assuming $\sigma(t)$ constant and $X_0 \in \Sigma, \nabla J(X_0)P_0 = 0$, ULLA has the stationary distribution $q_{T^*\Sigma} \propto \exp(-f(x) - \frac{1}{2}\|p\|^2)$ with respect to $d\sigma_{T^*\Sigma}$.*

**Error decomposition and benefits of ULLA.** Theoretically, generation error via backward process decomposes into mixing, discretization, and score estimation terms ($W_2(q_0, p_0^\theta) \leq \mathcal{E}_{\text{mix}} + \mathcal{E}_{\text{disc}} + \mathcal{E}_{\text{score}}$). In this point of view, ULLA significantly reduces $\mathcal{E}_{\text{mix}}$ via ballistic dynamics, accelerating convergence and enabling smaller trajectory lengths $N$ compared to OLLA. Additionally, the momentum variable mitigates score singularities near $t = 0$, yielding a potentially smoother training objective for $\mathcal{E}_{\text{score}}$. We refer to Remark 3 for a detailed discussion.

### 4.2 TRANSITION KERNELS FOR FORWARD / BACKWARD PROCESS

In this section, we outline the discretization of the OLLA and ULLA processes. For the notations, we let $x_k$ for $k \in \{1, ..., N\}$ to be the position vector at $k$-th discrete step of the diffusion process, and set $q_k, p_k, p_k^\theta$ to be the marginal probability densities for forward and backward processes, and the parametrized backward process of $x_k$. Also, we set $p_N, \rho(\cdot|x)$ to be the prior of position and momentum where the momentum prior is given by $\Pi(x)\zeta \sim \rho(\cdot|x)$ with $\zeta \sim \mathcal{N}(0, I_d)$. For detailed derivations of the discretization schemes below, we refer to subsection B.4.

**Discretization of OLLA.** For OLLA, the discretization is straightforward. We employ a standard Euler-Maruyama scheme to integrate the corresponding SDE as follows:

$$\begin{cases} x_{k+1} = x_k + \frac{\sigma_k^2 \Delta t}{2}\left[-\Pi(x_k)\nabla f(x_k) - \alpha\sigma_k^2(\nabla J^T G^\dagger J)(x_k)\right] + \sigma_k\sqrt{\Delta t}\Pi(x_k)\zeta_k + \kappa_k^O \\ \\ x_k = x_{k+1} + \frac{\sigma_{k+1}^2 \Delta t}{2}\Pi(x_{k+1})\left[\nabla f(x_{k+1}) + s_{k+1}(x_{k+1})\right] + \sigma_{k+1}\sqrt{\Delta t}\Pi(x_{k+1})\zeta_{k+1} \\ \qquad - \alpha\sigma_{k+1}^2 \Delta t(\nabla J^T G^\dagger J)(x_{k+1})\zeta_{k+1} + \kappa_{k+1}^O \end{cases}$$

where $\kappa_k^O := \frac{\sigma_k^2}{2}\mathcal{H}(x_k)\Delta t$ is the mean curvature term and $s_k(x_k) := 2\nabla \ln q_k(x_k)$, which can be learned by a neural network $s_{k+1}^\theta$.

**Discretization of ULLA.** For ULLA, we adopt a specialized scheme to achieve **2× memory efficiency**, which becomes critical for storing long forward trajectories during training. The method is based on the 1st order non-symmetric OBA splitting integrator. To eliminate the need to explicitly store the momentum trajectory, we first use an approximated $\tilde{B}$ step, which relies on the before-O step momentum on correction terms $\mathcal{H}_1, \mathcal{H}_2$, rather than after, and secondly, we leverage the recursive nature of the update rule to express the momentum at step $k$ as a function of positions at previous steps, using an approximated momentum vector $\tilde{p}_k$. This "collapses" the dynamics into the 2nd order Markov chain solely depending on position variables $x_k$ as follows:

$$\begin{cases} x_{k+1} = x_k + \sigma_k^2 \Delta t\Pi(x_k)\left[a_k\tilde{p}_k^{\text{fwd}} - \sigma_k^2 \Delta t\nabla f(x_k) + \sqrt{1-a_k^2}\zeta_k\right] - \alpha\sigma_k^2 \Delta t(\nabla J^T G^\dagger J)(x_k) \\ \qquad + \kappa_{k,\text{fwd}}^U \\ x_k = x_{k+1} - \sigma_{k+1}^2 \Delta t\Pi(x_{k+1})\left[a_{k+1}\tilde{p}_{k+1}^{\text{bwd}} + \sigma_{k+1}^2 \Delta t\left[\nabla f(x_{k+1}) + s_{k+1}(x_{k+1}, \tilde{p}_{k+1}^{\text{bwd}})\right]\right] \\ \qquad + \sigma_{k+1}^2 \Delta t\sqrt{1-a_{k+1}^2}\Pi(x_{k+1})\zeta_{k+1}' - \alpha\sigma_{k+1}^2 \Delta t(\nabla J^T G^\dagger J)(x_{k+1}) + \kappa_{k+1,\text{bwd}}^U, \end{cases}$$

where $a_k = e^{-\gamma \sigma_k^2 \Delta t} \in [0, 1]$ is a decaying factor, the approximated momentum are defined by $\tilde{p}_k^{\text{fwd}} := \Pi(x_k) \left( (x_k - x_{k-1})/(\sigma_{k-1}^2 \Delta t) \right)$, $\tilde{p}_{k+1}^{\text{bwd}} := \Pi(x_{k+1}) \left( (x_{k+2} - x_{k+1})/(\sigma_{k+2}^2 \Delta t) \right)$, and the curvature correction terms are provided as:

$$\begin{cases} \kappa_{k,\text{fwd}}^U := -\sigma_k^4 \Delta t^2 \nabla J(x_k)^T G^\dagger(x_k) \left[ \mathcal{H}_1(x, \tilde{p}_k^{\text{fwd}}) - \mathcal{H}_2(x_k, \tilde{p}_k^{\text{fwd}}) \right] \\ \kappa_{k+1,\text{bwd}}^U := -\sigma_{k+1}^4 \Delta t^2 \nabla J(x_{k+1})^T G^\dagger(x_{k+1}) \left[ \mathcal{H}_1(x, \tilde{p}_{k+1}^{\text{bwd}}) + \alpha \mathcal{H}_2(x_{k+1}, \tilde{p}_{k+1}^{\text{bwd}}) \right]. \end{cases}$$

Similarly, we set $s_{k+1}(x_{k+1}, \tilde{p}_{k+1}^{\text{bwd}}) := 2\gamma \left( \nabla_p \ln q_{k+1}(x_{k+1}, \tilde{p}_{k+1}^{\text{bwd}}) + \tilde{p}_{k+1}^{\text{bwd}} \right)$ and train a neural network $s_{k+1}^\theta$ to approximate this.

**Remark 1** (Discretization of Constrained Langevin dynamics via Newton solver)**.** Projection-based variants of proposed methods, denoted OLLA-P and ULLA-P, can be obtained by dropping all normal landing and correction terms, and instead solving a Lagrangian multiplier system at each step so that $h(x_k) = 0$. For the detailed derivation, we refer to subsection B.4.

### 4.3 CONDITIONAL WASSERSTEIN PATH MATCHING (CWPM)

Since the proposed OLLA and ULLA do not use per-step projections, intermediate samples $x_k$ may exhibit minor constraint violations and lie off $\Sigma$. This renders previously proposed training loss, such as DT-ELBO [34] or score matching [24, 25], theoretically unstable, as they rely on the assumption that $x_k \in \Sigma$, which can lead to singularity issues. This problem is particularly acute when measuring the NLL loss, where small violations can introduce substantial bias and undermine its reliability as a sample quality metric. To resolve these theoretical issues, we propose the CWPM framework as below, which is based on the Wasserstein distance rather than KL-divergence, eliminating such theoretical singularities. The derivation involves the relationship between Gelbrich distance and 2-Wasserstein distance [37, 38], and refer to Appendix D for detailed proofs and assumptions.

**Theorem 1** (CWPM variational bound – overdamped, informal)**.** *Let $T_{k+1}^\theta = p^\theta(x_k|x_{k+1})$ be the backward transition kernel of the discretized OLLA and define the circuitous density at step $k$ as*

$$\sigma_k := q_k T_k^\theta T_{k-1}^\theta, ..., T_1^\theta, \quad \sigma_0 := q_0.$$

*Assuming existence of $\Lambda_{k+1} > 0$ such that*

$$W_2(\sigma_k, \sigma_{k+1}) \leq \Lambda_{k+1} W_2(q_k, q_{k+1} T_{k+1}^\theta) + \mathcal{O}(\sqrt{\Delta t}), \quad k \in \{0, ..., N-1\}$$

*which holds under minor regularity assumptions on the score and constraint functions (Lemma D.1), we have*

$$W_2(q_0, p_0^\theta) \leq \mathbb{E}_{q(x_{0:N})} \left[ \sum_{k=0}^{N-1} \|\Pi(x_{k+1})(x_k - \mu_{k+1}^o(x_{k+1}))\|^2 \right] + C^o,$$

*where $\mu_{k+1}^o(x_{k+1})$ is the tangential part mean of the parametrized backward process of OLLA defined by*

$$\mu_{k+1}^o(x_{k+1}) := x_{k+1} + \frac{\sigma_{k+1}^2 \Delta t}{2} \Pi(x_{k+1}) \left[ \nabla f(x_{k+1}) + s_\theta^{k+1}(x_{k+1}) \right]$$

*and $C^o$ is a constant independent of $\theta$.*

Similarly, we have the following version for the underdamped case:

**Theorem 2** (CWPM variational bound – underdamped, informal)**.** *Let $y_k = (x_k, x_{k+1}) \in \mathbb{R}^{2d}$ where $x_k \sim q_k, x_{k+1} \sim q_{k+1}$. Define $\bar{q}_k$ to be the law of $y_k$ and set $\bar{T}_{k+1}^\theta = p^\theta(y_k|y_{k+1})$ to be the associated backward transition kernel to $y_k$. We set the circuitous density at step $k$ as*

$$\bar{\sigma}_k := \bar{q}_k \bar{T}_k^\theta \bar{T}_{k-1}^\theta, ..., \bar{T}_1^\theta, \quad \bar{\sigma}_0 := \bar{q}_0.$$

*Assuming existence of $\bar{\Lambda}_{k+1} > 0$ such that*

$$W_2(\bar{\sigma}_k, \bar{\sigma}_{k+1}) \leq \bar{\Lambda}_{k+1} W_2(\bar{q}_k, \bar{q}_{k+1} \bar{T}_{k+1}^\theta) + \mathcal{O}(\Delta t), \quad k \in \{0, ..., N-1\},$$

*which holds under minor regularity assumptions on the score and constraint functions (Lemma D.2), we have*

$$W_2(q_0, p_0^\theta) \leq \mathbb{E}_{q(x_{0:N})}\mathbb{E}_{\rho_N(p_N|x_N)}\left[\sum_{k=0}^{N-1}\|\Pi(x_{k+1})(x_k - \mu_{k+1}^u(x_{k+1}, x_{k+2}))\|^2\right] + C^u$$

*where $\mu_{k+1}^u(x_{k+1})$ is the tangential part mean of the parametrized backward process of ULLA defined by*

$$\mu_{k+1}^u := x_{k+1} - \sigma_{k+1}^2\Delta t\Pi(x_{k+1})\left[a_{k+1}\tilde{p}_{k+1}^{bwd} + \sigma_{k+1}^2\Delta t\left[\nabla f(x_{k+1}) + s_\theta^{k+1}(x_{k+1}, \tilde{p}_{k+1}^{bwd})\right]\right]$$

*and $C^u$ is a constant independent of $\theta$.*

**Choice of Training Loss.**   We remark that other works on diffusion models [1, 39, 40] demonstrated that choosing the training loss weight $\lambda(k)$ proportional to the inverse of the variance up to a proportionality constant of $1/2$ (in our case, $\lambda(k) = 1/(2\sigma_{k+1}^2\Delta t)$ for overdamped and $\lambda(k) = 1/(2\sigma_{k+1}^4\Delta t^2(1 - a_{k+1}^2))$ for underdamped) is helpful for training the score network. Notably, the resulting training losses

$$\begin{cases} L_{\text{CWPM}}^{\text{over}}(\theta) = \mathbb{E}_{q(x_{0:N})}\left[\sum_{k=0}^{N-1}\frac{\|\Pi(x_{k+1})\left(x_k - \mu_{k+1}^o(x_{k+1})\right)\|^2}{2\sigma_{k+1}^2\Delta t}\right] \\[4mm] L_{\text{CWPM}}^{\text{under}}(\theta) = \mathbb{E}_{q(x_{0:N})}\mathbb{E}_{\rho_N(p_N|x_N)}\left[\sum_{k=0}^{N-1}\frac{\|\Pi(x_{k+1})\left(x_k - \mu_{k+1}^u(x_{k+1}, x_{k+2})\right)\|^2}{2\sigma_{k+1}^4\Delta t^2(1 - a_{k+1}^2)}\right] \end{cases}$$

lead to exactly the same training loss provided in DT-ELBO ([34], or Lemma C.1 and Lemma C.2) without the requirement $x_k \in \Sigma$. For the complete algorithms, refer to Algorithm 1 and Algorithm 2.

## 5 EXPERIMENTS

We evaluate on benchmarks largely following RDDPM [34]—Earth/climate datasets, mesh data, the $SO(10)$ manifold, and Alanine dipeptide—and add a 7-Degree of Freedom (DOF) robot arm trajectory task. We compare against constrained generative model algorithms such as RFM [26] and RDDPM, as well as Euclidean forward with backward variants baselines to highlight the importance of handling intrinsic geometry and learning the score function on $\Sigma$. We leave experimental setup, baseline descriptions, and hyperparameters in Appendix E. Also, following a similar technique in a landing-based constrained sampling [41], we set the correction terms $\kappa = 0$ to save computational cost by avoiding Hessian-related terms.

### 5.1 EQUALITY-ONLY SCENARIO TASKS

**Earth and climate science datasets.**   This benchmark lives on the 2-sphere $S^2$, where nearest-point projection is globally available, so landing-based dynamics are not strictly required.

Nevertheless, we use this dataset to (i) assess the intrinsic benefits of **underdamped** dynamics and (ii) quantify the sampling quality – computational cost trade-off under landing. From Figure 1, due to faster mixing of the underdamped dynamics, underdamping markedly reduce the needed forward length $N$: ULLA-P is stable without projection failure even at $N = 40$, whereas RD-DPM (OLLA-P) requires at least $N \approx 150$ to avoid failures. Thus, smaller $N$ yields large training-time savings while preserving comparable sample quality. Although exact projections are available here, Table 1 indicate that ULLA incurs comparable sampling quality under minor constraint violations; visual comparisons of ULLA (subsection E.3) show similar generated distribution to projection-based methods, supporting the practical value of landing.

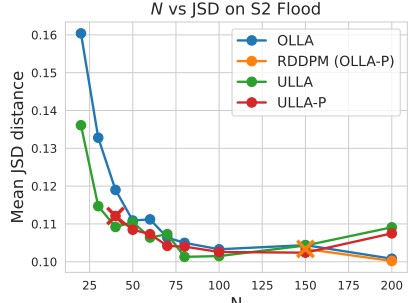

Figure 1: Mean JSD on $S^2$ flood versus trajectory length $N$. Cross mark ($\times$) indicates the smallest $N$ values after which projection failures no longer occur during the forward process.

Table 1: **Generative performance on Earth & Climate distributions (Top) and Mesh histograms (Bottom).** We report the Jensen-Shannon Distance (JSD) computed on 2D spherical coordinate $(\theta, \phi)$ histograms (Earth & Climate) and mesh face histograms (Mesh) (lower is better). The rightmost columns report the average of total training wall-clock time in seconds across all experiments; values in parentheses denote the time spent on forward trajectory simulation during training (or ODE solving time for RFM). We also report the average equality violation $|h|$ of generated samples. Mean $\pm$ standard errors are calculated over 5 independent runs. Across both domains, our landing-based methods (OLLA, ULLA) achieve comparable JSDs while being significantly faster than Riemannian-based diffusion model baselines.

| | **Volcano** | **Earthquake** | **Flood** | **Fire** | **Training time** | **Avg. $|h|$** |
|---|---|---|---|---|---|---|
| *Method / dataset size* | (827) | (6120) | (4875) | (12809) | (Sampling time) | |
| | | | | | | |
| *Riemannian-based* | | | | | | |
| RFM ($N=1000$) | $0.116_{\pm.002}$ | $0.089_{\pm.001}$ | $0.108_{\pm.002}$ | $0.058_{\pm.001}$ | 4019 (0) | $5.24 \times 10^{-8}$ |
| RDDPM ($N=400$) | $0.123_{\pm.004}$ | $0.093_{\pm.002}$ | $0.106_{\pm.002}$ | $0.051_{\pm.001}$ | 12388 (3302) | $1.67 \times 10^{-8}$ |
| | | | | | | |
| *Euclidean fwd. + bwd. variant* | | | | | | |
| Euclidean ($N=50$) | $0.158_{\pm.005}$ | $0.163_{\pm.004}$ | $0.135_{\pm.016}$ | $0.140_{\pm.001}$ | 1130 (221) | $4.14 \times 10^{-2}$ |
| Projected ($N=50$) | $0.156_{\pm.005}$ | $0.152_{\pm.003}$ | $0.133_{\pm.012}$ | $0.133_{\pm.001}$ | 1026 (221) | $2.11 \times 10^{-8}$ |
| Lagrangian ($N=50$) | $0.156_{\pm.004}$ | $0.152_{\pm.003}$ | $0.137_{\pm.011}$ | $0.133_{\pm.001}$ | 1027 (220) | $8.47 \times 10^{-10}$ |
| Guided ($N=50$) | $0.160_{\pm.006}$ | $0.174_{\pm.003}$ | $0.146_{\pm.021}$ | $0.158_{\pm.002}$ | 1048 (220) | $2.07 \times 10^{-2}$ |
| | | | | | | |
| *Ours* | | | | | | |
| OLLA ($N=100$) | $0.128_{\pm.005}$ | $0.096_{\pm.002}$ | $0.103_{\pm.002}$ | $0.060_{\pm.001}$ | 1686 (749) | $3.88 \times 10^{-9}$ |
| ULLA-P ($N=100$) | $0.122_{\pm.007}$ | $0.092_{\pm.001}$ | $0.103_{\pm.002}$ | $0.053_{\pm.001}$ | 1631 (1154) | $8.41 \times 10^{-9}$ |
| ULLA-P ($N=50$) | $0.126_{\pm.005}$ | $0.096_{\pm.001}$ | $0.108_{\pm.001}$ | $0.064_{\pm.002}$ | 1018 (568) | $1.37 \times 10^{-8}$ |
| ULLA ($N=50$) | $0.125_{\pm.005}$ | $0.099_{\pm.002}$ | $0.110_{\pm.001}$ | $0.069_{\pm.002}$ | 1021 (530) | $2.14 \times 10^{-9}$ |

| **Mesh Data (Grid Laplacian Eigenfunctions with index $k$)** | | | | | | |
|---|---|---|---|---|---|---|
| | **Stanford Bunny** | | **Spot the Cow** | | **Training time** | **Avg. $|h|$** |
| *Method* | $k=50$ | $k=100$ | $k=50$ | $k=100$ | (Sampling time) | |
| | | | | | | |
| *Riemannian-based* | | | | | | |
| RFM ($N=1000$) | $0.035_{\pm.001}$ | $0.047_{\pm.001}$ | $0.043_{\pm.002}$ | $0.050_{\pm.002}$ | 145424 (112244) | $1.47 \times 10^{-4}$ |
| RDDPM ($N=400$) | $0.032_{\pm.001}$ | $0.034_{\pm.000}$ | $0.046_{\pm.001}$ | $0.034_{\pm.001}$ | 1916 (126.4) | $1.56 \times 10^{-5}$ |
| | | | | | | |
| *Euclidean fwd. + bwd. variant* | | | | | | |
| Euclidean ($N=30$) | $0.040_{\pm.001}$ | $0.047_{\pm.001}$ | $0.048_{\pm.001}$ | $0.063_{\pm.001}$ | 212 (0.3) | $4.20 \times 10^{-2}$ |
| Projected ($N=30$) | $0.049_{\pm.000}$ | $0.051_{\pm.001}$ | $0.057_{\pm.001}$ | $0.068_{\pm.001}$ | 210 (0.3) | $6.36 \times 10^{-8}$ |
| Lagrangian ($N=30$) | $0.047_{\pm.001}$ | $0.050_{\pm.001}$ | $0.056_{\pm.001}$ | $0.067_{\pm.001}$ | 233 (0.3) | $1.13 \times 10^{-4}$ |
| Guided ($N=30$) | $0.068_{\pm.005}$ | $0.050_{\pm.002}$ | $0.051_{\pm.001}$ | $0.065_{\pm.001}$ | 221 (0.3) | $8.23 \times 10^{-1}$ |
| | | | | | | |
| *Ours* | | | | | | |
| OLLA ($N=100$) | $0.030_{\pm.000}$ | $0.032_{\pm.001}$ | $0.047_{\pm.001}$ | $0.035_{\pm.001}$ | 642 (4.0) | $3.44 \times 10^{-6}$ |
| ULLA-P ($N=50$) | $0.040_{\pm.001}$ | $0.038_{\pm.001}$ | $0.040_{\pm.001}$ | $0.035_{\pm.001}$ | 387 (10.5) | $4.20 \times 10^{-5}$ |
| ULLA ($N=30$) | $0.029_{\pm.001}$ | $0.033_{\pm.001}$ | $0.044_{\pm.001}$ | $0.036_{\pm.001}$ | 360 (2.0) | $1.52 \times 10^{-7}$ |

**3D Mesh data on learned manifold.** Unlike the 2-sphere, meshes lie on manifolds where nearest-point projection is not globally defined and, in our benchmark, must be approximated by a Newton solver because the constraint $h(x) = 0$ is represented by a learned neural network - making projection-based sampling computationally expensive. In this regime, landing becomes particularly effective: as shown in Table 1, ULLA and ULLA-P show comparable JSD of RDDPM and RFM with far fewer steps ($N = 30, 50$), yielding $5\times$ faster training and up to $47\times$ faster sampling than RDDPM. The gains stem from the combinations of the following facts: (i) the underdamped dynamics permits much smaller $N$, and (ii) landing (particularly without curvature corrections) requires only a single constraint-gradient evaluation per step, avoiding iterative projections. These improvements indicate that, for complex learned manifolds where projection is expensive, ULLA provides a scalable and efficient alternative.

**High-dimensional special orthogonal group: SO(10).** This experiment evaluates scalability on the high-dimensional Lie group $SO(10) \subset \mathbb{R}^{100}$, defined by 55 equality constraints ($X^T X = I$); $\det(X) = 1$ condition is checked based on rejection. The synthetic distribution is multimodal with $m$ modes, and sampling quality is assessed by power-trace statistics. As shown in Figure 2 and subsection E.3, landing-based methods (ULLA/ULLA-P/OLLA) remain efficient on this complex manifold, producing high-quality samples with a forward trajectory length of $N = 50$, whereas RDDPM requires at least $N \approx 150$ to avoid projection failures.

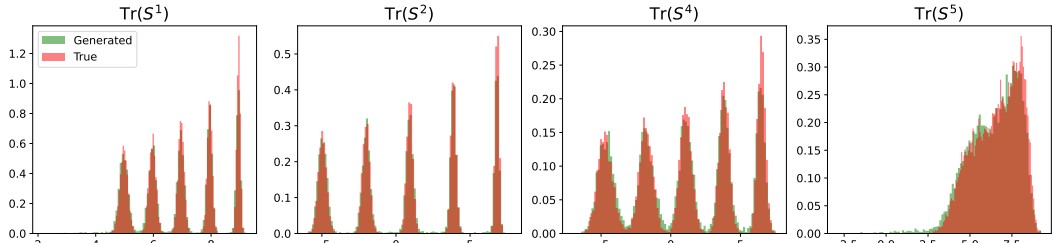

Figure 2: Histograms of the power-trace statistics $\mathsf{Tr}\left(S^k\right)$ for $k \in \{1, 2, 4, 5\}$ on $SO(10)$ with $m = 5$. Green: samples generated by ULLA; Red: ground truth. Forward trajectory length $N = 50$. The close overlap indicates that ULLA matches the target marginal on $\mathsf{Tr}\left(S^k\right)$.

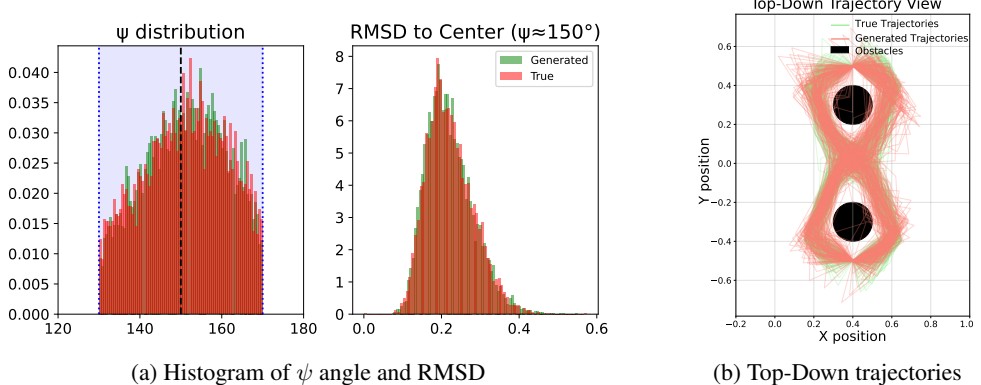

(a) Histogram of $\psi$ angle and RMSD

(b) Top-Down trajectories

Figure 3: Comparison of constrained molecule / robot arm trajectory generation: (a) histogram of $\psi$ angle / Root Mean Square Deviation (RMSD) to center reference point; blue shaded area is feasible region by inequality constraints $\psi \in [130°, 170°]$, (b) top-down trajectory view of generated and ground-truth trajectory samples in the 7-DOF robot arm experiment.

## 5.2 MIXED SCENARIO TASKS

**Alanine dipeptide and 7-DOF robot arm.** We further evaluate our landing algorithms under complicated mixed constraints setup.

The provided feasible set $\Sigma$ are defined by complex intersections of equality and inequality constraints. In these settings, exact projections are often numerically unstable or computationally prohibitive. As summarized in Table 4, standard baselines encounter significant difficulties: the *Projected* Euclidean variant failed in the high-dimensional 7-DOF robot arm task due to severe projection failures, while the *Lagrangian* Euclidean variant failed to converge to a high-quality distribution in the Dipeptide task. For similar issues, OLLA, OLLA-P, ULLA-P failed in this setup.

In contrast, our proposed ULLA method demonstrates superior performance compared to the valid Euclidean forward-backward variants. ULLA not only achieves significantly lower JSDs, consistently outperforming Euclidean baselines even as the dimension scales, but also

Table 2: Effect of $\alpha$ on JSD, $\mathbb{E}[|h|]$

| $\alpha$ | JSD | $\mathbb{E}[|h(x)|]$ |
|---|---|---|
| 1.0 | $0.134_{\pm 0.004}$ | $5.51 \times 10^{-3}$ |
| 10.0 | $0.053_{\pm 0.004}$ | $1.02 \times 10^{-3}$ |
| 50.0 | $0.033_{\pm 0.002}$ | $7.20 \times 10^{-5}$ |
| 100.0 | $0.051_{\pm 0.003}$ | $3.70 \times 10^{-5}$ |

Table 3: Effect of $\epsilon$ on JSD, $\mathbb{E}[g^+]$

| $\epsilon$ | JSD | $\mathbb{E}[g^+(x)]$ |
|---|---|---|
| 0.01 | $0.048_{\pm 0.002}$ | $5.49 \times 10^{-7}$ |
| 0.05 | $0.033_{\pm 0.002}$ | $7.95 \times 10^{-8}$ |
| 0.1 | $0.052_{\pm 0.007}$ | $5.26 \times 10^{-4}$ |
| 0.5 | $0.081_{\pm 0.009}$ | $1.98 \times 10^{-1}$ |

Table 4: **Generative performance on Mixed Constraint Tasks.** The top panel presents the scalability analysis on the 7-DOF robot arm trajectory generation with increasing ambient dimensions $d$. For this task, JSD is measured on 2D histograms formed by projecting generated data via PCA fitted on the ground truth. The *Projected* method is excluded here due to severe projection failures during sampling. The bottom panel reports performance on Alanine Dipeptide conformation generation, measuring JSD on the marginal histograms of the dihedral angle $\psi$ and the RMSD to a reference structure. Constraint violations ($|h|, |g^+|$) are averaged over generated samples. The *Lagrangian* method is omitted in this task due to failure of convergence.

| 7-DOF Robot Arm Trajectory (Scalability with dimension $d$) | | | | | | |
|---|---|---|---|---|---|---|
| | **JSD across dimension ($d$)** | | | | **Avg. $|h|$** | **Avg. $|g^+|$** |
| *Method* | $d = 140$ | $d = 280$ | $d = 420$ | $d = 560$ | | |
| *Euclidean fwd. + bwd. variant* | | | | | | |
| Euclidean ($N = 100$) | $0.498_{\pm.028}$ | $0.656_{\pm.034}$ | $0.647_{\pm.022}$ | $0.750_{\pm.025}$ | $8.3 \times 10^{-1}$ | $5.3 \times 10^{-3}$ |
| Lagrangian ($N = 100$) | $0.769_{\pm.043}$ | $0.816_{\pm.005}$ | $0.831_{\pm.001}$ | $0.831_{\pm.002}$ | $4.9 \times 10^{-2}$ | $1.6 \times 10^{-3}$ |
| Guided ($N = 100$) | $0.499_{\pm.028}$ | $0.655_{\pm.041}$ | $0.665_{\pm.013}$ | $0.740_{\pm.033}$ | $8.0 \times 10^{-1}$ | $4.9 \times 10^{-3}$ |
| *Ours* | | | | | | |
| ULLA ($N = 100$) | $0.275_{\pm.011}$ | $0.295_{\pm.006}$ | $0.366_{\pm.005}$ | $0.391_{\pm.012}$ | $1.8 \times 10^{-5}$ | $1.5 \times 10^{-9}$ |

| Alanine Dipeptide Conformation | | | | |
|---|---|---|---|---|
| *Method* | **JSD ($\psi$ angle)** | **JSD (RMSD)** | **Avg. $|h|$** | **Avg. $|g^+|$** |
| *Euclidean fwd. + bwd. variant* | | | | |
| Euclidean ($N = 100$) | $0.150_{\pm.003}$ | $0.057_{\pm.001}$ | $5.7 \times 10^{-2}$ | $2.4 \times 10^{-3}$ |
| Projected ($N = 100$) | $0.145_{\pm.002}$ | $0.073_{\pm.005}$ | $5.4 \times 10^{-8}$ | $3.3 \times 10^{-3}$ |
| Guided ($N = 100$) | $0.152_{\pm.003}$ | $0.057_{\pm.002}$ | $5.7 \times 10^{-2}$ | $2.4 \times 10^{-3}$ |
| *Ours* | | | | |
| ULLA ($N = 100$) | $0.031_{\pm.002}$ | $0.035_{\pm.002}$ | $1.4 \times 10^{-7}$ | $6.0 \times 10^{-11}$ |

maintains extremely low constraint violations (e.g., avg. $|h| \approx 10^{-5}$ and $|g^+| \approx 10^{-9}$), effectively respecting the complex geometry without the need for expensive multiple projection steps.

**Effect of landing rate $\alpha$ and repulsion rate $\epsilon$.** We analyze how the landing rate $\alpha$ and boundary repulsion rate $\epsilon$ impact generation quality (JSD) and constraint satisfaction on the Alanine Dipeptide task with ULLA (fixing $\epsilon = 0.05$ for $\alpha$ ablation and $\alpha = 50$ for $\epsilon$ ablation). As shown in Table 2, increasing $\alpha$ significantly reduces equality violations and improves JSD by strengthening the drift toward $\Sigma$. However, excessively large $\alpha$ may introduce large discretization error, which can degrade sample quality (see Table 8 for full ablation). Similarly, Table 3 indicates that $\epsilon$ requires a balanced choice: while too small $\epsilon$ causes boundary "stickiness," overly large $\epsilon$ aggressively pushes trajectories inward, distorting the distribution and increasing inequality violations due to numerical instability. Visualizations in Figure 8 confirm that a moderate $\epsilon$ yields the best trade-off.

## 6 CONCLUSION & FUTURE WORKS

We introduce a landing-based overdamped and underdamped Langevin process that avoids costly projections on general constrained sets described by both equality and inequality constraints, resulting in a unified constrained diffusion model framework. By leveraging the fast mixing property of underdamped Langevin dynamics, we shorten the forward trajectory (thus reducing function evaluations and memory), achieving reasonable sample quality while drastically cutting computational cost for training and sampling. A current limitation is that our sampling quality does not consistently and fully surpass projection-based methods. As a promising direction, we suggest training the score network to learn both tangential and normal components – so that normal components, assisted by our landing mechanism, actively reduce constraint violation – potentially improving sample quality without sacrificing efficiency.

## REPRODUCIBILITY STATEMENT

We include anonymized code (training, evaluation, and data preprocessing scripts), exact hyperparameters in Appendix E to reproduce the provided results. We also report hardware specs, software on the Appendix E and all theoretical proofs are provided in the appendix for theoretical completeness.

## ETHICS STATEMENT

This work uses public or synthetic data only, with no human subjects or personal data. The method may be applied to safety-critical domains (e.g., robotics); we caution against deployment without domain-specific evaluation and monitoring.

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

TABLE OF CONTENTS

## A  TABLE OF KEY NOTATION, ADDITIONAL REMARKS, AND ALGORITHMS

Table 5: Table of Key Notations

| Symbol | Definition | Descriptions |
|---|---|---|
| $h$ | $h(x) = [h_1(x), \ldots, h_m(x)]^T$ | Equality constraints |
| $g$ | $g(x) = [g_1(x), \ldots, g_l(x)]^T$ | Inequality constraints |
| $\Sigma$ | $\{x \in \mathbb{R}^d \mid h(x) = 0, g(x) \leq 0\}$ | Constraint manifold |
| $I_x$ | $\{i \in [l] \mid g_i(x) \geq 0\} = \{i_1, .., i_{|I_x|}\}$ | Active index set of inequalities |
| $g_{I_x}$ | $g_{I_x}(x) = [g_{i_1}(x), ... g_{i_{|I_x|}}(x)]^T$ | Active inequality constraints |
| $J(x)$ | $\{h(x)^T, g_{i_1}(x) + \epsilon, ..., g_{i_{|I_x|}}(x) + \epsilon\}^T$ | Constraint-correction vector |
| $\Pi(x)$ | $I - \nabla J(x)^T G(x)^\dagger \nabla J(x)$ | Orthogonal projector onto $T_x\Sigma$ |
| $T_x\Sigma$ | $\{p \in \mathbb{R}^d \mid \nabla h(x)v = 0, \nabla g_{I_x}(x)v = 0\}$ | Tangent space of $\Sigma$ at $x$ |
| $T^*\Sigma$ | $\{(x, p) \in \mathbb{R}^{2d} \mid x \in \Sigma, p \in T_x\Sigma\} \simeq T\Sigma$ | Cotangent bundle of $\Sigma$ |
| $\nabla_\Sigma f$ | $\Pi(x)\nabla f(x)$ | Intrinsic gradient on $\Sigma$ |
| $\mathsf{div}_\Sigma X$ | $\mathsf{Tr}(\Pi(x)\nabla X(x))$ | Intrinsic divergence on $\Sigma$ |
| $d\sigma_\Sigma$ | Surface (Hausdorff) measure of $\Sigma$ | Natural measure on $\Sigma$ |
| $d\sigma_{T^*\Sigma}$ | Liouville measure of $T^*\Sigma$ | Natural measure on $T^*\Sigma$ |
| $G(x)$ | $\nabla J(x)\nabla J(x)^T$ | Gram matrix, full rank assumed |
| $\epsilon$ | Boundary repulsion rate | Controls effect of repulsion. |
| $\alpha$ | Landing rate | Controls constraint decay |
| $\gamma$ | Friction coefficient | Used in ULLA, ULLA-P |
| $\rho_\Sigma$ | Target (stationary) density on $\Sigma$ | Proportional to $\exp(-f)d\sigma_\Sigma$ |
| $\mathsf{KL}^\Sigma(\rho\|\pi)$ | $\int_\Sigma \rho \ln \frac{\rho}{\pi} d\sigma_\Sigma$ | KL-divergence on $\Sigma$ |
| $T, N$ | Continuous and discrete terminal time | Relationship: $T = N\Delta t$ |
| $\sigma(t), \sigma_k$ | $\sigma_{\min} + \frac{t}{T}(\sigma_{\max} - \sigma_{\min}), \quad \sigma_k = \sigma(k\Delta t)$ | Noise schedule function |
| $q_t, p_t^\theta$ | Continuous time marginal densities at $t$ | Forward $q_t$, Backward $p_t^\theta$ |
| $q_k, p_k^\theta$ | Discrete time marginal densities at $k$ | Forward $q_k$, Backward $p_k^\theta$ |
| $p_N, \rho(\cdot|x)$ | Prior distribution of $x$ and $p$ ($p_N$ varies) | $\rho(\cdot|x) \sim \Pi(x)\zeta, \ \zeta \sim \mathcal{N}(0, I)$ |
| $\tilde{p}_k^{\mathsf{fwd}}$ | $\Pi(x_k)\left(\frac{x_k - x_{k-1}}{\sigma_{k-1}^2 \Delta t}\right)$ | Forward approximated momentum |
| $\tilde{p}_k^{\mathsf{bwd}}$ | $\Pi(x_{k+1})\left(\frac{x_{k+2} - x_{k+1}}{\sigma_{k+2}^2 \Delta t}\right)$ | Backward approximated momentum |

**Remark 2** (Comments on relaxed Constant Rank Constraint Qualification (rCRCQ)). In this remark, we further clarify the definition of the relaxed Constant Rank Constraint Qualification (rCRCQ) and its relationship with other constraint qualifications.

We first recall the definitions of the Linear Independence Constraint Qualification (LICQ), Constant Rank Constraint Qualification (CRCQ) [42], and its relaxed version (rCRCQ) [35].

---

**Definition A.1** (LICQ, CRCQ, and rCRCQ; [43]). Let $\Sigma := \left\{ x \in \mathbb{R}^d \mid h(x) = 0, g(x) \leq 0 \right\}$ be the feasible set, and denote $I_x = \left\{ i \in [l] \mid g_i(x) \geq 0 \right\}$ to be the active index set of inequalities.

- **LICQ** [44]: LICQ holds at $x \in \Sigma$ if the set $\{\nabla h_i(x)\}_{i=1}^m \cup \{\nabla g_j(x)\}_{j \in I_x}$ is linearly independent.

- **CRCQ** [42]: CRCQ holds at $x \in \Sigma$ if there exists a neighborhood $U \subset \mathbb{R}^d$ of $x$ such that for *any* subsets of indices $I \subset [m]$ and $J \subset I_x$, the family of gradients $\{\nabla h_i(y)\}_{i \in I} \cup \{\nabla g_j(y)\}_{j \in J}$ has a constant rank for all $y \in U$.

- **rCRCQ** [35]: rCRCQ holds at $x \in \Sigma$ if there exists a neighborhood $U \subset \mathbb{R}^d$ of $x$ such that for any subset of active inequalities $J \subset I_x$, the family of gradients $\{\nabla h_i(y)\}_{i=1}^m \cup \{\nabla g_j(y)\}_{j \in J}$ has a constant rank for all $y \in U$.

---

The core reason for assuming rCRCQ lies in the stability of the SDE coefficients. It is a fundamental result in matrix analysis [45] that the Moore-Penrose pseudo-inverse $A(x)^\dagger$ is continuous at a point $x_0$ if and only if the rank of $A(x)$ is constant in a neighborhood of $x_0$. By assuming rCRCQ, we guarantee that the Jacobian $\nabla J(x)$ maintains locally constant rank (even as the active set changes across strata), which ensures that the pseudo-inverse $G(x)^\dagger$ and the resulting projection operator $\Pi(x)$ are continuous and well-defined. Therefore, this guarantees the drift vector and diffusion matrix of the OLLA and ULLA dynamics to be well defined.

**Hierarchy of Constraint Qualifications.** We remark that, from the variational analysis and optimization literature (e.g., [43]), rCRCQ is a strictly weaker condition than CRCQ, and CRCQ is strictly weaker than LICQ, therefore, their logical implication is as follows:

$$\text{LICQ} \implies \text{CRCQ} \implies \text{rCRCQ}.$$

In particular, rCRCQ can relax the gradient degeneracy problem appearing in LICQ.

To illustrate a case where LICQ fails due to gradient degeneracy while rCRCQ holds, consider a feasible set $\Sigma \subset \mathbb{R}^3$ representing the $z$-axis. It is defined by two equality constraints and one redundant inequality constraint with a nonlinear term:

$$h_1(x) = x_1 = 0, \quad h_2(x) = x_2 = 0,$$
$$g(x) = x_1 + x_2 + x_1^2 \leq 0.$$

On the manifold $\Sigma$ (where $x_1 = x_2 = 0$), the inequality is active since $g(0) = 0$.

- **LICQ fails:** The gradients at the origin $x = 0$ are $\nabla h_1 = (1, 0, 0)^T$, $\nabla h_2 = (0, 1, 0)^T$, and $\nabla g = (1, 1, 0)^T$. We observe that $\nabla g = \nabla h_1 + \nabla h_2$, meaning the gradients are linearly dependent. Thus, the Gram matrix is singular, and LICQ is violated.

- **rCRCQ holds:** Now consider the Jacobian matrix of the active constraints for an arbitrary point $x \in \mathbb{R}^3$:

$$J(x) = \begin{bmatrix} \nabla h_1(x)^T \\ \nabla h_2(x)^T \\ \nabla g(x)^T \end{bmatrix} = \begin{bmatrix} 1 & 0 & 0 \\ 0 & 1 & 0 \\ 1 + 2x_1 & 1 & 0 \end{bmatrix}.$$

Regardless of the location $x$, the rank of $J(x)$ is constant and equal to two in the entire neighborhood, satisfying rCRCQ and ensuring that the projection operator $\Pi(x)$ via the pseudo-inverse $G(x)^\dagger$ remains well-defined and continuous.

**Extended Usage in Our Framework.** While the standard definition of rCRCQ is checking the condition at a point "$x \in \Sigma$", we appropriately extend this usage in our diffusion model context.

In particular, since our landing-based discretized sampling algorithms (OLLA, ULLA) involves noise that may push particles slightly off the manifold, we implicitly assume that this constant rank property extends to an sufficiently large neighborhood of $\Sigma$ which contains all discretized samples $\{X_k\}_{k=0}^N$, or to the entire ambient space $\mathbb{R}^d$. This ensures that the projection operator $\Pi(x)$ and the drift terms are well-defined not just on $\Sigma$, but in the surrounding ambient space $\mathbb{R}^d$ where the landing mechanism operates.

**Remark 3** (Error decomposition and probable benefit of ULLA). Recent theoretical progress on diffusion models (e.g., [46, 47]) suggests that the total generation error can be naturally decomposed into three distinct components. In the 2-Wasserstein distance, this can be viewed as:

$$W_2(q_0, p_0^\theta) \leq \underbrace{\mathcal{E}_{\mathrm{mix}}}_{\text{Mixing}} + \underbrace{\mathcal{E}_{\mathrm{disc}}}_{\text{Discretization}} + \underbrace{\mathcal{E}_{\mathrm{score}}}_{\text{Score estimation}}$$

1. **Discretization error ($\mathcal{E}_{\mathrm{disc}}$) & Mixing error ($\mathcal{E}_{\mathrm{mix}}$):** Regarding discretization, our ULLA implementation employs a memory-efficient first-order splitting scheme; thus, both ULLA and the baseline OLLA share the same convergence order with respect to the step size. However, ULLA gains a significant advantage in the *mixing error* due to the ballistic behavior of underdamped dynamics, which theoretically accelerates convergence to $\mathcal{O}(\sqrt{d}/\epsilon)$ compared to the diffusive $\mathcal{O}(d/\epsilon^2)$ of overdamped dynamics [48, 49]. This allows for a significantly smaller trajectory length $N$ to reach the stationary prior, thereby reducing the computational cost for training and storage.

2. **Score estimation error ($\mathcal{E}_{\mathrm{score}}$):** Employing a constrained forward process with the proposed landing mechanism allows the model to faithfully capture the intrinsic geometry of $\Sigma$. Crucially, because the landing mechanism analytically handles the ill-conditioned normal component, the score network $s_\theta^t$ is only required to learn the smoother tangential component $\Pi(x)s_{\mathrm{true}}^t$ [50]. Adopting underdamped dynamics introduces a trade-off: learning on the extended phase space potentially increases regression complexity compared to position-only models. However, since empirical data distributions are usually supported on some data manifold $\Sigma_{\mathrm{data}} \subset \Sigma$, standard overdamped models suffer from score singularities where $\|s_{\mathrm{true}}^t\|_2 \propto \mathcal{O}(1/t)$ near $t = 0$ [50]. In contrast, as highlighted in Dockhorn et al. [51], underdamped dynamics yield a smoother training objective that bypasses this singularity problem due to the existence of momentum variable.

Figure 4 provides empirical evidence of this effect on the volcano experiment. The underdamped model exhibits Jacobian norms that are several orders of magnitude smaller across all times and, in particular, does *not* show the sharp blow-up near $t \approx 0$ that appears in the overdamped case. This suggests that ULLA provides a numerically better-conditioned score regression problem, which can potentially reduce $\mathcal{E}_{\mathrm{score}}$ in practice.

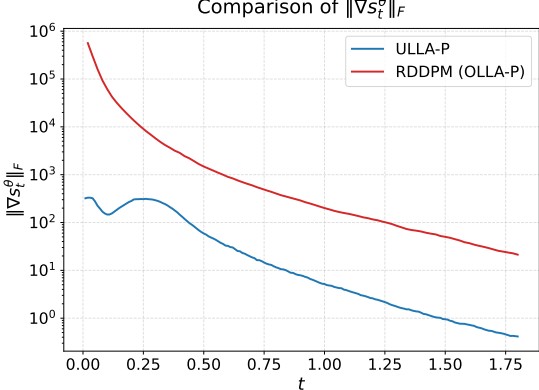

Figure 4: Comparison of the Frobenius norm of the score Jacobian $\|\nabla s_t^\theta\|_F$ over time on the volcano experiment. The overdamped RDDPM (OLLA-P) baseline (red) exhibits very large Jacobian norms and a pronounced singular behavior as $t \to 0$, while the underdamped ULLA-P sampler (blue) remains several orders of magnitude smaller and shows no blow-up near $t \approx 0$. Jacobian is taken over to position $x$ for the overdamped and to momentum $p$ for the underdamped.

---

**Algorithm 1** Full Diffusion Pipeline for OLLA / OLLA-P (=RDDPM [34])

---

1: **Input:** Data distribution $q_{\text{data}}$, initial score network $s_k^\theta(x)$, number of steps $N$, terminal step $N$, landing rate $\alpha$, boundary repulsion rate $\epsilon$, constraints $h, g$.
2: **Options:** mode $\in \{\text{OLLA}, \text{OLLA-P}\}$, use_curvature $\in \{\text{True}, \text{False}\}$
3: **Output:** Trained score network $s_k^\theta(x)$, generated sample $x_0$

---

**Part 1: Forward Process (Noising)** ▷ Run Forward Process per $l_f$ iterations
4: Sample $x_0 \sim q_{\text{data}} = q_0$
5: **for** $k \in \{0, \dots, N-1\}$ **do**
6:     Compute $\nabla f(x_k), J(x_k), \nabla J(x_k), G(x_k)^\dagger, \Pi(x_k)$
7:     $\bar{\mu}_k^o(x_k) \leftarrow x_k - \frac{1}{2}\sigma_k^2 \Delta t \Pi(x_k)\nabla f(x_k)$         ▷ Prior drift term
8:     **if** mode = OLLA-P **then**         ▷ Projection-based noising
9:         $x_{k+1} \leftarrow \text{Proj}_\Sigma(\bar{\mu}_k^o(x_k) + \sigma_k\sqrt{\Delta t}\Pi(x_k)\zeta_k), \quad \zeta_k \sim \mathcal{N}(0, I_d)$
10:     **else**         ▷ Landing-based noising (OLLA)
11:         $\mathcal{H}(x_k) \leftarrow 0$
12:         **if** use_curvature **then**
13:             $\text{Tr} \leftarrow [\text{Tr}(\Pi\nabla^2 J_1), \dots, \text{Tr}(\Pi\nabla^2 J_{m+|I_{x_k}|})]^T$
14:             $\mathcal{H}(x_k) \leftarrow \nabla J(x_k)^T G(x_k)^\dagger \text{Tr}$
15:         **end if**
16:         $L_k(x_k) \leftarrow -\alpha\sigma_k^2\Delta t\nabla J(x_k)^T G(x_k)^{-1}J(x_k)$     ▷ Landing term
17:         $\kappa_k^O(x_k) \leftarrow \frac{1}{2}\sigma_k^2\Delta t\mathcal{H}(x_k)$     ▷ Curvature term
18:         $x_{k+1} \leftarrow \bar{\mu}_k^O(x_k) + L_k(x_k) + \kappa_k^O(x_k) + \sigma_k\sqrt{\Delta t}\Pi(x_k)\zeta_k, \quad \zeta_k \sim \mathcal{N}(0, I_d)$
19:     **end if**
20: **end for**
21: $x_N \leftarrow \text{Proj}_\Sigma(x_N)$     ▷ Terminal projection by Newton's method
22: Store trajectory $\{x_k\}_{k=0}^N$

---

**Part 2: Score Network Training**
23: $L_{\text{CWPM}}^{\text{over}}(\theta) \leftarrow \sum_{k=0}^{N-1} \frac{\|\Pi(x_{k+1})(x_k - \mu_{k+1}^o(x_{k+1}))\|^2}{2\sigma_{k+1}^2\Delta t}$
24: Update network parameters: $\theta \leftarrow \theta - \eta\nabla_\theta L_{\text{CWPM}}^{\text{over}}(\theta)$     ▷ $\eta$ is the learning rate

---

**Part 3: Backward Process (Sampling)**
25: Sample $x_N \sim p_N$ (prior)
26: **for** $k \in \{N, \dots, 1\}$ **do**
27:     Compute $\nabla f(x_k), J(x_k), \nabla J(x_k), G(x_k)^\dagger, \Pi(x_k)$
28:     $\mu_k^o(x_k) \leftarrow x_k + \frac{1}{2}\sigma_k^2\Delta t\Pi(x_k)[\nabla f(x_k) + s_k^\theta(x_k)]$
29:     **if** mode = OLLA-P **then**         ▷ Projection-based variant
30:         $x_{k-1} \leftarrow \text{Proj}_\Sigma\left(\mu_k^o(x_k) + \sigma_k\sqrt{\Delta t}\Pi(x_k)\zeta_k\right)$
31:     **else**         ▷ Landing-based variant (OLLA)
32:         $\mathcal{H}(x_k) \leftarrow 0$
33:         **if** use_curvature **then**
34:             $\text{Tr} \leftarrow [\text{Tr}(\Pi\nabla^2 J_1), \dots, \text{Tr}(\Pi\nabla^2 J_{m+|I_{x_k}|})]^T$
35:             $\mathcal{H}(x_k) \leftarrow \nabla J(x_k)^T G(x_k)^\dagger \text{Tr}$
36:         **end if**
37:         $L_k(x_k) \leftarrow -\alpha\sigma_k^2\Delta t\nabla J(x_k)^T G(x_k)^\dagger J(x_k)$
38:         $\kappa_k^o(x_k) \leftarrow \frac{1}{2}\sigma_k^2\Delta t\mathcal{H}(x_k)$
39:         $x_{k-1} \leftarrow \mu_k^o(x_k) + L_k(x_k) + \kappa_k^o(x_k) + \sigma_k\sqrt{\Delta t}\Pi(x_k)\zeta_k$
40:     **end if**
41: **end for**
42: $x_0 \leftarrow \text{Proj}_\Sigma(x_0)$     ▷ Terminal projection by Newton's method
43: **return** $x_0$

---

---

**Algorithm 2** Full Diffusion Pipeline for ULLA / ULLA-P

---

1: **Input:** Data distribution $q_{\text{data}}$, initial score network $s_k^\theta(x,p)$, number of steps $N$, terminal time $T$, landing rate $\alpha$, boundary repulsion rate $\epsilon$, friction $\gamma$, constraints $h, g$.
2: **Options:** $\texttt{mode} \in \{\text{ULLA}, \text{ULLA-P}\}$, $\texttt{use\_curvature} \in \{\text{True}, \text{False}\}$
3: **Output:** Trained score network $s_k^\theta(x,p)$, generated sample $x_0$

---

**Part 1: Forward Process (Noising)**   $\triangleright$ Run Forward Process per $l_f$ iterations
4: Sample $x_0 \sim q_{\text{data}}$, $p_0 \sim \mathcal{N}(0, I_d)$ and set $\tilde{p}_0 \leftarrow \Pi(x_0)p_0$
5: $x_{-1} \leftarrow x_0 - \sigma_{-1}^2 \Delta t \tilde{p}_0$   $\triangleright$ Create pseudo-point for first momentum
6: **for** $k \in \{0, \dots, N-1\}$ **do**
7:   Compute $\nabla f(x_k), J(x_k), \nabla J(x_k), G(x_k)^\dagger, \Pi(x_k)$
8:   $\tilde{p}_k^{\text{fwd}} \leftarrow \Pi(x_k)\left(\frac{x_k - x_{k-1}}{\sigma_{k-1}^2 \Delta t}\right)$   $\triangleright$ Approximate momentum from positions
9:   $a_k \leftarrow e^{-\gamma \sigma_k^2 \Delta t}$
10:   $\bar{\mu}_k^u(x_k, \tilde{p}_k^{\text{fwd}}) \leftarrow x_k + \sigma_k^2 \Delta t \Pi(x_k)[a_k \tilde{p}_k^{\text{fwd}} - \sigma_k^2 \Delta t \nabla f(x_k)]$   $\triangleright$ Prior drift term
11:   **if** $\texttt{mode} = \text{ULLA-P}$ **then**   $\triangleright$ Projection-based noising
12:     $x_{k+1} \leftarrow \text{Proj}_\Sigma(\bar{\mu}_k^u(x_k, \tilde{p}_k^{\text{fwd}}) + \sigma_k^2 \Delta t \sqrt{1-a_k^2}\,\Pi(x_k)\zeta_k), \quad \zeta_k \sim \mathcal{N}(0, I_d)$
13:   **else**   $\triangleright$ Landing-based noising (ULLA)
14:     $\mathcal{H}_1(x_k, \tilde{p}_k^{\text{fwd}}), \mathcal{H}_2(x_k, \tilde{p}_k^{\text{fwd}}) \leftarrow 0, 0$
15:     **if** $\texttt{use\_curvature}$ **then**
16:       Compute $\mathcal{H}_1, \mathcal{H}_2$ using $x_k, \tilde{p}_k^{\text{fwd}}$
17:     **end if**
18:     $L_k(x_k) \leftarrow -\alpha \sigma_k^2 \Delta t \nabla J(x_k)^T G(x_k)^\dagger J(x_k)$   $\triangleright$ Landing term
19:     $\kappa_{k,\text{fwd}}^U(x_k, \tilde{p}_k^{\text{fwd}}) \leftarrow -\sigma_k^4 \Delta t^2 \nabla J(x_k)^T G^\dagger(x_k)[\mathcal{H}_1 - \alpha\mathcal{H}_2]$   $\triangleright$ Curvature term
20:     $x_{k+1} \leftarrow \bar{\mu}_k^u(x_k, \tilde{p}_k^{\text{fwd}}) + L_k(x_k) + \kappa_{k,\text{fwd}}^U(x_k, \tilde{p}_k^{\text{fwd}}) + \sigma_k^2 \Delta t \sqrt{1-a_k^2}\,\Pi(x_k)\zeta_k$
21:   **end if**
22: **end for**
23: $x_N \leftarrow \text{Proj}_\Sigma(x_N)$   $\triangleright$ Terminal projection by Newton's method
24: Store trajectory $\{x_k\}_{k=0}^N$

---

**Part 2: Score Network Training**
25: $L_{\text{CWPM}}^{\text{under}}(\theta) \leftarrow \sum_{k=0}^{N-1} \frac{\|\Pi(x_{k+1})(x_k - \mu_{k+1}^u(x_{k+1}, x_{k+2}))\|^2}{2\sigma_{k+1}^4 \Delta t^2 (1-a_{k+1}^2)}$
26: Update network parameters: $\theta \leftarrow \theta - \eta \nabla_\theta L_{\text{CWPM}}^{\text{under}}(\theta)$   $\triangleright$ $\eta$ is the learning rate

---

**Part 3: Backward Process (Sampling)**
27: Sample $x_N \sim p_N$ (prior), $p_N \sim \mathcal{N}(0, I_d)$. Set $\tilde{p}_N \leftarrow \Pi(x_N)p_N$.
28: $x_{N+1} \leftarrow x_N + \sigma_N^2 \Delta t \tilde{p}_N$ CommentCreate pseudo-point for terminal momentum
29: **for** $k \in \{N, \dots, 1\}$ **do**
30:   Compute $J(x_k), \nabla J(x_k), G(x_k)^\dagger, \Pi(x_k)$
31:   $\tilde{p}_k \leftarrow \Pi(x_k)\left(\frac{x_{k+1} - x_k}{\sigma_{k+1}^2 \Delta t}\right)$
32:   $a_k \leftarrow e^{-\gamma \sigma_k^2 \Delta t}$
33:   $\mu_k^u(x_k, \tilde{p}_k^{\text{bwd}}) \leftarrow x_k - \sigma_k^2 \Delta t \Pi(x_k)[a_k \tilde{p}_k^{\text{bwd}} + \sigma_k^2 \Delta t(\nabla f(x_k) + s_k^\theta(x_k, \tilde{p}_k^{\text{bwd}}))]$
34:   **if** $\texttt{mode} = \text{ULLA-P}$ **then**   $\triangleright$ Projection-based variant
35:     $x_{k-1} \leftarrow \text{Proj}_\Sigma\left(\mu_k^u(x_k, \tilde{p}_k^{\text{bwd}}) + \sigma_k \sqrt{\Delta t(1-a_k^2)}\,\Pi(x_k)\zeta_k\right)$
36:   **else**   $\triangleright$ Landing-based variant (ULLA)
37:     $\mathcal{H}_1(x_k, \tilde{p}_k^{\text{bwd}}), \mathcal{H}_2(x_k, \tilde{p}_k^{\text{bwd}}) \leftarrow 0, 0$
38:     **if** $\texttt{use\_curvature}$ **then**
39:       Compute $\mathcal{H}_1, \mathcal{H}_2$ using $x_k, \tilde{p}_k^{\text{bwd}}$
40:     **end if**
41:     $L_k(x_k) \leftarrow -\alpha \sigma_k^2 \Delta t \nabla J(x_k)^T G(x_k)^\dagger J(x_k)$
42:     $\kappa_k^U(x_k, \tilde{p}_k^{\text{bwd}}) \leftarrow -\sigma_k^4 \Delta t^2 \nabla J(x_k)^T G^\dagger(x_k)[\mathcal{H}_1 + \alpha\mathcal{H}_2]$
43:     $x_{k-1} \leftarrow \mu_k^u(x_k, \tilde{p}_k^{\text{bwd}}) + L_k(x_k) + \kappa_k^U(x_k, \tilde{p}_k^{\text{bwd}}) + \sigma_k \sqrt{\Delta t(1-a_k^2)}\,\Pi(x_k)\zeta_k$
44:   **end if**
45: **end for**
46: $x_0 \leftarrow \text{Proj}_\Sigma(x_0)$   $\triangleright$ Terminal projection by Newton's method
47: **return** $x_0$

---

## B CONSTRAINED LANGEVIN DYNAMICS

In this section, we review the constrained Langevin dynamics and introduce their landing versions.

### B.1 CONSTRUCTION OF OLLA

**Notations and Background for overdamped setup.** We consider the constrained set $\Sigma :=$ $\left\{ x \in \mathbb{R}^d \mid h(x) = 0, g(x) \leq 0 \right\}$, assumed to be a stratified manifold $\mathbb{R}^d$ with rCRCQ satisfied. We define the stacked active constraint map and its Jacobian as

$$J(x) := [h(x), g_{I_x}(x) + \epsilon] \in \mathbb{R}^{m + |I_x|}, \quad \nabla J(x) \in \mathbb{R}^{(m + |I_x|) \times d}$$

where $I_x$ denotes the set of active inequality constraints, i.e., $I_x := \{i \in [l] \mid g_i(x) \geq 0\}$. Denote the Gram matrix $G(x) := \nabla J(x) \nabla J(x)^T \in \mathbb{R}^{(m + |I_x|) \times (m + |I_x|)}$. The orthogonal projector onto the tangent space of $T_x \Sigma := \left\{ p \in \mathbb{R}^d \mid \nabla J(x) p = 0 \right\}$ is given by $\Pi(x) = I - \nabla J(x)^T G(x)^\dagger \nabla J(x)$. On this manifold $\Sigma$, all intrinsic differential operators are defined via the projector $\Pi$. For a smooth scalar function $\phi$ and smooth vector field $X$ on $\Sigma$, we have

$$\nabla_\Sigma \phi := \Pi \nabla \phi, \quad \mathsf{div}_\Sigma(X) = \mathsf{Tr}\left(\Pi \nabla X\right)$$

and the Laplace-Betrami operator is $\Delta_\Sigma \phi := \mathsf{div}_\Sigma(\nabla_\Sigma \phi)$, where $\Delta$ denotes ambient Euclidean gradient or Jacobian. For comprehensive backgrounds on constrained overdamped Langevin dynamics, see Chapter 3.2 in Rousset et al. [36].

**Proposition B.1** (Construction of OLLA). *Consider the following Lagrangian-form constrained overdamped Langevin dynamics:*

$$dX_t = -\frac{1}{2}\sigma(t)^2 \nabla f(X_t) dt + \sigma(t) \circ dW_t + \nabla J(X_t)^T d\lambda_t \tag{3}$$

*where $d\lambda_t$ is the adapted process such that $dJ(X_t) = -\alpha\sigma(t)^2 J(X_t)$. The explicit minimum norm solution of $d\lambda_t$ is given by*

$$d\lambda_t = G^\dagger(X_t)\left[-\alpha\sigma(t)^2 J(X_t)dt + \frac{1}{2}\sigma(t)^2 \nabla J(X_t)\nabla f(X_t)dt - \sigma(t)\nabla J(X_t) \circ dW_t\right],$$

*with $G(X_t) := \nabla J(X_t)\nabla J(X_t)^T$ defined as the Gram matrix. Therefore, the closed form SDE of (3) is as follows:*

$$dX_t = -\left[\frac{\sigma(t)^2}{2}\Pi(X_t)\nabla f(X_t) + \alpha\sigma(t)^2 \nabla J(X_t)^T G^\dagger(X_t)J(X_t)\right]dt + \frac{\sigma(t)^2}{2}\mathcal{H}(X_t)dt$$
$$+ \sigma(t)\Pi(X_t)dW_t,$$

*where $\mathcal{H}$ is the mean curvature correction term defined as*

$$\mathcal{H}(x) := -\nabla J(x)^T G^\dagger(x)\left[\mathsf{Tr}\left(\nabla^2 J_1(x)\Pi(x)\right), ..., \mathsf{Tr}\left(\nabla^2 J_{m + |I_x|}(x)\Pi(x)\right)\right]^T.$$

*Proof.* From the Stratonovich chain rule, it holds that

$$-\alpha\sigma(t)^2 J(X_t)dt = \nabla J(X_t) \circ dX_t = -\frac{1}{2}\sigma(t)^2 \nabla J(X_t)\nabla f(X_t)dt + \sigma(t)\nabla J(X_t) \circ dW_t + G(X_t)d\lambda_t$$

Among the many solutions $d\lambda_t$ satisfying the above equation, we choose the (unique) minimum norm solution of $d\lambda_t$ process:

$$d\lambda_t = G^\dagger(X_t)\left[-\alpha\sigma(t)^2 J(X_t)dt + \frac{1}{2}\sigma(t)^2 \nabla J(X_t)\nabla f(X_t)dt - \sigma(t)\nabla J(X_t) \circ dW_t\right].$$

We remark that $\nabla J(X_t)^T d\lambda_t$ is unique regardless of the choice of solution $d\lambda_t$. Substituting back to the SDE (3) gives the following Stratonovich version of the unique closed-form SDE:

$$dX_t = -\left[\frac{1}{2}\sigma(t)^2 \Pi(X_t)\nabla f(X_t) + \alpha\sigma(t)^2 \nabla J(X_t)^T G^\dagger(X_t)J(X_t)\right]dt + \sigma(t)\Pi(X_t) \circ dW_t.$$

To recover the Itô version of the closed-form SDE, we observe that the Itô-Stratonovich correction term coincides with the mean curvature term of a stratum $\Sigma_{I_x} := \left\{ x \in \mathbb{R}^d \mid J(x) = 0 \right\}$ and its representation is given by

$$\frac{1}{2}\nabla\left(\sigma(t)\Pi\right)\left(\sigma(t)\Pi\right) = \frac{\sigma(t)^2}{2}\nabla(\Pi)\Pi = \frac{\sigma(t)^2}{2}\sum_{k=1}^{d}\nabla(\Pi_k)\Pi_k.$$

From the same tensor-calculus technique of Equation 3.46 in Rousset et al. [36], we observe that $(\nabla\Pi)\Pi$ is given by

$$\nabla\Pi(x)\Pi(x) = -\nabla J(x)^T G^\dagger(x) \left[\mathsf{Tr}\left(\nabla^2 J_1(x)\Pi(x)\right), ..., \mathsf{Tr}\left(\nabla^2 J_{m+|I_x|}(x)\Pi(x)\right)\right]^T.$$

Therefore, this gives the following Itô version of the closed-form SDE:

$$dX_t = -\left[\frac{\sigma(t)^2}{2}\Pi(X_t)\nabla f(X_t) + \alpha\sigma(t)^2\nabla J(X_t)^T G^\dagger(X_t)J(X_t)\right]dt + \frac{\sigma(t)^2}{2}\mathcal{H}(X_t)dt$$
$$+ \sigma(t)\Pi(X_t)dW_t.$$

$\square$

**Theorem B.1** (Fokker-Planck equation [52, 25] and the generator [53] on Riemannian manifold). *Let $Z_t \in \Sigma$ be a stochastic process following the SDE:*

$$dZ_t = V_0 dt + \sum_{k=1}^{d} V_k \circ dB_t^k,$$

*where $V_0, V_k$ are smooth vector fields on $\Sigma$ for each $k \in [d]$ and $B_t^k$ are kth components of Brownian motion $B_t$. Then, the law $\rho_t$ of the stochastic process $Z_t$ satisfies the following Fokker-Planck equation:*

$$\partial_t \rho_t = -\mathsf{div}_\Sigma(\rho_t V_0) + \frac{1}{2}\sum_{k=1}^{d}\mathsf{div}_\Sigma(\mathsf{div}_\Sigma(\rho_t V_k)V_k).$$

*Also, the generator of $\mathcal{L}$ of the corresponding SDE is provided as*

$$\mathcal{L}\phi = V_0\phi + \frac{1}{2}\sum_{k=1}^{d}V_k(V_k\phi)$$

*for any smooth function $\phi$ on $\Sigma$.*

**Lemma B.1** (Boundary condition of OLLA). *Assuming $X_0 \in \Sigma$, OLLA (3) satisfies the following boundary condition on $\partial\Sigma$ and property for $t \geq 0$:*

$$(1)\ \langle J_t(x), n(x)\rangle = 0 \quad a.e.\ on\ \partial\Sigma, \qquad (2)\ X_t \in \Sigma \quad a.s$$

*where $J_t(x)$ is the probability current density defined by $\partial_t \rho_t = -\mathsf{div}_\Sigma(J_t)$ and $n(x)$ is the outward unit normal vector on $\partial\Sigma$.*

*Proof.* First, we show that $\mathbb{P}(g_k(X_t) \leq 0) = 1$ for $t \geq 0$ and $k \in [l]$. To show this, we define a convex smooth violation penalty function $\Psi_\delta^k(x) : \mathbb{R}^d \to \mathbb{R}$ as follows:

$$\Psi_\delta^k(x) := \phi_\delta(g_k(x)), \qquad \phi_\delta(r) := \begin{cases} \frac{r^2}{2\delta}, & 0 \leq r \leq \delta \\ r - \delta/2, & r \geq \delta \\ 0 & r < 0. \end{cases}$$

Then, $\phi_\delta$ is convex, $C^1$, and satisfies

$$\phi_\delta \downarrow (r)_+, \quad \phi_\delta'(r) \to \mathbb{1}_{\{r>0\}}, \quad \text{as } \delta \downarrow 0$$

with $(r)^+ := \max\{r, 0\}$. Now, we observe that, on $\{g_k \geq 0\}$, the Stratonovich chain rule (as in Lemma B.4) gives

$$dg_k(X_t) = -\alpha\sigma(t)^2\left(g_k(X_t) + \epsilon\right)dt.$$

Therefore, applying Itô's lemma on $\phi_\delta(g_i(X_t))$ gives

$$d\phi_\delta(g_k(X_t)) = \phi_\delta'(g_k(X_t))dg_k(X_t) + \frac{1}{2}\phi_\delta''(g_k(X_t))\underbrace{d\langle g_k(X_t), g_k(X_t)\rangle_t}_{\text{Quadratic variation}=0}$$

$$= -\alpha\sigma(t)^2(g_k(X_t) + \epsilon)\phi_\delta'(g_k(X_t))dt.$$

For the case $\{g_k < 0\}$, it trivially holds that $\phi_\delta(g_k(X_t)) = 0$ with $\phi_\delta'(g_k(X_t)) = 0$. Therefore, the above observations lead to the following relation for $t \geq 0$:

$$\frac{d}{dt}\mathbb{E}[\Psi_\delta^k(X_t)] = \frac{d}{dt}\mathbb{E}[\phi_\delta(g_k(X_t))] = -\alpha\sigma(t)^2\mathbb{E}\left[(g_k(X_t) + \epsilon)\phi_\delta'(g_k(X_t))\right].$$

At this moment, we note that the non-decreasing property of $\phi_\delta'(r)$ implies, for $\forall r \geq 0$,

$$\phi_\delta(r) = \int_0^r \phi_\delta'(s)ds \leq \int_0^r \phi_\delta'(r)ds \leq (r + \epsilon)\phi_\delta'(r) \quad \Rightarrow \quad \Psi_\delta^k(x) \leq (g_k(x) + \epsilon)\phi_\delta'(g_k(x))$$

where the inequality $\phi_\delta(r) \leq (r + \epsilon)\phi_\delta'(r)$ also holds trivially for $r < 0$. Hence, we finally have

$$\frac{d}{dt}\mathbb{E}[\Psi_\delta^k(X_t)] = -\alpha\sigma(t)^2\mathbb{E}\left[(g_k(X_t) + \epsilon)\phi_\delta'(g_k(X_t))\right] \leq -\alpha\sigma(t)^2\mathbb{E}[\Psi_\delta(X_t)]$$

and the Grönwall inequality gives

$$0 \leq \mathbb{E}[\Psi_\delta^k(X_t)] \leq e^{-\alpha\int_0^t \sigma(s)^2 ds}\mathbb{E}[\Psi_\delta^k(X_0)] = 0 \quad (\because X_0 \in \Sigma)$$

which leads to $(g_k(X_t))_+ = 0 \Rightarrow g_k(X_t) \leq 0$ for $k \in [l]$ by letting $\delta \downarrow 0$ and applying the monotone convergence theorem. This proves $\mathbb{P}(g(X_t) \leq 0) = 1$ for $t \geq 0$ and $X_t \in \Sigma$ a.s.

Next, we prove $\langle J_t(x), n(x)\rangle = 0$ for $x \in \partial\Sigma$. We first observe that Theorem B.1 gives the following Fokker-Planck equation for $g(x) > 0$:

$$\partial_T\rho_t = -\mathsf{div}_\Sigma\left(\rho_t\left[-\frac{\sigma^2}{2}\nabla_\Sigma f - \alpha\sigma^2\nabla J^T G^\dagger J\right]\right) + \frac{\sigma^2}{2}\sum_{k=1}^d \mathsf{div}_\Sigma(\mathsf{div}_\Sigma(\rho_t f_k)f_k)$$

$$= -\mathsf{div}_\Sigma\left(\rho_t\left[-\frac{\sigma^2}{2}\nabla_\Sigma f - \alpha\sigma^2\nabla J^T G^\dagger J\right] - \frac{\sigma^2}{2}\nabla_\Sigma\rho_t\right),$$

where $f_k := \Pi e_k$ and $e_k$ being the $k$-th standard basis of $\mathbb{R}^d$. Also, we remark that the last equality holds using the property:

$$\sum_{k=1}^d \mathsf{div}_\Sigma(\rho_t f_k)f_k = \sum_{k=1}^d \langle\nabla_\Sigma\rho_t, f_k\rangle f_k + \rho_t\underbrace{\sum_{k=1}^d \mathsf{div}_\Sigma(f_k)f_k}_{=0} = \nabla_\Sigma\rho_t.$$

Therefore, the probability current density $J_t$ is given as follows

$$J_t = -\rho_t\left[\frac{\sigma^2}{2}\nabla_\Sigma f + \alpha\sigma^2\nabla J^T G^\dagger J\right] - \frac{\sigma^2}{2}\nabla_\Sigma\rho_t,$$

and we have

$$0 = \frac{d}{dt}\int_\Sigma \rho_t(x)d\sigma_\Sigma = -\int_\Sigma \mathsf{div}_\Sigma(J_t(x))d\sigma_\Sigma = -\int_{\partial\Sigma} \langle J_t(x), n(x)\rangle d\sigma_{\partial\Sigma}$$

$$= \int_{\partial\Sigma} \alpha\rho_t\sigma^2\underbrace{\langle\nabla J^T G^\dagger J, n\rangle}_{>0} d\sigma_{\partial\Sigma} \geq 0.$$

This implies $\rho_t = 0$ a.e on $\partial\Sigma$ and the following boundary condition holds almost everywhere on $\partial\Sigma$

$$\langle J_t, n\rangle = \langle\rho_t\left[-\frac{\sigma^2}{2}\nabla_\Sigma f - \alpha\sigma^2\nabla J^T G^\dagger J\right] - \frac{\sigma^2}{2}\nabla_\Sigma\rho_t, n\rangle = -\alpha\sigma^2\rho_t\langle\nabla J^T G^\dagger J, n\rangle = 0.$$

$\square$

**Lemma B.2** (Huang et al. [25]). *Let $\{f_k\}_{k=1}^d$ be a set of vectors defined by $f_k = \Pi(x)e_k$, where $\Pi(x)$ is the orthogonal projector onto $T_x\Sigma$ and $e_k$ is the kth standard basis vector of $\mathbb{R}^d$. Then, it holds that*

$$\sum_{k=1}^d (\mathsf{div}_\Sigma f_k) f_k = 0.$$

*Proof.* Let $r$ be the rank of the $\nabla J(x)$ and $\{n_1(x), \ldots, n_r(x)\}$ be an orthonormal basis of $\mathrm{Im}(\nabla J(x)^T)$. Since $\Pi(x)$ is the orthogonal projector onto the tangent space, it can be written using the projector onto the normal space as:

$$\Pi(x) = I - \sum_{l=1}^r n_l(x) n_l(x)^T.$$

Note that by definition, $n_l(x) \in \mathrm{Im}(\nabla J(x)^T)$ implies $n_l(x)^T \Pi(x) = 0$ for all $l \in [r]$.

Next, we define a vector field $F(x)$ by $F(x) = \Pi(x)\mathsf{div}_\Sigma(\Pi(x))$ where $(\mathsf{div}_\Sigma \Pi(x))_k := \mathsf{div}_\Sigma(f_k(x))$ for the vector field $f_k(x) = \Pi(x)^T e_k = \Pi(x)e_k$. With this definition, we have $\mathsf{div}_\Sigma \Pi = -\sum_{l=1}^r \mathsf{div}_\Sigma(n_l n_l^T)$ and observe that for any component index $k \in [d]$,

$$(\mathsf{div}_\Sigma(n_l n_l^T))_k = \mathsf{Tr}\left(\Pi\nabla(n_l n_l^T e_k)\right) = \sum_{i,j=1}^d \Pi_{ij}\partial_i(n_{lj}n_{lk}) = \sum_{i,j=1}^d \left[\Pi_{ij}(\partial_i n_{lj})n_{lk} + \Pi_{ij}n_{lj}(\partial_i n_{lk})\right]$$

$$= (\mathsf{div}_\Sigma n_l)n_{lk} + \sum_{i=1}^d \underbrace{(n_l^T \Pi)_i}_{=0} \partial_i n_{lk} = (\mathsf{div}_\Sigma n_l)n_{lk},$$

where we used the property that $n_l$ is orthogonal to the tangent space ($n_l^T \Pi = 0$). From this fact, we have the following result:

$$\mathsf{div}_\Sigma \Pi = -\sum_{l=1}^r \mathsf{div}_\Sigma(n_l n_l^T) = -\sum_{l=1}^r (\mathsf{div}_\Sigma n_l)n_l \;\Rightarrow\; F = \Pi\mathsf{div}_\Sigma(\Pi) = -\sum_{l=1}^r (\mathsf{div}_\Sigma n_l)\underbrace{\Pi n_l}_{=0} = 0.$$

Finally, the definition of $F$ gives $\sum_{k=1}^d (\mathsf{div}_\Sigma f_k) f_k = F$, which is zero by the argument above. $\square$

**Theorem B.2** (Stationarity of OLLA). *Assume $\sigma(t)$ is constant for $\forall t \geq 0$ and $X_0 \in \Sigma$. Then, OLLA (3) has the following stationary distribution $\rho_\Sigma$ with respect to measure $d\sigma_\Sigma$:*

$$\rho_\Sigma(x) = \frac{1}{Z_\Sigma} e^{-f(x)}, \quad x \in \Sigma$$

*where $d\sigma_\Sigma$ is the surface (or Hausdorff) measure on $\Sigma$ and $Z_\Sigma := \int_\Sigma e^{-f(x)} d\sigma_\Sigma$ is the normalization constant.*

*Proof.* To prove stationarity, we observe that

$$\int_\Sigma \mathcal{L}\phi\rho_t d\sigma_\Sigma = \frac{d}{dt}\int_\Sigma \phi\rho_t d\sigma_\Sigma = \int_\Sigma \phi\partial_t\rho_t d\sigma_\Sigma = -\int_\Sigma \phi\mathsf{div}_\Sigma(J_t)d\sigma_\Sigma = \int_\Sigma \langle J_t, \nabla_\Sigma\phi\rangle d\sigma_\Sigma$$

where the last equality comes from the boundary condition in Lemma B.1. Since $J_t$ is given by

$$J_t = -\frac{\sigma^2}{2}\left(\rho_t \nabla_\Sigma f + \nabla_\Sigma \rho_t\right) = -\frac{\sigma^2}{2} e^{-f}\nabla_\Sigma(\rho e^f),$$

on the interior of $\Sigma$, we conclude that

$$\int_\Sigma \mathcal{L}\phi\rho_\Sigma d\sigma_\Sigma = \int_\Sigma \langle J_t, \nabla_\Sigma\phi\rangle d\sigma_\Sigma = -\frac{\sigma^2}{2}\int_\Sigma \langle e^{-f}0, \nabla_\Sigma\phi\rangle d\sigma_\Sigma = 0,$$

where $J_t = 0$ due to the fact that $\rho_\Sigma \propto e^{-f}$. This proves that $\rho_\Sigma$ is the stationary distribution of the OLLA. $\square$

## B.2 CONSTRUCTION OF ULLA

**Notations and Background for underdamped setup.** In the constrained underdamped Langevin case, assuming $X_0 \in \Sigma$, the natural space of $(X_t, P_t)$ is the cotangent bundle $T^*\Sigma := \left\{ (x,p) \in \mathbb{R}^d \times \mathbb{R}^d \mid x \in \Sigma, \langle \nabla J(x), p \rangle = 0 \right\}$, where $\Sigma := \left\{ x \in \mathbb{R}^d \mid h(x) = 0, g(x) \leq 0 \right\}$ is a stratified manifold. Because there is no boundary for the cotangent space $T_x^*\Sigma := \left\{ p \in \mathbb{R}^d \mid \langle \nabla J(x), p \rangle = 0 \right\}$, the boundary of $T^*\Sigma$ is given as $\partial T^*\Sigma = \partial \Sigma \times T_x^*\Sigma$. In this cotangent bundle $T^*\Sigma$, the canonical reference measure is the Liouville (symplectic) measure $\sigma_{T^*\Sigma}$ defined as $d\sigma_{T^*\Sigma}(x, p) := d\sigma_\Sigma(x) \otimes dp(x)$ with $d\sigma_\Sigma$ the surface measure on $\Sigma$ and $dp(x)$ the Lebesgue measure on the cotangent space $T_x^*\Sigma$ induced by the inner product $\langle u, v \rangle = u^T v$.

For the notations, we write $\nabla_\Sigma$ and $\mathsf{div}_\Sigma$ for the intrinsic gradient and divergence in $x$, and $\nabla_{T_x^*\Sigma}, \mathsf{div}_{T_x^*\Sigma}$ for the intrinsic gradient and divergence in $p$. Under these notations, for a smooth function $\phi$ and smooth vector field $X$ on $\Sigma$, we have

$$\nabla_\Sigma \phi = \Pi \nabla_x \phi, \qquad \mathsf{div}_\Sigma(X) = \mathsf{Tr}\left( \Pi \nabla_x X \right)$$

Similarly, for a smooth function $\psi$ and smooth vector $Y$ on $T_x^*\Sigma$, we have

$$\nabla_{T_x^*\Sigma} \psi = \Pi \nabla_p \psi, \qquad \mathsf{div}_{T_x^*\Sigma}(Y) = \mathsf{Tr}\left( \Pi \nabla_p Y \right),$$

where $\nabla_x$ and $\nabla_p$ represent ambient Euclidean partial gradient or Jacobian operators with respect to $x$ or $p$. Also, the global gradient with respect to $\sigma_{T^*\Sigma}$ is given by $\nabla_{T^*\Sigma}\phi = [\nabla_\Sigma \phi, \nabla_{T_x^*\Sigma}\phi]^T$ for any smooth function $\phi$ on $T^*\Sigma$ and the global divergence with respect to $\sigma_{T^*\Sigma}$ can be represented by $\mathsf{div}_{T^*\Sigma}([V^x, V^p]^T) = \mathsf{div}_\Sigma(V^x) + \mathsf{div}_{T_x^*\Sigma}(V^p)$ for any smooth tangent field $[V^x, V^p]^T \in T^*\Sigma$. For comprehensive backgrounds on constrained underdamped Langevin dynamics, see Chapter 3.3 in Rousset et al. [36].

**Proposition B.2** (Construction of ULLA). *Consider the following Lagrangian-form constrained underdamped Langevin dynamics:*

$$\begin{cases} dX_t = \sigma(t)^2 P_t dt + \nabla J(X_t)^T d\lambda_t \\ dP_t = -\sigma(t)^2 \nabla f(X_t) dt - \sigma(t)^2 \gamma P_t dt + \sigma(t)\sqrt{2\gamma} \circ dW_t + \nabla J(X_t)^T d\mu_t, \end{cases} \tag{4}$$

*where $d\lambda_t, d\mu_t$ are the adapted processes such that $dJ(X_t) = -\alpha\sigma(t)^2 J(X_t)$ (position constraint) and $\nabla J(X_t) P_t = 0$ (momentum tangency constraint), respectively.*

*Assuming $\nabla J(X_0) P_0 = 0$, the explicit minimum norm solution of $d\lambda_t, d\mu_t$ are given by*

$$\begin{cases} d\lambda_t = -\alpha\sigma(t)^2 G^\dagger(X_t) J(X_t) dt \\ d\mu_t = G^\dagger(X_t) \nabla J(X_t) \left[ \sigma(t)^2 \nabla f(X_t) + \sigma(t)^2 \gamma P_t dt + \sigma(t)\sqrt{2\gamma} \circ dW_t \right] + \\ \qquad G^\dagger(X_t) \left[ -\sigma(t)^2 \mathcal{H}_1(X_t, P_t) + \alpha\sigma(t)^2 \mathcal{H}_2(X_t, P_t) \right] dt, \end{cases}$$

*where $G(x) := \nabla J(x) \nabla J(x)^T$ is the Gram matrix and $\Pi(x) = I - \nabla J(x)^T G^\dagger(x) \nabla J(x)$ is the tangential projection map.*

*Therefore, the closed form SDE of (4) is given as follows:*

$$\begin{cases} dX_t = \sigma(t)^2 P_t dt - \alpha\sigma(t)^2 \nabla J(X_t) G^\dagger(X_t) J(X_t) dt \\ dP_t = \Pi(X_t) \left[ -\sigma(t)^2 \nabla f(X_t) - \sigma(t)^2 \gamma P_t dt + \sigma(t)\sqrt{2\gamma} \circ dW_t \right] \\ \qquad - \sigma(t)^2 \nabla J(X_t)^T G^\dagger(X_t) \left[ \mathcal{H}_1(X_t, P_t) - \alpha\mathcal{H}_2(X_t, P_t) \right] dt \end{cases}$$

*where $\mathcal{H}_1 \in \mathbb{R}^{m+|I_x|}, \mathcal{H}_2 \in \mathbb{R}^{(m+|I_x|) \times (m+|I_x|)}$ are the curvature correction terms defined as*

$$[\mathcal{H}_1(x, p)]_i := p^T \nabla^2 J_i(x) p,$$

$$[\mathcal{H}_2(x, p)]_i := p^T \nabla^2 J_i(x) (\nabla J(x)^T G^\dagger(x) J(x))$$

*with $[\mathcal{H}_1(x,p)]_i, [\mathcal{H}_2(x,p)]_i$ being the ith entry and column of $\mathcal{H}_1(x,p), \mathcal{H}_2(x,p)$ respectively.*

*Proof.* From the the Stratonovich chain rule, we observe that

$$-\alpha\sigma(t)^2 J(X_t) dt = dJ(X_t) = \nabla J(X_t) \circ dX_t = \sigma(t)^2 \underbrace{\nabla J(X_t) P_t}_{=0} dt + G(X_t) d\lambda_t.$$

Because the initial condition gives $\nabla J(X_0)P_0 = 0$ and $d\mu_t$ imposes $d(\nabla J(X_t)P_t) = 0$, the first term becomes zero and the minimum norm solution of $d\lambda_t$ simplifies to

$$d\lambda_t = -\alpha\sigma(t)^2 G^\dagger(X_t)J(X_t)dt.$$

To find the explicit minimum norm solution for the process $d\mu_t$, we consider the momentum tangency constraint $\nabla J(X_t)^T P_t = 0$. Using the Stratonovich chain rule again, we have

$$0 = d(\nabla J(X_t)P_t) = P_t^T \nabla^2 J(X_t)dX_t + \nabla J(X_t)dP_t.$$

By substituting $dX_t$ and $dP_t$, the previous equation simplifies to

$$
\begin{aligned}
0 =& P_t^T \nabla^2 J(X_t) \left[\sigma(t)^2 P_t dt + \nabla J(X_t)^T d\lambda_t\right] \\
&+ \nabla J(X_t)\left[-\sigma(t)^2 \nabla f(X_t) - \sigma(t)^2 \gamma P_t dt + \sigma(t)\sqrt{2\gamma} \circ dW_t + \nabla J(X_t)^T d\mu_t\right] \\
=& \left[\sigma^2 \mathcal{H}_1 - \alpha\sigma^2 \mathcal{H}_2 - \sigma^2 \nabla J \nabla f - \sigma^2 \gamma \nabla J P_t\right]dt + \sigma \nabla J\sqrt{2\gamma} \circ dW_t + G d\mu_t.
\end{aligned}
$$

This gives the following minimum norm solution of $d\mu_t$:

$$d\mu_t = G^\dagger \left[\sigma^2 \nabla J \nabla f + \sigma^2 \gamma \nabla J P_t - \sigma^2 \mathcal{H}_1 + \alpha\sigma^2 \mathcal{H}_2\right]dt - \sigma G^\dagger \nabla J\sqrt{2\gamma} \circ dW_t.$$

Therefore, we recover the following $dX_t, dP_t$ by plugging the adapted process $d\lambda_t, d\mu_t$ into the previous equations:

$$
\begin{cases}
dX_t = \sigma(t)^2 P_t dt - \alpha\sigma(t)^2 \nabla J(X_t)G^\dagger(X_t)J(X_t)dt \\
dP_t = \Pi(X_t)\left[-\sigma(t)^2 \nabla f(X_t)dt - \sigma(t)^2 \gamma P_t dt + \sigma(t)\sqrt{2\gamma} \circ dW_t\right] \\
\qquad - \sigma(t)^2 \nabla J(X_t)^T G^\dagger(X_t)\left[\mathcal{H}_1(X_t, P_t) - \alpha\mathcal{H}_2(X_t, P_t)\right]dt.
\end{cases}
$$

We note that this is the unique closed form SDE because $\nabla J(X_t)^T d\lambda_t$ and $\nabla J(X_t)^T d\mu_t$ are unique among many solutions $d\lambda_t, d\mu_t$ satisfying the properties.

Finally, we observe that the Itô-Stratonovich correction term $\frac{1}{2}(\nabla B)B = 0$ where $B = [0, \sigma\sqrt{2\gamma}\Pi]^T \in \mathbb{R}^{2d\times d}$ is the diffusion matrix. This is because the position entries of $B$ are zero and the momentum entries of $B$ depend only on position. Therefore, we have the same formula on the Itô version of the above Stratonovich SDE. $\qquad\square$

**Lemma B.3** (Boundary condition of ULLA). *Assuming $X_0 \in \Sigma$ and $\nabla J(X_0)P_0 = 0$, ULLA (4) satisfies the following boundary condition on $\partial\Sigma$ and property for $t \geq 0$:*

$$(1) \ \langle J_t(x, p), n(x)\rangle = 0 \quad \text{a.e on } \partial T^*\Sigma, \qquad (2) \ (X_t, P_t) \in T^*\Sigma,$$

*where $J_t(x)$ is the probability current density defined by $\partial_t\rho_t = -\text{div}_\Sigma(J_t)$ and $n(x)$ is the outward unit normal vector on $\partial T^*\Sigma$.*

*Proof.* First, we prove $\mathbb{P}(g_k(X_t) > 0) = 0$ for $t \geq 0, k \in [l]$. In particular, this implies $(X_t, P_t) \in T^*\Sigma$ a.s. for all $t \geq 0$. To show this, we observe that Lemma B.4 gives

$$dg_k(X_t) = -\alpha\sigma(t)^2(g_k(X_t) + \epsilon)dt$$

for $\{g_k \geq 0\}$. Thus, the same proof introduced in Lemma B.1 gives $\mathbb{P}(g(X_t) \leq 0) = 1$ for $t \geq 0$ and, therefore, we have $(X_t, P_t) \in T^*\Sigma$ a.s for $t \geq 0$.

Next, we show the boundary condition of this SDE. To demonstrate this, we first formulate the SDE of ULLA in the following form:

$$dZ_t = V_0 dt + \sum_{k=1}^{2d} V_k \circ dB_t^k$$

with $Z_t := [X_t, P_t]^T \in \mathbb{R}^{2d}$, $V_0(x, p) := \left[\sigma^2 p - \alpha\sigma^2 \nabla J G^\dagger J, -\sigma^2 \Pi\nabla f - \sigma^2 \gamma\Pi p + \kappa\right]^T \in \mathbb{R}^{2d}$, and $V_k(x, p) := [0, Be_k] \in \mathbb{R}^{2d}$, where $\kappa := -\sigma^2 \nabla J^T G^\dagger\left[\mathcal{H}_1 - \alpha\mathcal{H}_2\right]$ is the curvature correction

related term and $B := \sigma\Pi\sqrt{2\gamma}$. Now, we use the Fokker-Planck equation on the interior of $T^*\Sigma$ (Theorem B.1) applied with $\text{div}_{T^*\Sigma}$, and this gives the following equation:

$$\partial_t \rho_t = -\text{div}_{T^*\Sigma}(\rho_t V_0) + \frac{1}{2}\sum_{k=1}^{2d}\text{div}_{T^*\Sigma}\left(\text{div}_{T^*\Sigma}(\rho_t V_k)V_k\right),$$

where

$$-\text{div}_{T^*\Sigma}(\rho_t V_0) = -\text{div}_\Sigma(\rho_t\left(\sigma^2 p - \alpha\sigma^2\nabla JG^\dagger J\right)) - \text{div}_{T_x^*\Sigma}\left(\rho_t\left(-\sigma^2\Pi\nabla f - \sigma^2\gamma\Pi p + \kappa\right)\right)$$

and

$$\frac{1}{2}\sum_{k=1}^{2d}\text{div}_{T^*\Sigma}\left(\text{div}_{T^*\Sigma}(\rho_t V_k)V_k\right) = \frac{1}{2}\sum_{k=1}^{d}\text{div}_{T_x^*\Sigma}\left(\langle Be_k, \nabla_{T_x^*\Sigma}\rho_t\rangle Be_k\right) = \frac{1}{2}\text{div}_{T_x^*\Sigma}\left((BB^T)\nabla_{T_x^*\Sigma}\rho_t\right).$$

Therefore, we recover the following equation:

$$\partial_t\rho_t = -\text{div}_\Sigma(\underbrace{\rho_t\left(\sigma^2 p - \alpha\sigma^2\nabla JG^\dagger J\right)}_{:=J_t^x(x,p)})$$

$$-\text{div}_{T_x^*\Sigma}\left(\underbrace{\rho_t\left(-\sigma^2\Pi\nabla f - \sigma^2\gamma\Pi p + \kappa - \sigma^2\gamma\Pi\Pi^T\nabla_{T_x^*\Sigma}\rho_t\right)}_{:=J_t^p(x,p)}\right)$$

$$= -\text{div}_{T^*\Sigma}\left([J_t^x, J_t^p]^T\right) = -\text{div}_{T^*\Sigma}(J_t)$$

Finally, we observe that, on the boundary $\partial T^*\Sigma = \partial\Sigma \times T_x^*\Sigma$, the outward normal is given by $n = [n_x, 0]^T \in \mathbb{R}^{2d}$ with $n_x$ being the outward unit normal vector on $\partial\Sigma$. Therefore, we have

$$0 = \frac{d}{dt}\int_{T^*\Sigma}\rho_t(x,p)d\sigma_{T^*\Sigma} = -\int_{T^*\Sigma}\text{div}_{T^*\Sigma}(J_t(x,p))d\sigma_{T^*\Sigma} = -\int_{\partial T^*\Sigma}\langle J_t(x,p), n\rangle d\sigma_{\partial T^*\Sigma}$$

$$= -\int_{\partial T^*\Sigma}\langle J_t^x(x,p), n_x\rangle d\sigma_{\partial T^*\Sigma} = -\int_{\partial T^*\Sigma}\langle\rho_t\left(\sigma^2 p - \alpha\sigma^2\nabla JG^\dagger J\right), n_x\rangle d\sigma_{\partial T^*\Sigma}$$

$$= \alpha\sigma^2\int_{\partial T^*\Sigma}\rho_t\underbrace{\langle\nabla JG^\dagger J, n_x\rangle}_{>0}d\sigma_{\partial T^*\Sigma}$$

and it implies $\rho_t(x,p) = 0$ a.e. on $\partial T^*\Sigma$. Lastly, we conclude the proof by observing that the following holds a.e on $\partial T^*\Sigma$:

$$\langle J_t, n\rangle = \langle J_t^x, n_x\rangle = \langle\rho_t\left(\sigma^2 p - \alpha\sigma^2\nabla JG^\dagger J\right), n_x\rangle = -\alpha\sigma^2\rho_t\langle\nabla J^T G^\dagger J, n_x\rangle = 0.$$

$\square$

**Theorem B.3** (Stationarity of ULLA). *Assume $\sigma(t)$ is constant for $\forall t \geq 0$, $X_0 \in \Sigma$, and the tangency constraint $\nabla J(X_0)P_0 = 0$ holds. Then, the ULLA (4) has the following stationary distribution $\rho_{T^*\Sigma}$ with respect to the measure $d\sigma_{T^*\Sigma}$:*

$$\rho_{T^*\Sigma}(x,p) = \frac{1}{Z_{T^*\Sigma}}\exp\left(-f(x) - \frac{1}{2}\|p\|_2^2\right), \quad (x,p) \in T^*\Sigma$$

*where $d\sigma_{T^*\Sigma}$ is the Liouville measure on $T^*\Sigma$ and $Z_{T^*\Sigma} := \int_{T^*\Sigma}e^{\left(-f(x)-\frac{1}{2}\|p\|_2^2\right)}d\sigma_{T^*\Sigma}$ is the normalization constant.*

*Proof.* First, from Lemma B.3, we know that the SDE of ULLA can be rewritten as follows on the interior of $T^*\Sigma$ :

$$dZ_t = V_0 dt + \sum_{k=1}^{2d}V_k \circ dB_t^k$$

with $Z_t := [X_t, P_t]^T \in \mathbb{R}^{2d}$, $V_0(x,p) := \left[\sigma^2 p, -\sigma^2\Pi\nabla f - \sigma^2\gamma\Pi p + \kappa\right]^T \in \mathbb{R}^{2d}$, and $V_k(x,p) := [0, Be_k] \in \mathbb{R}^{2d}$, where $\kappa := -\sigma^2\nabla J^T G^\dagger\mathcal{H}_1$ is the curvature correction related term and $B :=$

$\sigma\Pi\sqrt{2\gamma}$. Therefore, Theorem B.1 gives the following generator $\mathcal{L}$ for any smooth function $\phi$ on $T^*\Sigma$:

$$\mathcal{L}\phi = V_0\phi + \frac{1}{2}\sum_{k=1}^{2d} V_k(V_k\phi).$$

Because we have

$$V_0\phi = \langle\nabla_\Sigma\phi, \sigma^2 p\rangle + \langle\nabla_{T_x^*\Sigma}\phi, -\sigma^2\Pi\nabla f - \sigma^2\gamma\Pi p + \kappa\rangle$$

and

$$\frac{1}{2}\sum_{k=1}^{2d} V_k(V_k\phi) = \frac{1}{2}\sum_{k=1}^{d}\langle Be_k, \nabla_p(\langle Be_k, \nabla_p\phi\rangle)\rangle = \frac{1}{2}\sum_{k=1}^{d}\langle\nabla_p^2\phi Be_k, Be_k\rangle = \sigma^2\gamma\mathsf{Tr}\left(\Pi\nabla_p^2\phi\right)$$

on the interior of $T^*\Sigma$, we can simplify $\mathcal{L}\phi$ as follows:

$$\mathcal{L}\phi = \sigma^2\left[\underbrace{\langle\nabla_\Sigma\phi, p\rangle - \langle\Pi\nabla f, \nabla_{T_x^*\Sigma}\phi\rangle + \langle\kappa, \nabla_{T_x^*\Sigma}\phi\rangle}_{\mathcal{L}_H\phi} + \underbrace{\gamma\left(\Delta_{T_x^*\Sigma}\phi - \langle\nabla_{T_x^*\Sigma}\phi, p\rangle\right)}_{\mathcal{L}_{OU}\phi}\right]$$

where we used $\Pi p = p$ (tangency constraint) and $\Delta_{T_x^*\Sigma}\phi = \mathsf{div}_{T_x^*\Sigma}(\Pi\nabla_p\phi) = \mathsf{Tr}\left(\Pi\nabla_p^2\phi\right)$. Next, we note the following identity:

$$\mathsf{div}_{T_x^*\Sigma}\left(e^{-\|p\|^2/2}\nabla_{T_x^*\Sigma}\phi\right) = e^{-\|p\|^2/2}\left(\Delta_{T_x^*\Sigma}\phi - \langle\nabla_{T_x^*\Sigma}\phi, p\rangle\right).$$

Under this identity, we observe that

$$\int_{T^*\Sigma}\mathcal{L}_{OU}\phi\rho_{T^*\Sigma}d\sigma_{T^*\Sigma} = \gamma\int_{T^*\Sigma}\left(\Delta_{T_x^*\Sigma}\phi - \langle\nabla_{T_x^*\Sigma}\phi, p\rangle\right)e^{-H}d\sigma_{T^*\Sigma} = 0$$

where $H := f(x) + \frac{1}{2}\|p\|^2$ and the last equality holds because $T_x^*\Sigma$ does not have boundary and $d\sigma_{T^*\Sigma}(x, p) := d\sigma_\Sigma(x) \otimes dp(x)$ holds. Lastly, we define $X_H := \left[p - \alpha\nabla JG^\dagger J, -\Pi\nabla f + \kappa\right]^T \in \mathbb{R}^{2d}$ so that $X_H = [p, -\Pi\nabla f + \kappa]^T$ and $\mathcal{L}_H\phi = \langle X_H, \nabla_{T^*\Sigma}\phi\rangle$ at the interior of $T^*\Sigma$. Then, we have

$$\langle X_H, \nabla_{T^*\Sigma}H\rangle = \langle p, \nabla_\Sigma f\rangle + (-\Pi\nabla f + \kappa)\Pi p = \underbrace{\langle(\nabla_\Sigma f - \Pi\nabla f), \Pi p\rangle}_{=0} + \underbrace{\langle\kappa, \Pi p\rangle}_{=0} = 0,$$

where the last equality holds due to tangency constraint of $p$. In addition to this, the following identity holds on the interior of $T^*\Sigma$ due to Liouville's theorem, which is the preservation property of Liouville measure on $T^*\Sigma$ under the constrained Hamiltonian field (see Chapter 1.2.2 and Proposition 3.46 in Rousset et al. [36]):

$$\mathsf{div}_{T^*\Sigma}X_H = \mathsf{div}_\Sigma(p) + \mathsf{div}_{T_x^*\Sigma}(-\Pi\nabla f + \kappa) = 0.$$

Hence, using these properties and $\rho_{T^*\Sigma} \propto e^{-H}$, we have

$$\int_{T^*\Sigma}\mathcal{L}_{OU}\phi\rho_{T^*\Sigma}d\sigma_{T^*\Sigma} = \int_{T^*\Sigma}\langle X_H, \nabla_{T^*\Sigma}\phi\rangle\rho_{T^*\Sigma}d\sigma_{T^*\Sigma} \overset{(1)}{=} -\int\phi\mathsf{div}_{T^*\Sigma}\left(\rho_{T^*\Sigma}X_H\right)d\sigma_{T^*\Sigma}$$

$$= -\int_{T^*\Sigma}\phi\left(\langle X_H, \nabla_{T^*\Sigma}\rho_{T^*\Sigma}\rangle + \underbrace{\mathsf{div}_{T^*\Sigma}(X_H)}_{=0}\rho_{T^*\Sigma}\right)d\sigma_{T^*\Sigma}$$

$$\overset{(2)}{=} \int_{T^*\Sigma}\phi\rho_{T^*\Sigma}\langle X_H, \nabla_{T^*\Sigma}H\rangle d\sigma_{T^*\Sigma} = 0,$$

where (1) comes from the boundary condition in Lemma B.3 and (2) comes from the fact that $-\nabla_{T^*\Sigma}H = \nabla_{T^*\Sigma}\ln\rho_{T^*\Sigma} = \nabla_{T^*\Sigma}\rho_{T^*\Sigma}/\rho_{T^*\Sigma}$.

By combining the observations above, we have

$$\int_{T^*\Sigma}\mathcal{L}\phi\rho_{T^*\Sigma}d\sigma_{T^*\Sigma} = \int_{T^*\Sigma}\mathcal{L}_{OU}\phi\rho_{T^*\Sigma}d\sigma_{T^*\Sigma} + \int_{T^*\Sigma}\mathcal{L}_H\phi\rho_{T^*\Sigma}d\sigma_{T^*\Sigma} = 0,$$

which proves the theorem. $\qquad\square$

B.3 PROPERTIES AND BACKWARD PROCESSES OF CONSTRAINED LANGEVIN DYNAMICS
WITH LANDING

**Exponential decaying properties of constrained Langevin dynamics with landing**  Due to our previous construction, the landing term $-\alpha \nabla J(X_t)G^\dagger(X_t)J(X_t)$ appears on both constrained overdamped or underdamped Langevin dynamics so that the processes always satisfy $dJ(X_t) = -\alpha\sigma(t)^2 J(X_t)$ deterministically $\forall t \geq 0$. Therefore, even if $X_t \notin \Sigma$, the process can approach $\Sigma$ exponentially fast as illustrated in the following Lemma B.4.

**Lemma B.4** (Exponential decay of constraint functions). *Let $J(x)$ be the constraint function vector defined as*

$$J(x) = \left[h_1(x), ..., h_m(x), g_{i_1}(x) + \epsilon, ..., g_{i_{|I_x|}}(x) + \epsilon\right]^T \in \mathbb{R}^{m+|I_x|} \tag{5}$$

*where $h : \mathbb{R}^d \to \mathbb{R}^m, g : \mathbb{R}^d \to \mathbb{R}^l$ are equality and inequality constraint functions respectively, and $I_x := \{i \in [l] : g(x) \geq 0\}$ is the active index set of inequality constraints.*

*Under this setup, the constrained Langevin dynamics (3 or 4) satisfy the following constraint satisfaction property almost surely:*

$$h_i(X_t) = h_i(X_0)e^{-\alpha S(t)}, \quad t \geq 0$$

*and*

$$\begin{cases} g_j(X_t) = -\epsilon + (g_j(X_0) + \epsilon)e^{-\alpha S(t)}, & t \leq \tau_{j,\epsilon} \\ g_j(X_t) \leq 0, & t \geq \tau_{j,\epsilon}, \end{cases}$$

*where $S(t) := \int_0^t \sigma(s)^2 ds$ and $\tau_{j,\epsilon}$ is defined to be*

$$\tau_{j,\epsilon} := \inf\left\{t \geq 0 \mid S(t) \geq \frac{1}{\alpha}\ln\left(\frac{g_j(X_0) + \epsilon}{\epsilon}\right)\right\}, \quad \forall j \in I_{x_0}.$$

*Proof.* From the Stratonovich chain rule, it holds almost surely that

$$dJ(X_t) = \nabla J(X_t) \circ dX_t = -\alpha\sigma(t)^2 J(X_t)dt.$$

For each equality constraint $h_i$, the component is active for $\forall t \geq 0$. Therefore, we have:

$$dh_i(X_t) = -\alpha\sigma(t)^2 h_i(X_t)dt$$

and solving this ODE yields:

$$h_i(X_t) = h_i(X_0)e^{-\alpha S(t)}, \quad t \geq 0.$$

For the inequality constraints, we fix $j \in I_{X_0} := \{j \in [l] \mid g(X_0) \geq 0\}$. While the $j$-th inequality is active, we have $J_{m+j} = g_j + \epsilon$ in the constraint vector, and the same chain rule gives

$$d(g_j(X_t) + \epsilon) = -\alpha\sigma(t)^2 (g_j(X_t) + \epsilon)\, dt.$$

Hence, before time $t \leq \tau_{j,\epsilon} := \inf\left\{t \geq 0 \mid S(t) \geq \frac{1}{\alpha}\ln\left(\frac{g_j(X_0)+\epsilon}{\epsilon}\right)\right\}$, we have

$$g_j(X_t) = -\epsilon + (g_j(X_0) + \epsilon)\, e^{-\alpha s(t)}.$$

After $t \geq \tau_{j,\epsilon}$, the particle $X_t$ is instantaneously repelled into the interior of $\Sigma$ whenever it hits the boundary $\partial\Sigma$. Therefore, $g_j(X_t) \leq 0$ holds for $t \geq \tau_{j,\epsilon}$. □

**Backward process on manifold**  Now, we discuss how to construct the backward process of the proposed constrained Langevin dynamics. Define $\mathcal{M} \subset \mathbb{R}^D$ to be a smooth, compact, embedded Riemannian manifold endowed with an induced metric, and we restrict the choice of $\mathcal{M}$ to be either $\mathcal{M} = \Sigma$ $(D = d)$ or $\mathcal{M} = T^*\Sigma$ $(D = 2d)$. For $x \in \mathcal{M}$, let $\Pi$ be the tangential projection map on $\mathcal{M}$. We consider the stochastic process $X_t \sim q_t$ driven by the following Stratonovich SDE on $\mathcal{M}$ with $X_0 \sim q_0 = q_{\text{data}}$:

$$dX_t = \Pi(X_t)b(X_t, t)dt + \kappa(t)\Pi(X_t) \circ dW_t, \quad t \in [0, T] \tag{6}$$

where $b(y, t) : \mathcal{M} \times \mathbb{R} \to \mathbb{R}^D$, $\kappa(t) : \mathbb{R} \to \mathbb{R}^{D \times D}$ are the drift vector and diagonal diffusion matrix for the corresponding SDE. In practice, $\kappa(t) : \mathbb{R} \to \mathbb{R}^{D \times D}$ is chosen to satisfy $X_T \sim q_T \approx p_{\text{prior}}$ with $p_{\text{prior}}$ being the easy-to-sample prior distribution on $\mathcal{M}$ and $T$ the terminal time for sampling.

Then, Lemma B.5 shows that the stochastic process $\overleftarrow{X}_t \sim p_t$ driven by the following Stratonovich SDE becomes the backward process of $X_t$:

$$d\overleftarrow{X}_t = \Pi(\overleftarrow{X}_t) \left[ -b(\overleftarrow{X}_t, T - t) + \kappa(T - t)^2 \nabla \ln q_{T-t}(\overleftarrow{X}_t) \right] dt + \kappa(T-t)\Pi(\overleftarrow{X}_t) \circ d\bar{W}_t, \quad t \in [0, T]. \tag{7}$$

It is assumed that the forward process (6) and the backward process (7) have the same boundary conditions, so that $q_t = p_{T-t}$ holds for $t \in [0, T]$.

**Lemma B.5** (Backward process verification). *Let $X_t, \overleftarrow{X}_t \in \mathcal{M}$ be the stochastic process driven by the forward process (6) and the backward process (7), respectively. If $q_T = p_0$ and the boundary conditions of $q_t$ and $p_{T-t}$ appearing in the Fokker-Planck equation are the same, then the following relation holds:*

$$q_t(x) = p_{T-t}(x), \quad x \in \mathcal{M}, \ t \in [0, T],$$

*where $q_t, p_t$ are the probability densities of each $X_t$ and $\overleftarrow{X}_t$.*

*Proof.* Let $\nabla_{\mathcal{M}} := \Pi\nabla$ and $\text{div}_{\mathcal{M}}$ be the intrinsic gradient and divergence on $\mathcal{M}$, and let $\Delta_{\mathcal{M}} := \text{div}_{\mathcal{M}}(\nabla_{\mathcal{M}} \cdot)$ be the intrinsic Laplacian operator on $\mathcal{M}$. From Theorem B.1 with $V_0(y, t) = \Pi(y)b(y, t), V_k(y, t) = \kappa(t)\Pi(x)e_k$, and $e_k$ being $k$th standard basis of $\mathbb{R}^d$, the Fokker-Planck equation of (6) is given by

$$\partial_t q_t = -\text{div}_{\mathcal{M}}(q_t V_0) + \frac{1}{2} \sum_{k=1}^{D} \text{div}_{\mathcal{M}}(\text{div}_{\mathcal{M}}(q_t V_k)V_k) = -\text{div}_{\mathcal{M}}(q_t \Pi b) + \frac{1}{2}\kappa(t)^2 \Delta_{\mathcal{M}} q_t,$$

where we used the property $\sum_{k=1}^{D} \text{div}_{\mathcal{M}}(\text{div}_{\mathcal{M}}(q_t \Pi e_k)(\Pi e_k)) = \Delta_{\mathcal{M}} p_t$.

Now observe that the process $\overleftarrow{X}_t$ driven by (7) has the following drift and diffusion term:

$$\tilde{V}_0(y, t) = \Pi(y) \left[ -b(y, T - t) + \kappa(T - t)^2 \nabla_{\mathcal{M}} \ln q_{T-t}(x) \right], \quad \tilde{V}_k(y, t) = g(T - t)\Pi(y)e_k.$$

Therefore, Theorem B.1 again implies the following Fokker-Planck equation:

$$\partial_t p_{T-t} = -\partial_s p_s \mid_{s=T-t} = \text{div}_{\mathcal{M}}(p_{T-t}\tilde{V}_0) - \frac{1}{2} \sum_{k=1}^{d} \text{div}_{\mathcal{M}}(\text{div}_{\mathcal{M}}(p_{T-t}\tilde{V}_k)\tilde{V}_k)$$

$$= -\text{div}_{\mathcal{M}}(p_{T-t}\Pi b) + \kappa(t)^2 \text{div}_{\mathcal{M}}(q_{T-t}\nabla_{\mathcal{M}} \ln p_{T-t}) - \frac{1}{2}\kappa(t)^2 \Delta_{\mathcal{M}} p_{T-t}$$

$$= -\text{div}_{\mathcal{M}}(p_{T-t}\Pi b) + \frac{1}{2}\kappa(t)^2 \Delta_{\mathcal{M}} p_{T-t}.$$

In the case of manifold with boundary, it is assumed that the boundary condition on the Fokker-Planck equation of $q_t$ and $p_{T-t}$ are the same and $q_T = p_0$. This implies that $\overleftarrow{X}_t$ achieves the desired property as stated in the theorem. $\qquad \square$

**Backward process of Constrained Langevin with landing** From Proposition B.1, we note that the forward process $X_t \sim q_t$ of OLLA is given as follows in the interior of $\Sigma$:

$$dX_t = -\frac{1}{2}\sigma(t)^2 \Pi(X_t)\nabla f(X_t)dt + \sigma(t)\Pi(X_t) \circ dW_t.$$

Therefore, applying Lemma B.5 on the interior of $\Sigma$, the backward process of the OLLA is given as:

$$d\overleftarrow{X}_t = \frac{1}{2}\sigma(T - t)^2 \Pi(\overleftarrow{X}_t) \left[ \nabla f(\overleftarrow{X}_t) + 2\nabla \ln q_{T-t}(\overleftarrow{X}_t) \right] dt + \sigma(t)\Pi(\overleftarrow{X}_t) \circ d\bar{W}_t.$$

Now, we note that the similar construction and proof provided in Proposition B.1 and Lemma B.1 can demonstrate that adding the landing term $-\alpha\sigma^2\nabla JG^\dagger J$ to the previous SDE enforces it to have the same boundary condition ($\langle J_t, n\rangle = 0$ a.e on $\partial\Sigma$) imposed on the forward process. Therefore, the backward process $\overleftarrow{X}_t$ is given as:

$$d\overleftarrow{X}_t = \frac{1}{2}\sigma(T-t)^2\Pi(\overleftarrow{X}_t)\left[\nabla f(\overleftarrow{X}_t) + 2\nabla\ln q_{T-t}(\overleftarrow{X}_t)\right]dt - \alpha\sigma(T-t)^2\nabla J(\overleftarrow{X}_t)^T G^\dagger(\overleftarrow{X}_t)J(\overleftarrow{X}_t)dt$$

$$+ \frac{\sigma(t)^2}{2}\mathcal{H}(\overleftarrow{X}_t)dt + \sigma(t)\Pi(\overleftarrow{X}_t)\circ d\bar{W}_t, \qquad\qquad\text{(Stratonovich-sense)}$$

which is equal to the following Itô version SDE involving the Itô-Stratonovich correction term $\mathcal{H}$:

$$d\overleftarrow{X}_t = \frac{1}{2}\sigma(T-t)^2\Pi(\overleftarrow{X}_t)\left[\nabla f(\overleftarrow{X}_t) + 2\nabla\ln q_{T-t}(\overleftarrow{X}_t)\right]dt + \frac{1}{2}\sigma(T-t)^2\mathcal{H}(\overleftarrow{X}_t)dt$$

$$\underbrace{-\alpha\sigma(T-t)^2\nabla J(\overleftarrow{X}_t)^T G^\dagger(\overleftarrow{X}_t)J(\overleftarrow{X}_t)}_{\text{Landing term}}dt + \sigma(T-t)\Pi(\overleftarrow{X}_t)\circ d\bar{W}_t. \qquad\text{(Itô-sense)}$$

Similarly, from Proposition B.2, the forward process $[X_t, P_t]^T \sim \overleftarrow{P}_t$ of ULLA is given as below in the interior of $\Sigma$:

$$\begin{cases} dX_t = \sigma(t)^2 P_t dt \\ dP_t = \Pi(X_t)\left[\sigma(t)^2\left[-\nabla f(X_t) - \gamma P_t\right]dt + \sigma(t)\sqrt{2\gamma}\circ dW_t\right] \\ \qquad - \sigma(t)^2\nabla J(X_t)^T G^\dagger(X_t)\mathcal{H}_1(X_t, P_t)dt. \end{cases}$$

Therefore, Lemma B.5 again implies that the following stochastic process $[\overleftarrow{X}_t, \overleftarrow{P}_t]^T \sim p_t$ becomes the backward process of $X_t$ on the interior of $\Sigma$:

$$\begin{cases} d\overleftarrow{X}_t = -\sigma(T-t)^2\overleftarrow{P}_t dt \\ d\overleftarrow{P}_t = \Pi(\overleftarrow{X}_t)\left[\sigma(T-t)^2\left[\nabla f(\overleftarrow{X}_t)dt + \gamma\overleftarrow{P}_t dt + 2\gamma\nabla_p\ln q_{T-t}(\overleftarrow{X}_t, \overleftarrow{P}_t)\right]dt + \sigma(T-t)\sqrt{2\gamma}\circ d\bar{W}_t\right] \\ \qquad + \sigma(T-t)^2\nabla J(\overleftarrow{X}_t)^T G^\dagger(\overleftarrow{X}_t)\mathcal{H}_1(\overleftarrow{X}_t, \overleftarrow{P}_t)dt. \end{cases}$$

Also, the similar construction and proof in Proposition B.2 and Lemma B.3 show that adding both the landing term $-\alpha\sigma^2\nabla JG^\dagger J$ and the corresponding landing correction term $+\alpha\sigma^2\nabla JG^\dagger\mathcal{H}_2$ to the previous SDE imposes the same boundary condition as in the forward process. Hence, the backward process $[\overleftarrow{X}_t, \overleftarrow{P}_t]^T$ is provided as:

$$\begin{cases} d\overleftarrow{X}_t = -\sigma(T-t)^2\overleftarrow{P}_t dt \underbrace{-\alpha\sigma(T-t)^2\nabla J(\overleftarrow{X}_t)G^\dagger(\overleftarrow{X}_t)J(\overleftarrow{X}_t)dt}_{\text{Landing term}} \\ d\overleftarrow{P}_t = \Pi(\overleftarrow{X}_t)\left[\sigma(T-t)^2\left[\nabla f(\overleftarrow{X}_t)dt + \gamma\overleftarrow{P}_t dt + 2\gamma\nabla_p\ln q_{T-t}(\overleftarrow{X}_t, \overleftarrow{P}_t)\right]dt + \sigma(T-t)\sqrt{2\gamma}\circ d\bar{W}_t\right] \\ \qquad + \sigma(T-t)^2\nabla J(\overleftarrow{X}_t)^T G^\dagger(\overleftarrow{X}_t)\left[\mathcal{H}_1(\overleftarrow{X}_t, \overleftarrow{P}_t) + \underbrace{\alpha\mathcal{H}_2(\overleftarrow{X}_t, \overleftarrow{P}_t)}_{\text{Landing correction term}}\right]dt. \end{cases}$$

Because the Itô-Stratonovich correction term vanishes in the underdamped case Proposition B.2, the Itô version SDE can be recovered from the Stratonovich version SDE by chaining $\circ$ to $\cdot$ in the Brownian motion term $d\bar{W}_t$.

## B.4 DISCRETIZATION OF CONSTRAINED LANGEVIN DYNAMICS

In the discretization setup, we use a uniform grid $t_k = k\Delta t$ for $k = 0, .., N$ with terminal time $T = N\Delta T$. The forward trajectory uses state notations $X_k := X_{t_k}$ (or $[X_k, P_k]^T := [X_{t_k}, X_{t_k}]^T$) and updates in ascending index as $X_k \leftarrow X_{k+1} + \dots$. The backward trajectory is written in

descending index, using states $X'_k := \overleftarrow{X}_{T-t_k}$ (or $[X'_k, P'_k]^T := [\overleftarrow{X}_{T-t_k}, \overleftarrow{P}_{T-t_k}]^T$) and updating as $X'_k \leftarrow X'_{k+1} + \ldots$ for $k \in \{N-1, \ldots 0\}$. We define the noise schedule in continuous time and evaluate the schedule on the discrete grid. This yields

$$\sigma(t) := \sigma_{\min} + \frac{t}{T}\left(\sigma_{\max} - \sigma_{\min}\right), \quad \sigma_k := \sigma(t_k) = \sigma_{\min} + \frac{k}{N}\left(\sigma_{\max} - \sigma_{\min}\right).$$

We also denote $q_k, p_k$ to be the probability densities of $X_k$ (or $[X_k, P_k]^T$) and $X'_k$ (or $[X'_k, P'_k]^T$) so that $q_k = q_{t_k}$ and $p_k = p_{T-t_k}$.

**Discretization of OLLA**  For the discretization of OLLA, we use straightforward Euler-Maruyama (EM) discretization.

**Discretization of Forward OLLA**  Recall that the Itô version of the forward process SDE is provided as follows:

$$dX_t = \frac{\sigma(t)^2}{2}\left[-\Pi(X_t)\nabla f(X_t) - \alpha\sigma(t)^2\nabla J(X_t)^T G^\dagger(X_t)J(X_t)\right]dt + \frac{\sigma(t)^2}{2}\mathcal{H}(X_t)dt$$
$$+ \sigma(t)\Pi(X_t)dW_t.$$

Therefore, the EM discretization of the forward process SDE becomes:

$$X_{k+1} = X_k + \frac{\sigma_k^2}{2}\left[-\Pi(X_k)\nabla f(X_k) - \alpha\sigma_k^2\nabla J(X_k)^T G^\dagger(X_k)J(X_k)\right]\Delta t$$
$$+ \frac{\sigma_k^2}{2}\mathcal{H}(X_k)\Delta t + \sigma_k\sqrt{\Delta t}\Pi(X_k)\zeta_k, \qquad \text{(Forward-OLLA)}$$

where $\zeta_k \sim \mathcal{N}(0, I)$ is the standard Gaussian noise.

**Discretization of Backward OLLA**  Similarly, we note that following the backward SDE

$$d\overleftarrow{X}_t = \frac{1}{2}\sigma(T-t)^2\Pi(\overleftarrow{X}_t)\left[\nabla f(\overleftarrow{X}_t) + 2\nabla\ln q_{T-t}(\overleftarrow{X}_t)\right]\Delta t + \frac{1}{2}\sigma(T-t)^2\mathcal{H}(\overleftarrow{X}_t)dt$$
$$- \alpha\sigma(T-t)^2\nabla J(\overleftarrow{X}_t)^T G^\dagger(\overleftarrow{X}_t)J(\overleftarrow{X}_t)dt + \sigma(T-t)\Pi(\overleftarrow{X}_t) \circ d\bar{W}_t$$

can be discretized as follows:

$$\overleftarrow{X}_{t_{k+1}} = \overleftarrow{X}_{t_k} + \frac{1}{2}\sigma(T-t_k)^2\Pi(\overleftarrow{X}_{t_k})\left[\nabla f(\overleftarrow{X}_{t_k}) + 2\nabla\ln q_{T-t_k}(\overleftarrow{X}_{t_k})\right]\Delta t + \frac{1}{2}\sigma(T-t_k)^2\mathcal{H}(\overleftarrow{X}_{t_k})\Delta t$$
$$- \alpha\sigma(T-t_k)^2\nabla J(\overleftarrow{X}_{t_k})^T G^\dagger(\overleftarrow{X}_{t_k})J(\overleftarrow{X}_{t_k})\Delta t + \sigma(t_k)\sqrt{\Delta t}\Pi(\overleftarrow{X}_{t_k})\bar{\zeta}_k.$$

In our notation, this is equivalent to

$$X'_k = X'_{k+1} + \frac{1}{2}\sigma_{k+1}^2\Pi(X'_{k+1})\left[\nabla f(X'_{k+1}) + 2\nabla\ln q_{k+1}(X'_{k+1})\right]\Delta t + \frac{1}{2}\sigma_{k+1}^2\mathcal{H}(X'_{k+1})\Delta t$$
$$- \alpha\sigma_{k+1}^2\nabla J(X'_{k+1})^T G^\dagger(X'_{k+1})J(X'_{k+1})\Delta t + \sigma_{k+1}\sqrt{\Delta t}\Pi(X'_{k+1})\zeta'_{k+1}$$
$$\text{(Backward-OLLA)}$$

by changing $k \leftarrow N - (k+1)$.

**Discretized algorithm of constrained overdamped via Lagrangian multiplier**  When we are available to use Newton's method, we replace the explicit normal drifts $\mathcal{H}, -\alpha\nabla J^T G^\dagger J$ terms by a position projection via Lagrangian multipliers $\lambda$ at each step so that $J(x) = 0$ is satisfied. Under this way, we recover the following discretization of constrained overdamped Langevin dynamics using Lagrangian multiplier:

$$X_{k+1} = X_k - \frac{\sigma_k^2}{2}\Pi(X_k)\nabla f(X_k)\Delta t + \sigma_k\sqrt{\Delta t}\Pi(X_k)\zeta_k + \nabla J(X_k)^T\lambda_k \quad \text{(Forward-OLLA-P)}$$

with $\lambda_k$ such that $J(X_{k+1}) = 0$. Similarly, the backward process can be obtained in a similar way as follows:

$$X'_k = X'_{k+1} + \frac{1}{2}\sigma_{k+1}^2\Pi(X'_{k+1})\left[\nabla f(X'_{k+1}) + 2\nabla\ln q_{k+1}(X'_{k+1})\right]\Delta t + \sigma_{k+1}\sqrt{\Delta t}\Pi(X'_{k+1})\zeta'_{k+1}$$
$$+ \nabla J(X'_{k+1})^T\lambda_{k+1} \qquad \text{(Backward-OLLA-P)}$$

with $\lambda_{k+1}$ such that $J(X'_k) = 0$.

**Discretiztaion of ULLA**  For the discretization of ULLA, we use a 1st order $O\tilde{B}A$ splitting scheme which uses an approximated $B$ process to compute the curvature correction term at $P_k$ with $\mathcal{O}(\Delta t)$ error. When discretizing, we use a collapsing technique to remove the momentum update rule. Because this collapsing technique requires saving only the position $X_k$'s. it is beneficial for saving memory, especially when saving forward trajectories is necessary.

**Discretization of Forward ULLA**  From the previous subsection, we recall the following Itô version of the forward process SDE:

$$\begin{cases} dX_t = \sigma(t)^2 P_t dt - \alpha\sigma(t)\nabla J(X_t)G^\dagger(X_t)J(X_t)dt \\ dP_t = \Pi(X_t)\left[\sigma(t)^2\left[-\nabla f(X_t) - \gamma P_t\right]dt + \sigma(t)\sqrt{2\gamma}dW_t\right] \\ \qquad - \sigma(t)^2\nabla J(X_t)^T G^\dagger(X_t)\left[\mathcal{H}_1(X_t, P_t) - \alpha\mathcal{H}_2(X_t, P_t)\right]dt. \end{cases}$$

Under $O\tilde{B}A$ scheme, this can be split into three parts $O, \tilde{B}, A$ as follows:

$$O: \begin{cases} dX_t = 0 \\ dP_t = -\sigma(t)^2\Pi(X_t)\gamma P_t dt + \sigma(t)\Pi(X_t)\sqrt{2\gamma}dW_t \end{cases} +$$

$$\tilde{B}: \begin{cases} dX_t = 0 \\ dP_t = -\sigma(t)^2\left(\Pi(X_t)\nabla f(X_t)dt + \nabla J(X_t)^T G^\dagger(X_t)\left[\mathcal{H}_1(X_t, P_t) - \alpha\mathcal{H}_2(X_t, P_t)\right]dt\right) \end{cases} +$$

$$A: \begin{cases} dX_t = \sigma(t)^2 P_t dt - \alpha\sigma(t)^2\nabla J(X_t)G^\dagger(X_t)J(X_t)dt \\ dP_t = 0. \end{cases}$$

First, we integrate $O$ step from $t = t_k$ to $t = t_{k+1}$ with $X_t$ and $\sigma(t)$ frozen at $t = t_k$. Then, it gives

$$\begin{cases} X^O_{k+1} = X_k \\ P^O_{k+1} = (1 - \Pi(X_k))P_k + a_k\Pi(X_k)P_k + \sqrt{1 - a_k^2}\Pi(X_k)\zeta_k = a_k\Pi(X_k)P_k + \sqrt{1 - a_k^2}\Pi(X_k)\zeta_k \end{cases}$$

where $a_k := e^{-\gamma\sigma_k^2\Delta t}$ and we used the assumption that $P_k \in T_{X_k}\Sigma$, while will be guaranteed on the collapsing technique. Next, we integrate $\tilde{B}$ step with same integration domain with $X_t, \sigma(t)$ frozen at $t = t_k$. Then, we have:

$$\begin{cases} X^{O\tilde{B}}_{k+1} = X^O_{k+1} + 0 = X_k \\ P^{O\tilde{B}}_{k+1} = P^O_{k+1} - \sigma_k^2\Pi(X_k)\nabla f(X_k)\Delta t - \sigma_k^2\nabla J(X_k)^T G^\dagger(X_k)\left[\mathcal{H}_1(X_k, P_k) - \alpha\mathcal{H}_2(X_k, P_k)\right]\Delta t. \end{cases}$$

Similarly, integrating $A$ step with frozen $X_t, \sigma(t)$ gives :

$$\begin{cases} X_{k+1} = X^{O\tilde{B}}_{k+1} + \sigma_k^2 P^{O\tilde{B}}_k\Delta t - \alpha\sigma_k^2\nabla J(X_k)^T G(X_k)^\dagger J(X_k)\Delta t \\ P_{k+1} = P^{O\tilde{B}}_{k+1} + 0 = P^{O\tilde{B}}_{k+1}, \end{cases}$$

which is equivalent to

$$\begin{cases} P_{k+1} = \Pi(X_k)\left[a_k P_k - \sigma_k^2\nabla f(X_k)\Delta t + \sqrt{1 - a_k^2}\zeta_k\right] \\ \qquad - \sigma_k^2\nabla J(X_k)^T G^\dagger(X_k)\left[\mathcal{H}_1(X_k, P_k) - \alpha\mathcal{H}_2(X_k, P_k)\right]\Delta t \\ X_{k+1} = X_k + \sigma_k^2 P_{k+1}\Delta t - \alpha\sigma_k^2\nabla J(X_k)^T G(X_k)^\dagger J(X_k)\Delta t. \end{cases}$$

We note that this can be collapsed into one update rule with respect to $X$ state as follows:

$$X_{k+1} = X_k + \sigma_k^2\Delta t\Pi(X_k)\left[a_k P_k - \sigma_k^2\nabla f(X_k)\Delta t + \sqrt{1 - a_k^2}\zeta_k\right]$$

$$- \alpha\sigma_k^2\nabla J(X_k)^T G(X_k)^\dagger J(X_k)\Delta t - \sigma_k^4\Delta t^2\nabla J(X_k)^T G(X_k)^\dagger\left[\mathcal{H}_1(X_k, P_k) - \alpha\mathcal{H}_2(X_k, P_k)\right].$$

Also, we observe that $P_k$ can be recovered as $P_k = \Pi(X_{k-1})\left(\frac{X_k - X_{k-1}}{\sigma_{k-1}^2\Delta t}\right)$ from the previous recursion which again can be approximated by $P_k \approx \tilde{P}_k := \Pi(X_k)\left(\frac{X_k - X_{k-1}}{\sigma_{k-1}^2\Delta t}\right)$ with error $\mathcal{O}(\Delta t)$,

guaranteeing the 1st order numerical error and $P_k \in T_{X_k}\Sigma$ assumption on $O$ step. Therefore, the final update rule for the forward process of ULLA becomes:

$$X_{k+1} = X_k + \sigma_k^2 \Delta t \Pi(X_k) \left[ a_k \tilde{P}_k - \sigma_k^2 \nabla f(X_k)\Delta t + \sqrt{1 - a_k^2}\zeta_k \right] \qquad \text{(Forward-ULLA)}$$

$$- \alpha\sigma_k^2 \nabla J(X_k)^T G(X_k)^\dagger J(X_k)\Delta t - \sigma_k^4 \Delta t^2 \nabla J(X_k)^T G^\dagger(X_k) \left[ \mathcal{H}_1(X_k, \tilde{P}_k) - \alpha\mathcal{H}_2(X_k, \tilde{P}_k) \right]$$

with $\tilde{P}_k := \Pi(X_k)\left( \frac{X_k - X_{k-1}}{\sigma_{k-1}^2 \Delta t} \right)$ for $k \in \{1, ..., N-1\}$ and $\tilde{P}_0 = \Pi(X_0)\zeta \in T_{X_0}^\Sigma$ where $\zeta \sim \mathcal{N}(0, I)$.

**Discretization of Backward ULLA**   For the discretization of backward process SDE, we recall the following backward process SDE of ULLA:

$$\begin{cases} d\overleftarrow{X}_t = -\sigma(T-t)^2 \overleftarrow{P}_t dt - \alpha\sigma(T-t)^2 \nabla J(\overleftarrow{X}_t)G^\dagger(\overleftarrow{X}_t)J(\overleftarrow{X}_t)dt \\ d\overleftarrow{P}_t = \Pi(\overleftarrow{X}_t)\left[ \sigma(T-t)^2 \left[ \nabla f(\overleftarrow{X}_t)dt + \gamma\overleftarrow{P}_t dt + 2\gamma\nabla_p \ln q_{T-t}(\overleftarrow{X}_t, \overleftarrow{P}_t) \right] dt + \sigma(T-t)\sqrt{2\gamma}d\bar{W}_t \right] \\ \qquad + \sigma(T-t)^2 \nabla J(\overleftarrow{X}_t)^T G^\dagger(\overleftarrow{X}_t) \left[ \mathcal{H}_1(\overleftarrow{X}_t, \overleftarrow{P}_t) + \alpha\mathcal{H}_2(\overleftarrow{X}_t, \overleftarrow{P}_t) \right] dt. \end{cases}$$

For the backward process of the $O\tilde{B}A$ scheme, the $2\sigma(T-t)^2\Pi(\overleftarrow{X}_t)\gamma\overleftarrow{P}_t$ is added to $\tilde{B}$ step to guarantee the stable non-exploding OU process at $O$ step. We remark that the similar trick to handle $O$ step was previously used in Dockhorn et al. [51]. Under this technique, each $O, \tilde{B}, A$ steps can be given as follows:

$$O: \begin{cases} d\overleftarrow{X}_t = 0 \\ d\overleftarrow{P}_t = -\sigma(T-t)^2\Pi(\overleftarrow{X}_t)\gamma\overleftarrow{P}_t dt + \sigma(T-t)\Pi(\overleftarrow{X}_t)\sqrt{2\gamma}d\bar{W}_t \end{cases} +$$

$$\tilde{B}: \begin{cases} d\overleftarrow{X}_t = 0 \\ d\overleftarrow{P}_t = \sigma(T-t)^2\Pi(\overleftarrow{X}_t)\left( \nabla f(\overleftarrow{X}_t) + 2\gamma\nabla_p \ln q_{T-t}(\overleftarrow{X}_t, \overleftarrow{P}_t) + 2\gamma\overleftarrow{P}_t \right) dt \\ \qquad + \nabla J(\overleftarrow{X}_t)^T G^\dagger(\overleftarrow{X}_t) \left[ \mathcal{H}_1(\overleftarrow{X}_t, \overleftarrow{P}_t) + \alpha\mathcal{H}_2(\overleftarrow{X}_t, \overleftarrow{P}_t) \right] dt \end{cases} +$$

$$A: \begin{cases} d\overleftarrow{X}_t = \sigma(T-t)^2 \left( -\overleftarrow{P}_t dt - \alpha\nabla J(\overleftarrow{X}_t)G^\dagger(\overleftarrow{X}_t)J(\overleftarrow{X}_t)dt \right) \\ d\overleftarrow{P}_t = 0. \end{cases}$$

Integrating $O$ step from $t = t_k$ to $t = t_{k+1}$ with $\overleftarrow{X}_t$ and $\sigma(T-t)$ frozen at $t = t_k$. Then, it gives

$$\begin{cases} \overleftarrow{X}_{t_{k+1}}^O = \overleftarrow{X}_{t_k} \\ \overleftarrow{P}_{t_{k+1}}^O = a_{N-k}\Pi(\overleftarrow{X}_{t_k})\overleftarrow{P}_{t_k} + \sqrt{1 - a_{N-k}^2}\Pi(\overleftarrow{X}_{t_k})\bar{\zeta}_k, \end{cases}$$

where $a_{N-k} = e^{-\gamma\sigma_{N-k}^2\Delta t}$ and we again used the assumption that $\overleftarrow{P}_{t_k} \in T_{\overleftarrow{X}_{t_k}}\Sigma$, which will be guaranteed via the collapsing technique later on. Next, integrating $\tilde{B}$ step with the same integration domain with $\overleftarrow{X}_t, \sigma(t)$ frozen at $t = t_k$ gives:

$$\begin{cases} \overleftarrow{X}_{t_{k+1}}^{O\tilde{B}} = \overleftarrow{X}_{t_{k+1}}^O + 0 = \overleftarrow{X}_{t_k} \\ \overleftarrow{P}_{t_{k+1}}^{O\tilde{B}} = \overleftarrow{P}_{t_{k+1}}^O + \sigma_{N-k}^2\Pi(\overleftarrow{X}_{t_k})\left( \nabla f(\overleftarrow{X}_{t_k}) + 2\gamma\nabla_p \ln q_{T-t_k}(\overleftarrow{X}_{t_k}, \overleftarrow{P}_{t_k}) + 2\gamma\overleftarrow{P}_{t_k} \right) \Delta t \\ \qquad + \sigma_{N-k}^2 \nabla J(\overleftarrow{X}_{t_k})^T G^\dagger(\overleftarrow{X}_{t_k}) \left[ \mathcal{H}_1(\overleftarrow{X}_{t_k}, \overleftarrow{P}_{t_k}) + \alpha\mathcal{H}_2(\overleftarrow{X}_{t_k}, \overleftarrow{P}_{t_k}) \right] \Delta t \end{cases}$$

Similarly, integrating $A$ step with frozen $\overleftarrow{X}_t, \sigma(t)$ gives :

$$
\begin{cases}
\overleftarrow{X}_{t_{k+1}} = \overleftarrow{X}_{t_{k+1}}^{O\tilde{B}} - \sigma_{N-k}^2 P_k^{O\tilde{B}}\Delta t - \alpha\sigma_{N-k}^2 \nabla J(\overleftarrow{X}_{t_k})^T G(\overleftarrow{X}_{t_k})^\dagger J(\overleftarrow{X}_{t_k})\Delta t \\
\overleftarrow{P}_{t_{k+1}} = \overleftarrow{P}_{t_{k+1}}^{O\tilde{B}} + 0 = \overleftarrow{P}_{t_{k+1}}^{O\tilde{B}}
\end{cases}
$$

which is equivalent to

$$
\begin{cases}
\overleftarrow{P}_{t_{k+1}} = \Pi(\overleftarrow{X}_{t_k})\left[a_{N-k}\overleftarrow{P}_{t_k} + \sigma_{N-k}^2\left(\nabla f(\overleftarrow{X}_{t_k}) + 2\gamma\nabla_p\ln q_{T-t_k}(\overleftarrow{X}_{t_k}, \overleftarrow{P}_{t_k}) + 2\gamma\overleftarrow{P}_{t_k}\right)\Delta t\right] \\
\quad + \sqrt{1-a_{N-k}^2}\Pi(\overleftarrow{X}_{t_k})\bar{\zeta}_k + \sigma_{N-k}^2\nabla J(\overleftarrow{X}_{t_k})^T G^\dagger(\overleftarrow{X}_{t_k})\left[\mathcal{H}_1(\overleftarrow{X}_{t_k}, \overleftarrow{P}_{t_k}) + \alpha\mathcal{H}_2(\overleftarrow{X}_{t_k}, \overleftarrow{P}_{t_k})\right]\Delta t \\
\overleftarrow{X}_{t_{k+1}} = \overleftarrow{X}_{t_k} - \sigma_{N-k}^2\overleftarrow{P}_{t_{k+1}}\Delta t - \alpha\sigma_{N-k}^2\nabla J(\overleftarrow{X}_{t_k})^T G(\overleftarrow{X}_{t_k})^\dagger J(\overleftarrow{X}_{t_k})\Delta t
\end{cases}
$$

Similarly as before, we note that $\overleftarrow{P}_{t_{k+1}}$ can be recovered as $\overleftarrow{P}_{t_{k+1}} = \Pi(\overleftarrow{X}_{t_{k-1}})\left(\frac{\overleftarrow{X}_{t_{k-1}} - \overleftarrow{X}_{t_k}}{\sigma_{N-k+1}^2\Delta t}\right)$

from the previous recursion, and it can be approximated by $\overleftarrow{\tilde{P}}_{t_{k+1}} \approx \Pi(\overleftarrow{X}_{t_k})\left(\frac{\overleftarrow{X}_{t_{k-1}} - \overleftarrow{X}_{t_k}}{\sigma_{N-k+1}^2\Delta t}\right)$ with

$\mathcal{O}(\Delta t)$ error. Therefore, we can collapse these two position-momentum updates into one single

position update with approximated $\overleftarrow{\tilde{P}}_{t_{k+1}}$ as follows:

$$
\overleftarrow{X}_{t_{k+1}} = \overleftarrow{X}_{t_k} + \sigma_{N-k}^2\Delta t\sqrt{1-a_{N-k}^2}\Pi(\overleftarrow{X}_{t_k})\bar{\zeta}_k - \alpha\sigma_{N-k}^2\nabla J(\overleftarrow{X}_{t_k})^T G(\overleftarrow{X}_{t_k})^\dagger J(\overleftarrow{X}_{t_k})\Delta t
$$
$$
- \sigma_{N-k}^2\Delta t\Pi(\overleftarrow{X}_{t_k})\left[a_{N-k}\overleftarrow{\tilde{P}}_{t_k} + \sigma_{N-k}^2\left(\nabla f(\overleftarrow{X}_{t_k}) + 2\gamma\left(\nabla_p\ln q_{T-t_k}(\overleftarrow{X}_{t_k}, \overleftarrow{\tilde{P}}_{t_k}) + \overleftarrow{\tilde{P}}_{t_k}\right)\right)\Delta t\right]
$$
$$
- \sigma_{N-k}^4\Delta t^2\nabla J(\overleftarrow{X}_{t_k})^T G^\dagger(\overleftarrow{X}_{t_k})\left[\mathcal{H}_1(\overleftarrow{X}_{t_k}, \overleftarrow{\tilde{P}}_{t_k}) + \alpha\mathcal{H}_2(\overleftarrow{X}_{t_k}, \overleftarrow{\tilde{P}}_{t_k})\right].
$$

Finally, we recover the following discretization of the backward SDE of ULLA by changing index $k \leftarrow N - (k+1)$:

$$
X'_k = X'_{k+1} + \sigma_{k+1}^2\Delta t\sqrt{1-a_{k+1}^2}\Delta t\Pi(X'_{k+1})\zeta'_{k+1} - \alpha\sigma_{k+1}^2\nabla J(X'_{k+1})^T G(X'_{k+1})^\dagger J(X'_{k+1})\Delta t -
$$
$$
\sigma_{k+1}^2\Delta t\Pi(X'_{k+1})\left[a_{k+1}\tilde{P}'_{k+1} + \sigma_{k+1}^2\left(\nabla f(X'_{k+1}) + 2\gamma\left(\nabla_p\ln q_{k+1}(X'_{k+1}, \tilde{P}'_{k+1}) + \tilde{P}'_{k+1}\right)\right)\Delta t\right]
$$
$$
- \sigma_{k+1}^4\Delta t^2\nabla J(X'_{k+1})^T G^\dagger(X'_{k+1})\left[\mathcal{H}_1(X'_{k+1}, \tilde{P}'_{k+1}) + \alpha\mathcal{H}_2(X'_{k+1}, \tilde{P}'_{k+1})\right]
$$
$$
\text{(Backward-ULLA)}
$$

with $\tilde{P}'_{k+1} := \Pi(X'_{k+1})\left(\frac{X'_{k+2} - X'_{k+1}}{\sigma_{k+2}^2\Delta t}\right)$ for $k \in \{0, ..., N-2\}$ and $\tilde{P}'_N = \Pi(X'_N)\zeta \in T_{X'_N}^\Sigma$ where $\zeta \sim \mathcal{N}(0, I)$.

**Discretized algorithm of constrained underdamped via Lagrangian multiplier**  Similar to constrained overdamped with Lagrangian multiplier, we replace the explicit normal drifts $-\alpha\nabla J^T G^\dagger J, \nabla J^T G^\dagger\mathcal{H}_1, \nabla J^T G^\dagger\mathcal{H}_2$ terms by a position projection via Lagrangian multipliers $\lambda$ at each step so that $J(x) = 0$ is satisfied. In the collapsed underdamped case, we also used the approximation to re-tangent the momentum $P$ by $\tilde{P}$. So, no separate momentum projection is required. Under this idea, we recover the following discretization of constrained underdamped Langevin dynamics via Lagrangian multiplier:

$$
X_{k+1} = X_k + \sigma_k^2\Delta t\Pi(X_k)\left[a_k\tilde{P}_k - \sigma_k^2\nabla f(X_k)\Delta t + \sqrt{1-a_k^2}\zeta_k\right] + \nabla J(X_k)^T\lambda_k
$$
$$
\text{(Forward-ULLA-P)}
$$

with $\lambda_k$ such that $J(X_{k+1}) = 0$. Similarly, we obtain the following backward discretization:

$$X'_k = X'_{k+1} + \sigma^2_{k+1}\Delta t\sqrt{1 - a^2_{k+1}}\Pi(X'_{k+1})\zeta'_{k+1} + \nabla J(X'_{k+1})\lambda_{k+1} -$$

$$\sigma^2_{k+1}\Delta t\Pi(X'_{k+1})\left[a_{k+1}\tilde{P}'_{k+1} + \sigma^2_{k+1}\left(\nabla f(X'_{k+1}) + 2\gamma\left(\nabla_p \ln q_{k+1}(X'_{k+1}, \tilde{P}'_{k+1}) + \tilde{P}'_{k+1}\right)\right)\Delta t\right]$$

(Backward-ULLA-P)

with $\lambda_{k+1}$ such that $J(X'_k) = 0$.

## C   DT-ELBO ON RIEMANNIAN MANIFOLD

In this section, we derive the DT-ELBO variational bounds of KL-divergence between the initial data distribution and the generated data distribution on the Riemannian manifold $\Sigma := \{x \in \mathbb{R}^d \mid h(x) = 0, g(x) \leq 0\}$.

Let $\{x_0, \ldots, x_N\} \subset \Sigma$ be the forward position trajectory produced by $N$ steps of a discretized sampler, and let $q(x_{k+1} \mid c_k)$ and $p_\theta(x_k \mid d_k)$ denote the forward and backward transition densities, respectively. The context $c_k, d_k$ depends on the history $\{x_0, \ldots, x_k\}$: for the overdamped case we take $c_k = \{x_k\}, d_k = \{x_{k+1}\}$, while for the (collapsed) underdamped case we use $c_k = \{x_k, x_{k-1}\}$ (with $c_0 = \{x_0, p_0\}$) and $d_k = \{x_{k+1}, x_{k+2}\}$ (with $d_{N-1} = \{x_N, p_N\}$). We set $q_0(x) = q_{\text{data}}(x)$ as the data distribution and $p(x) = p_N(x)$ as the prior. In what follows, all densities are understood with respect to the surface measure $d\sigma_\Sigma$ on $\Sigma$, and, for clarity of exposition, we will assume that Lagrangian multiplier methods (OLLA-P, ULLA-P) are used each iterate to satisfy $x_k \in \Sigma$; see the remark below for when this may fail under discretization.

**Remark 4** (On constraint enforcement and well-definedness of the DT-ELBO).   The proposed discretized constrained Langevin dynamics with landing does *not* automatically guarantee $x_k \in \Sigma$ at every step. By contrast, the RDDPM formulation [34] enforces feasibility at each time by solving the Lagrange multiplier system via Newton's method (see Proposition B.1, Proposition B.2) so that

$$J(X_t) = 0 \quad \text{with} \quad \nabla J(X_t)^T P_t = 0 \quad \text{(if underdamped)}$$

ensuring $X_t \in \Sigma$ and $P_t \in T_{X_t}\Sigma$ exactly. Throughout our DT-ELBO derivation we adopt the same *feasible-trajectory assumption*: we assume the multiplier solve succeeds by Newton's method so that the sampled forward path lies on $\Sigma$, i.e., $x_k \in \Sigma$ for all $k \in \{1, ..., N\}$.

This assumption is not merely cosmetic. If some $x_k \notin \Sigma$, the conditional densities that appear in the ELBO may be undefined or degenerate (e.g., in the overdamped case $p_\theta(x_k \mid x_{k+1})/q(x_{k+1} \mid x_k)$, and in the underdamped case $p_\theta(x_{k-1} \mid x_k, x_{k+1})/q(x_{k+1} \mid x_k, x_{k-1})$). Such off-manifold iterates can induce singularities in the ELBO and render the induced NLL numerically unstable even under small constraint violations. For this reason, in our experiments we do not report test NLL; instead, we evaluate with task-specific metrics which reflect downstream performance.

To solidify our framework, we later introduce the Conditional Wasserstein Path Matching (CWPM) framework in Appendix D, which drops the feasibility-trajectory assumption. Its training loss has the same DT-ELBO form up to the choice of training loss weights, justifying that the CWPM framework shares the same principle with DT-ELBO framework in the constrained Langevin dynamics with landing as the step size $\Delta t \to 0$.

**DT-ELBO for overdamped Langevin**   From the Markov property, the densities $q(x_0, x_{1:N})$, $p_\theta(x_{1:N}|x_0)$ are given by

$$q(x_1, \ldots x_N|x_0) = \prod_{k=0}^{N-1} q(x_{k+1}|x_k), \quad p_\theta(x_0, x_{1:N}) = p(x_N)\prod_{k=0}^{N-1} p_\theta(x_k|x_{k+1})$$

The common goal suggested in DDPM [1] (Euclidean space) or RDDPM [34] (Riemannian manifold) is to minimize $\mathsf{KL}^\Sigma(q_0||p_\theta)$. For this, we observe that

$$\mathsf{KL}^\Sigma(q_0(x_0)||p_\theta(x_0)) = \int q_0(x_0)\ln q_0(x_0)d\sigma_\Sigma(x_0)$$

$$- \underbrace{\int q_0(x_0)\ln\left(\int p_\theta(x_{0:N})d\sigma_\Sigma(x_{1:N})\right)d\sigma_\Sigma(x_0)}_{\text{Term (1)}}$$

and

$$\text{Term (1)} = \int q_0(x_0) \ln \left( \int \frac{p_\theta(x_{0:N})}{q(x_{1:N}|x_0)} q(x_{1:N}|x_0) d\sigma_\Sigma(x_{1:N}) \right) d\sigma_\Sigma(x_0)$$

$$\overset{\text{Jensen}}{\geq} \int q_0(x_0) q(x_{1:N}|x_0) \ln \left( \frac{p_\theta(x_{0:N})}{q(x_{1:N}|x_0)} \right) d\sigma_\Sigma(x_{1:N})$$

$$= \int q(x_{0:N}) \left( \sum_{k=0}^{N-1} \ln \frac{p_\theta(x_k|x_{k+1})}{q(x_{k+1}|x_k)} + \ln p(x_N) \right) d\sigma_\Sigma(x_{1:N})$$

Because we want to minimize $\mathsf{KL}(q_0(x_0)||p_\theta(x_0))$, the training loss $L^{\text{over}}(\theta)$ can be set to

$$L^{\text{over}}(\theta) = - \int q(x_{0:N}) \sum_{k=0}^{N-1} \ln p_\theta(x_k|x_{k+1}) d\sigma_\Sigma(x_{1:N}) = -\mathbb{E}_{q(x_{0:N})} \left[ \sum_{k=0}^{N-1} \ln p_\theta(x_k|x_{k+1}) \right]$$

where the product surface measure is defined to be $d\sigma_\Sigma(x_1, ..x_N) := \prod_{k=1}^N d\sigma_\Sigma(x_k)$.

**Lemma C.1** (Backward transition density – overdamped [34]). *Suppose $x_{k+1} \in int(\Sigma)$ and assume the followings hold:*

1. *There exists a measurable set $\mathcal{F}_{x_{k+1}} \subset T_{x_{k+1}}\Sigma$ such that, for every $\eta \in \mathcal{F}_{x_{k+1}}$, the Newton's method returns a unique pair $(x, \lambda), x \in int(\Sigma)$ solving*

$$x = x_{k+1} + \mu_{k+1}^o(x_{k+1}) + \sigma_{k+1}\sqrt{\Delta t}\eta + \nabla J(x_{k+1})\lambda, \quad \text{with } \lambda \text{ s.t. } J(x) = 0$$

   *with the minimal-displacement normal correction and it fails for $\eta \notin \mathcal{F}_{x_{k+1}}$*

2. *The solver success probability is $1 - \epsilon_{x_{k+1}} := \mathbb{P}(\eta \in \mathcal{F}_{x_{k+1}}) \in (0, 1]$.*

3. *The map $\Phi_{k+1} : \mathcal{F}_{x_{k+1}} \to \Sigma_{x_{k+1}} := \Phi_{k+1}(\mathcal{F}_{x_{k+1}}) \subset int(\Sigma)$ with $\Phi_{k+1}(\eta) = x$ is a $C^1$ bijection.*

*Then, the backward transition density of OLLA-P with respect to surface measure $d\sigma_\Sigma$ is given as $p_\theta(x_k|x_{k+1})$ :*

$$p_\theta(x_k|x_{k+1}) = \frac{\left| \det(U(x_{k+1})^T U(x_k)) \right|}{(2\pi\sigma_{k+1}^2 \Delta t)^{\frac{d-m}{2}} (1 - \epsilon_{x_{k+1}})} \exp \left( -\frac{\|\Pi(x_{k+1})(x_k - \mu_{k+1}^o(x_{k+1}))\|^2}{2\sigma_{k+1}^2 \Delta t} \right)$$

*for $x_k \in \Sigma_{x_{k+1}}$, and $p_\theta(x_k|x_{k+1}) = 0$ outside of $\Sigma_{x_{k+1}}$, where*

$$\mu_{k+1}^o(x_{k+1}) := x_{k+1} + \frac{1}{2}\sigma_{k+1}^2 \Delta t \Pi(x_{k+1}) \left[ \nabla f(x_{k+1}) + s_\theta^{(k+1)}(x_{k+1}) \right]$$

*is the backward mean vector, and $U(x)$ is the orthonormal matrix whose column vectors form an orthonormal basis of $T_x\Sigma$ so that $U(x)U(x)^T = \Pi(x)$*

*Proof.* Recall that the backward discretization of OLLA-P is given as below:

$$x_k = x_{k+1} + \frac{1}{2}\sigma_{k+1}^2 \Pi(x_{k+1}) \left[ \nabla f(x_{k+1}) + 2s_\theta^{(k+1)}(x_{k+1}) \right] \Delta t + \sigma_{k+1}\sqrt{\Delta t}\Pi(x_{k+1})\zeta_{k+1}$$

$$+ \nabla J(x_{k+1})\lambda_{k+1}$$

with $\lambda_{k+1}$ such that $J(x_k) = 0$.

Let $\eta \sim \mathcal{N}(0, I_{d-m})$ on $T_{x_{k+1}}\Sigma$. Conditioning on the success event $\{\eta \in \mathcal{F}_{x_{k+1}}\}$, the conditional density of $\eta$ is given by

$$\phi(\zeta) = \frac{1}{(2\pi)^{\frac{d-m}{2}} (1 - \epsilon_{x_{k+1}})} \exp \left( -\frac{1}{2}\|\eta\|^2 \right) \mathbb{1}_{\eta \in \mathcal{F}_{x_{k+1}}}$$

From the assumption, for each $\eta \in \mathcal{F}_{x_{k+1}}$, there is a unique $x = \Phi_{k+1}(\eta) \in \Sigma$ solving

$$x = \mu_{k+1}^o(x_{k+1}) + \sigma_{k+1}\sqrt{\Delta t}\eta + \nabla J(x_{k+1})\lambda$$

with $\lambda$ such that $J(x) = 0$. Because $\Phi_{k+1}$ is bijection from $\mathcal{F}_{x_{k+1}} \subset T_{x_{k+1}}\Sigma$ onto $\Sigma_{x_{k+1}} :=$ $\Phi_{k+1}(\mathcal{F}_{x_{k+1}}) \subset \Sigma$, we can define $G_{x_{k+1}} := \Phi_{k+1}^{-1}$ and its Jacobian is given by $DG_{x_{k+1}}(x) = (\sigma_{k+1}\sqrt{\Delta t})^{-1} U(x_{k+1})^T U(x)$ for $x \in \Sigma_{x_{k+1}}$. Hence, we have

$$\left|\det(DG_{x_{k+1}}(x))\right| = (\sigma_{k+1}\sqrt{\Delta t})^{-(d-m)}\left|\det(U(x_{k+1})^T U(x))\right|$$

Lastly, we observe that, for $x \in \Sigma_{x_{k+1}}$, the pushforward of the conditional density $\phi$ by $\Phi_{k+1}$ yields the following density with respect to $d\sigma_\Sigma$:

$$p_\theta(x_k|x_{k+1}) = \frac{\left|\det(U(x_{k+1})^T U(x_k))\right|}{(2\pi\sigma_{k+1}^2 \Delta t)^{\frac{d-m}{2}}(1 - \epsilon_{x_{k+1}})} \exp\left(-\frac{\|\Pi(x_{k+1})^T(x_k - \mu_{k+1}^o(x_{k+1}))\|^2}{2\sigma_{k+1}^2 \Delta t}\right)$$

which becomes zero outside $\Sigma_{x_{k+1}}$ (equivalently, when projection failure happens). $\qquad\square$

Therefore, assuming the forward path trajectory $\{x_0, ..., x_N\} \subset \text{int}(\Sigma)$, the training loss has the following upper bound:

$$L^{\text{over}}(\theta) = -\mathbb{E}_{q(x_{0:N})}\left[\sum_{k=0}^{N-1} \ln p_\theta(x_k|x_{k+1})\right] \qquad\qquad \text{(Training loss-OLLA-P)}$$

$$\leq \mathbb{E}_{q(x_{0:N})}\left[\sum_{k=0}^{N-1}\left(\frac{\|\Pi(x_{k+1})^T(x_k - \mu_{k+1}^o(x_{k+1}))\|^2}{2\sigma_{k+1}^2 \Delta t}\right)\right] + C$$

where $C := \sum_{k=0}^{N-1}\left[-\ln\left(\frac{\left|\det(U(x_{k+1})^T U(x_k))\right|}{(2\pi\sigma_{k+1}^2 \Delta t)^{\frac{d-m}{2}}}\right)\right]$ is constant with respect to $\theta$ and the last inequality is obtained using $\epsilon_{x_{k+1}} \leq 1$.

**DT-ELBO for underdamped Langevin**    In the collapsed underdamped setting, we keep only positions, which makes the forward chain a second-order Markov chain in $\{x_0, ...x_N\}$. Similarly, applying the same Jensen inequality argument used for the overdamped case gives the ELBO:

$$\text{Term (1)} = \int q_0(x_0) \ln\left(\int \frac{p_\theta(x_{0:N})}{q(x_{1:N}|x_0, p_0)} q(p_0|x_0) q(x_{1:N}|x_0, p_0) d\sigma_{T_{x_0}\Sigma}(p_0) d\sigma_\Sigma(x_{1:N})\right) d\sigma_\Sigma(x_0)$$

$$\overset{\text{Jensen}}{\geq} \int q(x_0, p_0) q(x_{1:N}|x_0, p_0) \ln\left(\frac{p_\theta(x_{0:N})}{q(x_{1:N}|x_0, p_0)}\right) d\sigma_{T_{x_0}\Sigma}(p_0) d\sigma_\Sigma(x_{0:N})$$

$$\overset{\text{Jensen}}{\geq} \mathbb{E}_{q(x_{0:N}, p_0)}\mathbb{E}_{p(p_N|x_N)}\left[\sum_{k=1}^{N-1}\ln\left(\frac{p_\theta(x_{k-1}|x_k, x_{k+1})}{q(x_{k+1}|x_k, x_{k-1})}\right) + \ln\left(\frac{p_\theta(x_{N-1}|x_N, p_N)}{q(x_1|x_0, p_0)}\right) + \ln p(x_N)\right]$$

where the last inequality holds because $p_\theta(x_{0:N})$ has the following form:

$$p_\theta(x_{0:N}) = p(x_N)\int p(p_N|x_N) p_\theta(x_{N-1}|x_N, p_N) \prod_{k=1}^{N-1} p_\theta(x_{k-1}|x_k, x_{k+1}) d\sigma_{T_{x_N}\Sigma}(p_N)$$

Therefore, the training loss $L^{\text{under}}(\theta)$ is naturally given as follows to minimize $\mathsf{KL}^\Sigma(q_0(x_0)\|p_\theta(x_0))$:

$$L^{\text{under}}(\theta) = -\mathbb{E}_{q(x_{0:N}, p_0)}\mathbb{E}_{\rho_N(p_N|x_N)}\left[\sum_{k=0}^{N-2}\ln p_\theta(x_k|x_{k+1}, x_{k+2}) + \ln p_\theta(x_{N-1}|x_N, p_N)\right]$$

$$= -\mathbb{E}_{q(x_{0:N}, p_0)}\mathbb{E}_{\rho_N(p_N|x_N)}\left[\sum_{k=0}^{N-1}\ln p_\theta(x_k|x_{k+1}, x_{k+2})\right]$$

where $\rho_N(p_N|x_N) \propto \exp\left(-\frac{1}{2}\|\Pi(x_N)p_N\|_2^2\right)$ is the density of the momentum prior density, and we used the notation abusing $x_{N+1} := p_N$ for notational convenience.

**Lemma C.2** (Backward transition density – underdamped). *Suppose $x_{k+1} \in int(\Sigma)$ and assume the followings hold:*

1. *There exists a measurable set $\mathcal{F}_{x_{k+1}} \subset T_{x_{k+1}}\Sigma$ such that, for every $\eta \in \mathcal{F}_{x_{k+1}}$, the Newton's method returns a unique pair $(x, \lambda), x \in int(\Sigma)$ solving*

$$x = \mu^u_{k+1}(x_{k+1}, x_{k+2}) + \sigma^2_{k+1}\Delta t\sqrt{1 - a^2_{k+1}}\eta + \nabla J(x_{k+1})\lambda, \text{ with } \lambda \text{ s.t. } J(x) = 0$$

*with the minimal-displacement normal correction and it fails for $\eta \notin \mathcal{F}_{x_{k+1}}$*

2. *The solver success probability is $1 - \epsilon_{x_{k+1}} := \mathbb{P}(\eta \in \mathcal{F}_{x_{k+1}}) \in (0, 1]$.*

3. *The map $\Phi_{k+1} : \mathcal{F}_{x_{k+1}} \to \Sigma_{x_{k+1}} := \Phi_{k+1}(\mathcal{F}_{x_{k+1}}) \subset int(\Sigma)$ with $\Phi_{k+1}(\eta) = x$ is a $C^1$ bijection.*

*Then, the backward transition density of ULLA-P with respect to surface measure $d\sigma_\Sigma$ is given as $p_\theta(x_k|x_{k+1})$ :*

$$p_\theta(x_k|x_{k+1}) = \frac{\left|\det(U(x_{k+1})^T U(x_k))\right|}{(2\pi\sigma^2_{k+1}\Delta t)^{\frac{d-m}{2}}(1 - \epsilon_{x_{k+1}})} \exp\left(-\frac{\|\Pi(x_{k+1})^T(x_k - \mu^u_{k+1}(x_{k+1}, x_{k+2}))\|^2}{2\sigma^4_{k+1}\Delta t^2(1 - a_{k+1})^2}\right)$$

*for $x_k \in \Sigma_{x_{k+1}}$, and $p_\theta(x_k|x_{k+1}) = 0$ outside of $\Sigma_{x_{k+1}}$, where*

$$\mu^u_{k+1} = x_{k+1} + \sigma^2_{k+1}\Delta t\Pi(x_{k+1})\left[a_{k+1}\tilde{p}_{k+1} + \sigma^2_{k+1}\Delta t\left(\nabla f(x_{k+1}) + s^\theta_{k+1}(x_{k+1}, \tilde{p}_{k+1})\right)\right]$$

*is the backward mean vector.*

*Proof.* Recall that the backward discretization of ULLA-P is given as below:

$$x_k = x_{k+1} + \sigma^2_{k+1}\Delta t\sqrt{1 - a^2_{k+1}}\Pi(x_{k+1})\zeta_{k+1} + \nabla J(x_{k+1})\lambda_{k+1} +$$
$$\sigma^2_{k+1}\Delta t\Pi(x_{k+1})\left[a_{k+1}\tilde{p}_{k+1} + \sigma^2_{k+1}\left(\nabla f(x_{k+1}) + s^\theta_{k+1}(x_{k+1}, \tilde{p}_{k+1})\right)\Delta t\right]$$

with $\lambda_{k+1}$ such that $J(x_k) = 0$. Using the same proof as the overdamped case, we observe that $\Phi_{k+1}$ is bijection from $\mathcal{F}_{x_{k+1}} \subset T_{x_{k+1}}\Sigma$ onto $\Sigma_{x_{k+1}} := \Phi_{k+1}(\mathcal{F}_{x_{k+1}}) \subset \Sigma$, and we can define $G_{x_{k+1}} := \Phi^{-1}_{k+1}$ whose Jacobian is given by $DG_{x_{k+1}}(x) = (\sigma_{k+1}\sqrt{\Delta t})^{-1}U(x_{k+1})^T U(x)$ for $x \in \Sigma_{x_{k+1}}$. Therefore, the determinant of this is:

$$\left|\det(DG_{x_{k+1}}(x))\right| = (\sigma_{k+1}\sqrt{\Delta t})^{-(d-m)}\left|\det(U(x_{k+1})^T U(x))\right|$$

Lastly, we observe that, for $x \in \Sigma_{x_{k+1}}$, the pushforward of the conditional density of $\eta$ (defined on the overdamped case proof) by $\Phi_{k+1}$ yields the following density with respect to $d\sigma_\Sigma$:

$$p_\theta(x_k|x_{k+1}) = \frac{\left|\det(U(x_{k+1})^T U(x_k))\right|}{(2\pi\sigma^2_{k+1}\Delta t)^{\frac{d-m}{2}}(1 - \epsilon_{x_{k+1}})} \exp\left(-\frac{\|\Pi(x_{k+1})^T(x_k - \mu^u_{k+1}(x_{k+1}, x_{k+2}))\|^2}{2\sigma^4_{k+1}\Delta t^2(1 - a_{k+1})^2}\right)$$

which becomes zero outside $\Sigma_{x_{k+1}}$ (equivalently, when projection failure happens). □

Therefore, assuming the forward path trajectory $\{x_0, ..., x_N\} \subset int(\Sigma)$, the training loss has the following upper bound (with notation $x_{N+1} := p_N$):

$$L^{\text{under}}(\theta) = -\mathbb{E}_{q(x_{0:N}, p_0)}\mathbb{E}_{\rho_N(p_N|x_N)}\left[\sum_{k=0}^{N-1} \ln p_\theta(x_k|x_{k+1}, x_{k+2})\right] \quad \text{(Training loss-ULLA-P)}$$

$$\leq \mathbb{E}_{q(x_{0:N})}\mathbb{E}_{\rho_N(p_N|x_N)}\left[\sum_{k=0}^{N-1}\left(\frac{\|\Pi(x_{k+1})^T(x_k - \mu^u_{k+1}(x_{k+1}, x_{k+2}))\|^2}{2\sigma^4_{k+1}\Delta t^2(1 - a_{k+1})^2}\right)\right] + C$$

where $C := \sum_{k=0}^{N-1}\left[-\ln\left(\frac{\left|\det(U(x_{k+1})^T U(x_k))\right|}{(2\pi\sigma^2_{k+1}\Delta t)^{\frac{d-m}{2}}}\right)\right]$ is constant with respect to $\theta$ and the last inequality is again obtained using $\epsilon_{x_{k+1}} \leq 1$.

## D  CONDITIONAL WASSERSTEIN PATH MATCHING (CWPM)

Let $q_0, q_1, ..., q_N$ be the marginal forward probability densities at each step, evolved by discrete forward transition kernels $Q_k = q(x_{k+1}|c_k)$. Similarly, define $p_N, p_{N-1}^\theta, ..., p_0^\theta$ to be the marginal backward probability densities driven by parameterized backward transition kernels $T_{k+1}^\theta = p^\theta(x_k|d_k)$. Similarly as in DT-ELBO, the context vector is fixed to be $c_k = \{x_k\}$, $d_k = \{x_{k+1}\}$ for the overdamped case, while for the (collapsed) underdamped case we use $c_k = \{x_k, x_{k-1}\}$ (with $c_0 = \{x_0, p_0\}$) and $d_k = \{x_{k+1}, x_{k+2}\}$ (with $d_{N-1} = \{x_N, p_N\}$).

Our goal is to minimize $W_2(q_0, p_0^\theta)$ so that the Wasserstein-2 distance between data distribution and generated data distribution becomes close to each other.

**CWPM framework for the overdamped**     We first define **circuitous** density at step $k$ as

$$\sigma_k := q_k T_k^\theta T_{k-1}^\theta, ..., T_1^\theta \quad \text{for } k \in \{1, ..., N\} \qquad \sigma_0 := q_0.$$

We assume that for any probability measure $\mu, \nu$ in $\mathbb{R}^d$, there exists $K_{k+1} > 0$ such that

$$W_2(\mu T_{k+1}^\theta, \nu T_{k+1}^\theta) \le K_{k+1} W_2(\mu, \nu) + \mathcal{O}(\sqrt{\Delta t}) \qquad \text{(stepwise-Lipschitz)}$$

Under this assumption, we can choose $\Lambda_{k+1} > 0$ such that

$$W_2(\sigma_k, \sigma_{k+1}) \le \Lambda_{k+1} W_2(q_k, q_{k+1} T_{k+1}^\theta) + \mathcal{O}(\sqrt{\Delta t})$$

for $k \in \{0, ..., N-1\}$. We note that Lemma D.1 implies that such $\Lambda_k$ exists without stepwise-Lipschitz assumption when the function class of score function and the constraint functions are sufficiently regular.

Under this setup, from the triangular inequality of $W_2$, it holds that

$$W_2(q_0, p_0^\theta) \le W_2(q_0, \sigma_N) + W_2(\sigma_N, p_0^\theta) \le \sum_{k=0}^{N-1} W_2(\sigma_k, \sigma_{k+1}) + W_2(\sigma_N, p_0^\theta)$$

$$\le \sum_{k=0}^{N-1} \Lambda_{k+1} W_2(q_k, q_{k+1} T_{k+1}^\theta) + \Lambda_{N+1} W_2(q_N, p_N) + \mathcal{O}(\sqrt{\Delta t})$$

$$\le \sum_{k=0}^{N-1} \Lambda_{k+1} \mathbb{E}_{x_{k+1} \sim q_{k+1}} \left[ W_2(q_{k|k+1}(\cdot|x_{k+1}), T_{k+1}^\theta(\cdot|x_{k+1}) \right] + \Lambda_{N+1} W_2(q_N, p_N) + \mathcal{O}(\sqrt{\Delta t})$$

Now, we know that $T_{k+1}^\theta(\cdot|x_{k+1}) = \mathcal{N}(\mu_{k+1}^o(x_{k+1}), \sigma_{k+1}^2 \Pi(x_{k+1}))$ and let $(\mu_{k|k+1}, \Sigma_{k|k+1})$ be the mean and covariance of $q_{k-1|k}$. Then, from the relation between Gelbrich distance and 2-Wasserstein distance [38, 37], we have:

$$\mathbb{E}_{x_{k+1} \sim q_{k+1}} W_2(q_{k|k+1}, T_{k+1}^\theta(\cdot|x_{k+1}))^2 = \mathbb{E}_{x_{k+1} \sim q_{k+1}} \|\mu_{k|k+1}(x_{k+1}) - \mu_{k+1}^o(x_{k+1})\|$$

$$+ \mathbb{E}_{x_{k+1} \sim q_{k+1}} \left[ \mathsf{Tr}\left(\Sigma_{k|k+1}(x_{k+1})\right) + \mathsf{Tr}\left(\sigma_{k+1}^2 \Pi(x_{k+1})\right) \right] + \mathbb{E}_{x_{k+1} \sim q_{k+1}} \Delta_{k+1}(x_{k+1})$$

$$- 2\mathbb{E}_{x_{k+1} \sim q_{k+1}} \left[ \mathsf{Tr}\left( \left(\Sigma_{k|k+1}^{1/2}(x_{k+1}) \sigma_{k+1}^2 \Pi(x_{k+1}) \Sigma_{k|k+1}^{1/2}(x_{k+1})\right)^{1/2} \right) \right]$$

$$= \mathbb{E}_{x_k \sim q_k, x_{k+1} \sim q_{k+1}} \|x_k - \mu_k^o(x_{k+1})\| + \mathsf{Tr}\left(\sigma_{k+1}^2 \Pi(x_{k+1})\right) + \mathbb{E}_{x_{k+1} \sim q_{k+1}} \Delta_{k+1}(x_{k+1})$$

$$- 2\mathbb{E}_{x_{k+1} \sim q_{k+1}} \left[ \mathsf{Tr}\left( \left(\Sigma_{k|k+1}^{1/2}(x_{k+1}) \sigma_{k+1}^2 \Pi(x_{k+1}) \Sigma_{k|k+1}^{1/2}(x_{k+1})\right)^{1/2} \right) \right]$$

where we used the law of total variance for the last equality, by observing that $x_k = \mu_{k|k+1}(x_{k+1}) + \epsilon \sim q_k$ with $\mathbb{E}[\epsilon|x_{k+1}] = 0$ and $\text{Cov}(\epsilon|x_{k+1}) = \Sigma_{k|k+1}$. Note that $\Delta_{k+1}(x_{k+1}) \ge 0$ is the distance gap between Gelbrich distance and 2-Wasserstein distance (independent of $\theta$) so that it becomes zero if the true conditional $x_k|x_{k+1}$ follows the Gaussian distribution as

$$x_k|x_{k+1} \sim \mathcal{N}(\mu_{k|k+1}(x_{k+1}), \Sigma_{k|k+1}(x_{k+1}))$$

**Training loss (Overdamped) from CWPM**     We note that from the above bound, minimizing $\mathbb{E}_{x_k \sim q_k, x_{k+1} \sim q_{k+1}} \|x_k - \mu_k^o(x_{k+1})\|$ is to close the Wasserstein distance between $q_0, p_0^\theta$. Because $\|x_k - \mu_k^o(x_{k+1})\|^2$ can be decomposed into

$$\|x_k - \mu_k^o(x_{k+1})\|^2 = \|\Pi(x_{k+1})(x_k - \mu_k^o(x_{k+1}))\|^2 + \underbrace{\|(I - \Pi(x_{k+1}))(x_k - \mu_k^o(x_{k+1}))\|^2}_{\text{constant w.r.t } \theta}$$

and $\mu_k^o$ does not have $\theta$ dependency on normal (second) term, the natural choice of loss (leveraging the saved forward trajectories) is

$$
L_{\text{CWPM}}^{\text{over}}(\theta) = \mathbb{E}_{q(x_{0:N})} \left[ \sum_{k=0}^{N-1} \lambda(k) \|\Pi(x_{k+1}) (x_k - \mu_k^o(x_{k+1}))\|^2 \right]
$$

$$
= \mathbb{E}_{q(x_{0:N})} \left[ \sum_{k=0}^{N-1} \frac{\|\Pi(x_{k+1}) (x_k - \mu_k^o(x_{k+1}))\|^2}{2\sigma_{k+1}^2 \Delta t} \right]
$$

with some training loss weight $\lambda(k)$. We note that other seminar works [1, 39, 40] in diffusion model choose the weight proportional to the inverse of variance of corresponding term, which, in our case, becomes $\lambda(k) = \frac{1}{2\sigma_{k+1}^2 \Delta t}$ with proportional constant $1/2$. And, notably, this leads to the exactly the same training loss provided in DT-ELBO, (Lemma C.1) without the requirement $x_k \in \Sigma$.

**Lemma D.1** (Sufficient condition for $\Lambda_k < \infty$ – Overdamped). *Let the one-step landing backward update of OLLA (Equation Backward-OLLA) be*

$$
F_\theta^k(y, \zeta) = y + \Pi(y) b_\theta^k(y) \Delta t + \sigma_k \sqrt{\Delta t} \Pi(y) \zeta + \sigma_k^2 \phi(y) \Delta t, \quad \zeta \sim \mathcal{N}(0, I_d)
$$

*where $b_\theta^k := \frac{\sigma_k^2}{2} \left[ \nabla f(y) + 2s_\theta^k(y) \right]$ is the drift term and $\phi(y)$ is the normal term. Assume:*

1. *(Regularity of constraint functions)    There exists constants $c_\phi^0, c_\phi^1 < \infty$ such that for any square-integrable $Y$,*

$$
\mathbb{E}\|\phi(Y)\|^2 \le c_\phi^0 + c_\phi^1 \mathbb{E}\|Y\|^2
$$

2. *(Regularity of function class)    There exists $L_s, B_s < \infty$ independent of $\theta$ with*

$$
\|s_\theta^k(x) - s_\theta^k(y)\| \le L_s \|x - y\|, \quad \|s_\theta^k(x)\| \le B_s + L_s \|x\|
$$

*for all $k \in \{1, ..., N\}$ and assume $\nabla f$ is Lipschitz with constant $L_f$ so that*

$$
\|b_\theta^k(x) - b_\theta^k(y)\| \le L_b \|x - y\|, \quad \|b_\theta^k(x)\| \le C(1 + \|x\|)
$$

*for some constant $L_b, C, k \in \{1, ..., N\}$*

*Let $T_k^\theta$ be the associated Markov kernel to $F_\theta^k$. That is, $T_k^\theta(\cdot|y) = Law(F_\theta^k(y, \zeta))$. Then, for any probability measures $\mu, \nu$ in $\mathbb{R}^d$, we have*

$$
W_2(\mu T_k^\theta, \nu T_k^\theta) \le K_k W_2(\mu, \nu) + \mathcal{O}\left( \sqrt{\Delta t} + \Delta t \left( 1 + \sqrt{\mathbb{E}_{Y \sim \mu}\|Y\|^2} + \sqrt{\mathbb{E}_{Y' \sim \nu}\|Y'\|^2} \right) \right)
$$

*for some constant $K_k$. Also, if $\mu$ is given as $\mu = \rho T_i^\theta T_{i-1}^\theta \ldots T_j^\theta$ for $i \ge j$ and some density $\rho$ independent of $\theta$, the supremum of second moment of $\mu$ is finite under $\theta \in \Theta$:*

$$
\sup_{\theta \in \Theta} \mathbb{E}_{Y_\mu \sim \mu} \|Y_\mu\|^2 < \infty
$$

*Combining these two, we can guarantee the existence of $\Lambda_k > 0$ such that*

$$
W_2(\sigma_{k-1}, \sigma_k) \le \Lambda_k W_2(q_{k-1}, q_k T_k^\theta) + \mathcal{O}(\sqrt{\Delta t})
$$

*for $k \in \{1, ...N\}$.*

*Proof.* Let $(Y, Y')$ be the synchronous coupling for $W_2(\mu, \nu)$ with shared noise $\zeta \sim \mathcal{N}(0, I)$. Set

$$
\Delta F_\theta^k := F_\theta^k(Y, \zeta) - F_\theta^k(Y', \zeta) = (Y - Y') + \underbrace{\Delta t \left( \Pi(Y) b_\theta^k(Y) - \Pi(Y') B_\theta^k(Y') \right)}_{:=B}
$$

$$
+ \underbrace{\sigma_k \sqrt{\Delta t}(\Pi(Y) - \Pi(Y'))\zeta}_{:=N} + \underbrace{\sigma_k^2 \Delta t \left( \phi(Y) - \phi(Y') \right)}_{:=L}
$$

Then, using $(a + b + c + d)^2 \leq 2(a^2 + b^2 + c^2 + d^2)$, we have:

$$\mathbb{E}\|\Delta F_\theta^k\|^2 \leq 2\mathbb{E}\|Y - Y'\|^2 + 2\mathbb{E}\|B\|^2 + 2\mathbb{E}\|N\|^2 + 2\mathbb{E}\|L\|^2$$

Now, we note that

$$B = \Delta t \left(\Pi(Y)(b_\theta(Y) - b_\theta(Y')) + (\Pi(Y) - \Pi(Y')) b_\theta(Y')\right)$$

and it implies

$$\|B\| \leq \Delta t \left(L_b\|Y - Y'\| + 2\|b_\theta(Y')\|\right)$$

Therefore, we have

$$\mathbb{E}\|B\|^2 \leq 2L_b^2 \Delta t \mathbb{E}\|Y - Y'\|^2 + 8C\Delta t^2(1 + \mathbb{E}\|Y'\|^2)$$
$$= 2L_b^2 \Delta t \mathbb{E}\|Y - Y'\|^2 + 4C\Delta t^2(1 + \mathbb{E}\|Y'\|^2 + \mathbb{E}\|Y\|^2)$$

where the last equality comes by swapping $Y, Y'$ and taking average. Also, we note that

$$\mathbb{E}\|N\|^2 = \sigma_k^2 \Delta t \mathbb{E}\left[\|(\Pi(Y) - \Pi(Y')) \zeta\|\right] \leq 4\sigma_k^2 \Delta t \mathbb{E}\|\zeta\|^2$$

using $\|\Pi\| \leq 1$. Similarly, we observe that

$$\mathbb{E}\|L\|^2 \leq 2\sigma_k^4 \Delta t^2 \left(\mathbb{E}\|\phi(Y)\|^2 + \mathbb{E}\|\phi(Y')\|^2\right) \leq 2\sigma_k^4 \Delta t^2 \left(2c_\phi^0 + c_\phi^1 \left(\mathbb{E}\|Y\|^2 + \mathbb{E}\|Y'\|^2\right)\right)$$

using $(a - b)^2 \leq 2a^2 + 2b^2$. By collecting all terms, we obtain

$$\mathbb{E}\|\Delta F\|^2 \leq 2(1 + L_b^2 \Delta t)\mathbb{E}\|Y - Y'\|^2 + 8\sigma_k^2 \Delta t \mathbb{E}\|\zeta\|^2 + \Delta t^2 \tilde{C}(1 + \mathbb{E}\|Y\|^2 + \mathbb{E}\|Y'\|^2)$$

for some $\tilde{C} > 0$. By taking square-root and using $\sqrt{u + v + w} \leq \sqrt{u} + \sqrt{v} + \sqrt{w}$, we get

$$W_2(\mu T^\theta, \nu T^\theta) \leq K_k W_2(\mu, \nu) + \mathcal{O}\left(\sqrt{\Delta t} + \Delta t \left(1 + \sqrt{\mathbb{E}_{Y \sim \mu}\|Y\|^2} + \sqrt{\mathbb{E}_{Y' \sim \nu}\|Y'\|^2}\right)\right)$$

for some constant $K_k$. Also, following the similar algebraic techniques, one can show the following using the regularity assumptions:

$$\mathbb{E}\|F_\theta^k(Y, \zeta)\|^2 \leq \left[1 + a_k \Delta t + \mathcal{O}(\Delta t^2)\right] \mathbb{E}\|Y\|^2 + b_k \Delta t + \mathcal{O}(\Delta t^2)$$

for some $a_k, b_k > 0$ independent of $\theta$. So, once $\mu$ is given as $\mu = \rho T_i^\theta T_{i-1}^\theta \ldots T_j^\theta$ for $i \geq j$ and some density $\rho$ independent of $\theta$, then, by applying the recursive inequality above, we get:

$$\mathbb{E}_{Y_\mu \sim \mu}\|Y_\mu\|^2 \leq \prod_{k=i}^{j} \left(1 + a_k \Delta t + \mathcal{O}(\Delta t^2)\right) \mathbb{E}_{Y_\rho \sim \rho}\|Y_\rho\|^2 + \mathcal{O}(\Delta t)$$

and taking supremum over $\theta$, it implies that

$$\sup_\theta \mathbb{E}_{Y_\mu \sim \mu}\|Y_\mu\|^2 < \infty$$

because the constants and the density $\rho$ is independent of $\theta$. $\qquad\square$

**CWPM framework for the underdamped**   Let $y_k := (x_k, x_{k+1}) \in \mathbb{R}^{2d}$ with $x_k \sim q_k, x_{k+1} \sim q_{k+1}$ and let $\bar{q}_k$ be the law of $y_k$. The forward pair-kernel $\bar{Q}_k$ and backward pair-kernel $\bar{T}_{k+1}^\theta$ are

$$\bar{Q}_k \left(x_{k+1}, x_{k+2}|x_k, x_{k+1}\right) = \delta_{x_{k+1}} \otimes Q_k(x_{k+2}|x_k, x_{k+1})$$
$$\bar{T}_{k+1}^\theta \left(x_k, x_{k+1}|x_{k+1}, x_{k+2}\right) = T_{k+1}^\theta(x_k|x_k, x_{k+1}) \otimes \delta_{x_{k+1}}$$

Now, we similarly define the **circuitous** densities on pairs as

$$\bar{\sigma}_0 := \bar{q}_0, \qquad \bar{\sigma}_k := \bar{q}_k \bar{T}_k^\theta \cdots \bar{T}_1^\theta, \quad \text{for } k \in \{1, ..., N - 1\}$$

Assume the stepwise Lipschitz inequality on pairs holds such that there exits $\bar{K}_{k+1} > 0$

$$W_2(\mu \bar{T}_{k+1}^\theta, \nu \bar{T}_{k+1}^\theta) \leq \bar{K}_{k+1} W_2(\mu, \nu) + \mathcal{O}(\Delta t)$$

for any probability measure $\mu, \nu$ in $\mathbb{R}^d$. Then there exists finite $\bar{\Lambda}_{k+1} > 0$ such that

$$W_2(\bar{\sigma}_k, \bar{\sigma}_{k+1}) \leq \bar{\Lambda}_{k+1} W_2(\bar{q}_k, \bar{q}_{k+1} \bar{T}_{k+1}^\theta) + \mathcal{O}(\Delta t)$$

for $k \in \{0, ..., N-2\}$. (As in Lemma D.2) such $\Lambda_k$ exists without stepwise-Lipschitz assumption under mild regularity of the score function class and constraints.

For the prior on pair chain setup, we let $\bar{p}_{N-1}^\theta$ be a terminal pair prior on $(X_{N-1}, X_N)$ induced by sampling $X_N \sim p_N, P_N \sim \Pi(X_N)\zeta$ with $\zeta \sim \mathcal{N}(0, I_d)$ so that $X_{N-1} \sim \bar{T}_N^\theta(\cdot|X_N, P_N)$. Then, we propagate backward by

$$\bar{p}_0^\theta := \bar{p}_{N-1}\bar{T}_{N-1}^\theta \cdots \bar{T}_1^\theta, \quad p_0^\theta := (\pi_1)_\# \bar{p}_0^\theta$$

where $\pi_1(x_0, x_1) = x_0$ is the projection map onto first coordinate. Since $\pi_1$ is 1-Lipschitz, $W_2(q_0, p_0^\theta) \le W_2(\bar{q}_0, \bar{p}_0^\theta)$ holds and, from the triangle inequality for $W_2$, we have

$$W_2(q_0, p_0^\theta) \le W_2(\bar{q}_0, \bar{p}_0^\theta) \le W_2(\bar{q}_0, \bar{\sigma}_{N-1}) + W_2(\bar{\sigma}_{N-1}, \bar{p}_0^\theta)$$

$$\le \sum_{k=0}^{N-2} W_2(\bar{\sigma}_k, \bar{\sigma}_{k+1}) + W_2(\bar{\sigma}_{N-1}, \bar{p}_0^\theta)$$

$$\le \sum_{k=0}^{N-2} \bar{\Lambda}_k W_2(\bar{q}_k, \bar{q}_{k+1}\bar{T}_{k+1}^\theta) + \bar{\Lambda}_N W_2(\bar{q}_{N-1}, \bar{p}_{N-1}^\theta) + \mathcal{O}(\Delta t).$$

Because in pair conditionals the second coordinate is a Dirac mass, the inner $W_2$ reduces to a position-only conditional mismatch:

$$W_2(\bar{q}_k, \bar{q}_{k+1}\bar{T}_{k+1}^\theta) = \mathbb{E}_{(x_{k+1}, x_{k+2}) \sim \bar{q}_{k+1}} \left[ W_2\left( q_{k|k+1}(\cdot|x_{k+1}, x_{k+2}), \bar{T}_{k+1}^\theta(\cdot|x_{k+1}, x_{k+2}) \right] \right.$$

Also, we note that that the following decomposition holds by triangle inequality:

$$W_2(\bar{q}_{N-1}, \bar{p}_{N-1}^\theta) \le W_2\left((q_N \otimes \rho_N)S_N, (p_N \otimes \rho_N)S_N\right) + W_2\left((p_N \otimes \rho_N)S_N, (p_N \otimes \rho_N)S_N^\theta\right)$$

where $p_N$ is the prior of position, $\rho_N(\cdot|X_N)$ is the prior of momentum defined by the law of $\Pi(x_N)\zeta$ with $\zeta \sim \mathcal{N}(0, I)$, $x_N \sim p_N$, and each $S_N$ and $S_N^\theta$ are defined by

$$S_N(x_N, p_N) := \bar{q}_{N-1|N}(\cdot|x_N, p_N) \otimes \delta_{x_N}, \quad S_N^\theta(x_N, p_N) := \bar{T}_N^\theta(\cdot|x_N, p_N) \otimes \delta_{x_N}$$

And, we observe that the first term is independent of $\theta$, and the second term is given by:

$$W_2\left((p_N \otimes \rho_N)S_N, (p_N \otimes \rho_N)S_N^\theta\right) = \mathbb{E}_{(x_N, p_N) \sim p_N \otimes \rho_N} \left[ W_2^2\left(\bar{q}_{N-1|N}(\cdot|x_N, p_N), \bar{T}_N^\theta(\cdot|x_N, p_N)\right) \right].$$

because the second coordinate is a Dirac delta. Therefore, we have the following bound:

$$W_2(q_0, p_0^\theta) \le \sum_{k=0}^{N-2} \bar{\Lambda}_k \mathbb{E}_{(x_k, x_{k+1}) \sim \bar{q}_k} \left[ W_2\left(\bar{q}_{k|k+1}(\cdot|x_{k+1}, x_{k+2}), T_{k+1}^\theta(\cdot|x_{k+1}, x_{k+2})\right) \right]$$

$$+ \bar{\Lambda}_N \mathbb{E}_{(X_N, p_N) \sim p_N \otimes \rho_N} \left[ W_2^2\left(\bar{q}_{N-1|N}(\cdot|x_N, p_N), T_N^\theta(\cdot|x_N, p_N)\right) \right]$$

$$+ \bar{\Lambda}_N W_2\left((q_N \otimes \rho_N)S_N, (p_N \otimes \rho_N)S_N\right) + \mathcal{O}(\Delta t).$$

Now, we recall that $T_{k+1}^\theta(\cdot|x_{k+1}, x_{k+2}) = \mathcal{N}(\mu_{k+1}^u(x_{k+1}, x_{k+1}), \sigma_{k+1}^4 \Delta t^2 (1 - a_{k+1}^2)\Pi(x_{k+1}))$ and let $(\mu_{k|k+1, k+2}, \Sigma_{k|k+1, k+2})$ be the mean and covariance of true one-step backward conditional $\bar{q}_{k|k+1}$. Then, from the relation between Gelbrich distance and 2-Wasserstein distance, we have:

$$\mathbb{E}_{x_{k+1} \sim q_{k+1}, x_{k+2} \sim q_{k+2}} W_2(\bar{q}_{k|k+1}, T_{k+1}^\theta(\cdot|x_{k+1}, x_{k+2}))^2$$

$$= \mathbb{E}_{x_{k+1} \sim q_{k+1}, x_{k+2} \sim q_{k+2}} \|\mu_{k|k+1, k+2}(x_{k+1}, x_{k+2}) - \mu_{k+1}^u(x_{k+1}, x_{k+2})\|$$

$$+ \mathbb{E}_{x_{k+1} \sim q_{k+1}, x_{k+2} \sim q_{k+2}} \left[ \mathsf{Tr}\left(\Sigma_{k|k+1, k+2}(x_{k+1}, x_{k+2})\right) + \mathsf{Tr}\left(\sigma_{k+1}^4 \Delta t^2 (1 - a_{k+1}^2)\Pi(x_{k+1})\right) \right]$$

$$+ \mathbb{E}_{x_{k+1} \sim q_{k+1}, x_{k+2} \sim q_{k+2}} \Delta_{k+1}(x_{k+1}, x_{k+2}) - 2\sigma_{k+1}^4 \Delta t (1 - a_{k+1}^2)$$

$$\times \mathbb{E}_{x_{k+1} \sim q_{k+1}, x_{k+2} \sim q_{k+2}} \left[ \mathsf{Tr}\left( \left(\Sigma_{k|k+1, k+2}^{1/2}(x_{k+1}, x_{k+2})\Pi(x_{k+1})\Sigma_{k|k+1, k+2}^{1/2}(x_{k+1}, x_{k+2})\right)^{1/2} \right) \right]$$

$$= \mathbb{E}_{x_k \sim q_k, x_{k+1} \sim q_{k+1}, x_{k+2} \sim q_{k+2}} \|X_k - \mu_k^u(x_{k+1}, x_{k+2})\| + \sigma_{k+1}^4 \Delta t^2 (1 - a_{k+1}^2)\mathsf{Tr}\left(\Pi(x_{k+1})\right)$$

$$+ \mathbb{E}_{x_{k+1} \sim q_{k+1}, x_{k+2} \sim q_{k+2}} \Delta_{k+1}(x_{k+1}, x_{k+2})$$

$$- 2\mathbb{E}_{x_{k+1} \sim q_{k+1}, x_{k+2} \sim q_{k+2}} \left[ \mathsf{Tr}\left( \left(\Sigma_{k|k+1}^{1/2}(x_{k+1})\sigma_{k+1}^2 \Pi(x_{k+1})\Sigma_{k|k+1}^{1/2}(x_{k+1})\right)^{1/2} \right) \right]$$

where we used the law of total variance for the last equality, by observing that $x_k = \mu_{k|k+1,k+2}(x_{k+1}, x_{k+2}) + \epsilon \sim q_k$ with $\mathbb{E}[\epsilon|x_{k+1}, x_{k+2}] = 0$ and $\mathrm{Cov}(\epsilon|x_{k+1}, x_{k+2}) = \Sigma_{k|k+1,k+2}$. Similar to overdamped case, $\Delta_{k+1}(x_{k+1}, x_{k+2}) \geq 0$ is the distance gap between Gelbrich distance and 2-Wasserstein distance (independent of $\theta$) so that it becomes zero if the true conditional $x_k|x_{k+1}, x_{k+2}$ follows the Gaussian distribution as

$$x_k|x_{k+1}, x_{k+2} \sim \mathcal{N}(\mu_{k|k+1,k+2}(x_{k+1}, x_{k+2}), \Sigma_{k|k+1,k+2}(x_{k+1}, x_{k+2}))$$

**Training loss (Underdamped) from CWPM**   Because $\|x_k - \mu_k^u(x_{k+1}, x_{k+2})\|^2$ can be decomposed into

$$\|x_k - \mu_k^u(x_{k+1}, x_{k+2})\|^2 = \|\Pi(x_{k+1})(x_k - \mu_k^u(x_{k+1}, x_{k+2}))\|^2$$
$$+ \underbrace{\|(I - \Pi(x_{k+1}))(x_k - \mu_k^u(x_{k+1}, x_{k+2}))\|^2}_{\text{constant w.r.t } \theta}$$

where $\mu_k^u$ does not have $\theta$ dependency on normal (second) term. Therefore, by abusing notation to set $p_N = x_{N+1}$, the choice of training loss becomes

$$L_{\mathrm{CWPM}}^{\mathrm{under}}(\theta) = \mathbb{E}_{q(x_{0:N})}\mathbb{E}_{\rho_N(p_N|x_N)}\left[\sum_{k=0}^{N-1} \lambda(k)\|\Pi(x_{k+1})(x_k - \mu_k^u(x_{k+1}, x_{k+2}))\|^2\right]$$

$$= \mathbb{E}_{q(x_{0:N})}\mathbb{E}_{\rho_N(p_N|x_N)}\left[\sum_{k=0}^{N-1} \frac{\|\Pi(x_{k+1})(x_k - \mu_k^u(x_{k+1}, x_{k+2}))\|^2}{2\sigma_{k+1}^4\Delta t^2(1 - a_{k+1}^2)}\right]$$

with some training loss weight $\lambda(k)$. In our case, the training loss weight proportional to the inverse of variance can be chosen by $\lambda(k) = \frac{1}{2\sigma_{k+1}^4\Delta t^2(1 - a_{k+1}^2)}$ with proportional constant $1/2$. And, notably, this leads to the exactly the same training loss provided in DT-ELBO, (Lemma C.2) without the requirement $x_k \in \Sigma$.

**Lemma D.2** (Sufficient condition for $\Lambda_k < \infty$ – Underdamped). *Let the one-step landing backward update of ULLA (Equation Backward-ULLA) be*

$$\bar{F}_\theta^k(x_+, x_{++}, \zeta) = x_+ - \sigma_k^2\Delta t\Pi(x_+)\left[a_{k+1}\tilde{p} + b_\theta^k(x_+, \tilde{p})\right]$$
$$+ \sigma_k^2\Delta t\sqrt{1 - a_k^2}\Pi(x_+)\zeta + \sigma_k^2\Delta t\phi(x_+, \tilde{p})$$

*with pseudo-momentum*

$$\tilde{p}(x_+, x_{++}) := \Pi(x_+)\left(\frac{x_{++} - x_+}{\sigma_{k+2}^2\Delta t}\right) \in T_{x_+}\Sigma$$

*, the normal term $\phi(x_+, \tilde{p})$, and the drift term $b_\theta^k(x_+, \tilde{p})$. Assume the following regularity:*

1. *(Regularity of constraint functions)   There exists constants $c_\phi^0, c_\phi^1 < \infty$ such that for any sqaure-integrable $Y = (X, P)$,*

$$\mathbb{E}\|\phi(Y)\|^2 \leq c_\phi^0 + c_\phi^1\mathbb{E}\left(\|X\|^2 + \|P\|^2\right)$$

2. *(Regularity of function class)   There exists constant $L_g, C$ such that*

$$\|b_\theta^k(x, p) - b_\theta^k(x', p')\| \leq L_g\left(\|x - x'\| + \|p - p'\|\right)$$
$$\|b_\theta^k(x, p)\| \leq C\left(1 + \|x\| + \|p\|\right)$$

*Let $\bar{T}_k^\theta$ be the associated Markov kernel to $\bar{F}_\theta^k$. That is, $T_k^\theta(\cdot|x_+, x_{++}) = Law(\bar{F}_\theta^k(x_+, x_{++}, \zeta))$. Then, for any probability measures $\mu, \nu$ in $\mathbb{R}^d$, we have*

$$W_2(\mu\bar{T}_k^\theta, \nu\bar{T}_k^\theta) \leq K_k W_2(\mu, \nu) + \mathcal{O}\left(\Delta t\left[1 + \sqrt{\mathbb{E}_{Y\sim\mu}\|Y\|^2} + \sqrt{\mathbb{E}_{Y'\sim\nu}\|Y'\|^2}\right]\right)$$

*for some constant $K_k$. Also, if $\mu$ is given as $\mu = \rho \bar{T}_i^\theta \bar{T}_{i-1}^\theta \ldots \bar{T}_j^\theta$ for $i \geq j$ and some density $\rho$ independent of $\theta$, the supremum of second moment of $\mu$ is finite under $\theta \in \Theta$:*

$$\sup_{\theta \in \Theta} \mathbb{E}_{Y_\mu \sim \mu} \|Y_\mu\|^2 < \infty$$

*Combining these two, we can guarantee the existence of $\Lambda_k > 0$ such that*

$$W_2(\bar{\sigma}_{k-1}, \bar{\sigma}_k) \leq \Lambda_k W_2(\bar{q}_{k-1}, \bar{q}_k \bar{T}_k^\theta) + \mathcal{O}(\Delta t)$$

*for $k \in \{1, ..., N-1\}$ and*

$$W_2(\bar{\sigma}_{N-1}, \bar{p}_0^\theta) \leq \Lambda_N W_2(\bar{q}_{N-1}, \bar{p}_{N-1}^\theta \bar{T}_k^\theta) + \mathcal{O}(\Delta t) \tag{8}$$

*Proof.* Let $(X_+, X_{++}), (X'_+, X'_{++})$ be the synchronous coupling for $W_2(\mu, \nu)$ with shared noise $\zeta \sim (0, I)$. Write $\Delta_+ := X_+ - X'_+$ and $\Delta_{++} := X_{++} - X'_{++}$, and

$$\Delta := \bar{F}_\theta^k(X_+, X_{++}, \zeta) - \bar{F}_\theta^k(X'_+, X'_{++}, \zeta)$$

Write $\tilde{p} := \tilde{p}(X_+, X_{++}), \tilde{p}' := \tilde{p}(X'_+, X'_{++})$, and

$$\Delta b := b_\theta^k(X_+, \tilde{p}) - b_\theta^k(X'_+, \tilde{p}'), \quad b' := b_\theta^k(X'_+, \tilde{p}')$$

Then,

$$\Delta = \Delta_+ + \underbrace{\sigma_k^2 \Delta t \Pi(X_+) a_{k+1}(\tilde{p} - \tilde{p}')}_{(1)} + \underbrace{\sigma_k^2 \Delta t \Pi(X_+) \Delta b}_{(2)} + \underbrace{\sigma_k^2 \Delta t (\Pi(X_+) - \Pi(X'_+))(a_{k+1}\tilde{p}' + b')}_{(3)}$$

$$+ \underbrace{\sqrt{1 - a_k^2} \sigma_k^2 \Delta t (\Pi(X_+) - \Pi(x'_+))\zeta}_{(4)} + \underbrace{\sigma_k^2 \Delta t (\phi(X_+, \tilde{p}) - \phi(X'_+, \tilde{p}'))}_{(5)}$$

Now, note that (1) term is bounded by

$$\mathbb{E}\|(1)\|^2 \leq C_1 \left( \mathbb{E}\|X_{++} - X_+\|^2 + \mathbb{E}\|X'_{++} - X'_+\|^2 \right) = \mathcal{O}(\Delta t^2)$$

for some constant $C_1$ that depends on $k$. The term (2) is also can be bounded by

$$\mathbb{E}\|(2)\|^2 \leq C_2 \Delta t^2 \mathbb{E}\|\Delta_+\|^2 + C_2 \left( \mathbb{E}\|X_{++} - X_+\|^2 + \mathbb{E}\|X'_{++} - X'_+\|^2 \right) = C\Delta t^2 \mathbb{E}\|\Delta_+\|^2 + \mathcal{O}(\Delta t^2)$$

for some $C_2$, because $\|\Delta b\| \leq L_g (\|\Delta_+\| + \|\tilde{p} - \tilde{p}'\|)$ For term (3), it is bounded by

$$\mathbb{E}\|(3)\|^2 \leq 4\sigma_k^4 \Delta t^2 (a_{k+1}\|\tilde{p}'\|^2 + \|b'\|^2) \leq C_3 \Delta t^2 \left( 1 + \mathbb{E}\|X'_+\|^2 \right)$$

for some constant $C_3$. Similarly, term (4) can be bounded by:

$$\mathbb{E}\|(4)\|^2 \leq 4\sigma_k^4 \Delta t^2 (1 - a_k^2) \mathbb{E}\|\zeta\|^2 = \mathcal{O}(\Delta t^2)$$

and term (5) is bounded by:

$$\mathbb{E}\|(5)\|^2 \leq 2\sigma_k^4 \Delta t^2 (\mathbb{E}\|\phi(X_+, \tilde{p})\|^2 + \mathbb{E}\|\phi(X'_+, \tilde{p}')\|^2) \leq C_5 \Delta t^2 \left( 1 + \mathbb{E}\|X_+\|^2 + \mathbb{E}\|X'_+\|^2 \right)$$

By combining these terms, we recover that

$$\mathbb{E}\|\Delta\|^2 \leq (1 + C_6 \Delta t) \mathbb{E}\|\Delta_+\|^2 + C_6 \Delta t^2 (1 + \mathbb{E}_{Y \sim \mu}\|Y\|^2 + \mathbb{E}_{Y' \sim \nu}\|Y'\|^2)$$

Since $W_2(\mu, \nu)^2 \geq \mathbb{E}\left(\|\Delta_+\|^2 + \|\Delta\|_{++}^2\right) \geq \mathbb{E}\|\Delta_+\|^2$ holds, we have

$$W_2(\mu \bar{T}_k^\theta, \nu \bar{T}_k^\theta) \leq K_k W_2(\mu, \nu) + \mathcal{O}\left( \Delta t \left( 1 + \sqrt{\mathbb{E}_{Y \sim \mu}\|Y\|^2} + \sqrt{\mathbb{E}_{Y' \sim \nu}\|Y'\|^2} \right) \right)$$

for some constant $K_k > 0$. Now, similarly as in overdamped case, one can show that similar algebraic techniques gives the following under regularity assumptions:

$$\mathbb{E}\|\bar{F}_\theta^k(X_+, X_{++}, \zeta)\|^2 \leq (1 + a_k \Delta t + \mathcal{O}(\Delta t^2)) \mathbb{E}(\|X_+\|^2 + \|X_{++}\|^2) + b_k \Delta t + \mathcal{O}(\Delta t^2)$$

for some constant $a_k, b_k$, using $\mathbb{E}\|X_{++} - X_+\|^2 = \mathcal{O}(\Delta t)$ from our pair chain setup. Hence, the same logic as in the overdamped shows

$$\sup_{\theta \in \Theta} \mathbb{E}_{Y_\mu \sim \mu}^2 \|Y_\mu\|^2 < \infty$$

for $\mu$ given as $\mu = \rho \bar{T}_i^\theta \bar{T}_{i-1}^\theta \ldots \bar{T}_j^\theta$ for $i \geq j$ and some density $\rho$ independent of $\theta$. $\qquad \square$

# E EXPERIMENT SETTINGS AND SUPPLEMENTARY RESULTS

**Settings.** All experiments were implemented in Python using the PyTorch framework [54] and run in a Linux (Ubuntu) environment. The computational hardware was tailored to the specific experimental group. We utilized an NVIDIA L40S GPU with 45GB of VRAM for the Earth and climate science datasets, and an NVIDIA H100 GPU with 80GB of VRAM for the 3D mesh data experiments. All other tasks, including the $SO(10)$ manifold, Alanine dipeptide, and the 7-DOF robot arm, were conducted on an NVIDIA H200 GPU with 141GB of VRAM.

## E.1 DESCRIPTION OF BASELINE ALGORITHMS

**Riemannian Flow Matching (RFM).** RFM [26] is a framework for training Continuous Normalizing Flows (CNF) [55] on Riemannian manifold by regressing a vector field $v_t$ to a conditional target vector field $u_t(x|x_1)$ for $t \in [0,1]$ defined via a user-specified premetric $d(\cdot, \cdot)$ (e.g., geodesics, spectral distances). The model minimizes the Riemannian Conditional Flow Matching objective given as :

$$\mathcal{L}_{\text{RCFM}} = \mathbb{E}_{t \sim \mathcal{U}(0,1), x_1 \sim q_{\text{data}}, x_0 \sim p_{\text{prior}}} \left[ \| v_t(x_t) - u_t(x_t \mid x_1) \|_g^2 \right]$$

where $x_t$ is as conditional flow sample interpolation between prior samples $x_0$ and the data point $x_1$, and $\|\|_g$ is the norm defined in the corresponding Riemannian manifold.

The computational requirements for $x_t$ may depend on the manifold's geometry. On simple manifold (e.g., spheres, tori), the geodesic distance can be used as the premetric, allowing $x_t$ to be computed in closed form via the exponential map, thus making the algorithm simulation-free. In contrast, on general geometries (e.g., triangular meshes) where exact geodesics are intractable, spectral distances such as the biharmonic distance are employed as the premetric. In this case, computing $x_t$ requires solving an ODE during the training process.

**Remark 5** (Implementation details on RFM). For RFM, we used the default configuration from the official code from authors. For Earth & Climate datasets, training iterations were reduced to $1/10$, whereas Mesh data experiments were conducted using the unaltered default configuration.

**Riemannian Denoising Diffusion Probabilistic Models (RDDPM).** RDDPM [34] is a constrained diffusion model framework that adapts Denoising Diffusion Probabilistic Models (DDPMs) [1] to Riemannian manifold $\Sigma := \left\{ x \in \mathbb{R}^d \mid h(x) = 0 \right\}$ setup by incorporating a Newton's method projection step into the diffusion process.

The method constructs forward and backward Markov chains that alternate between diffusion steps along tangential direction of $\Sigma$ and projecting the resulting sample back onto $\Sigma$ via Newton's method. While this guarantees feasibility at every step, the iterative nature of the projection leads to higher computational costs and potentially result in forward trajectory resampling due to projection failures. We remark that the projected version of OLLA (OLLA-P) corresponds to RDDPM under the equality-only scenario.

**Euclidean Forward with Backward Variants.** These baselines employ a standard unconstrained Euclidean diffusion process for the forward process, and distinguish themselves by the mechanism used to enforce constraints $h(x) = 0, g(x) \le 0$ during the backward process. In the forward process, it follows the update rule below:

$$x_{k+1} = x_k - \frac{\sigma_k^2 \Delta t}{2} \nabla f(x_k) + \sigma_k \sqrt{\Delta t} \zeta_k$$

with corresponding training loss

$$\mathcal{L}_{\text{Euclidean}}^{\text{over}}(\theta) = \mathbb{E}_{q(x_{0:N})} \left[ \sum_{k=0}^{N-1} \frac{\| x_k - \mu_{k+1}^o(x_{k+1}) \|^2}{2\sigma_{k+1}^2 \Delta t} \right]$$

and $\mu_{k+1}^o := x_{k+1} + \frac{\sigma_{k+1}^2 \Delta t}{2} \left[ \nabla f(x_{k+1}) + s_\theta^{k+1}(x_{k+1}) \right]$.

1. **Euclidean**: This method performs sampling using the standard Euclidean backward without any constraint enforcement. The backward update rule is given as:

$$x_k = x_{k+1} + \frac{\sigma_{k+1}^2 \Delta t}{2} \left[ \nabla f(x_{k+1}) + s_{k+1}^\theta(x_{k+1}) \right] + \sigma_{k+1} \sqrt{\Delta t} \zeta_{k+1}$$

This approach offers no guarantee that the generated samples lie on $\Sigma$.

2. **Projected**: This variant strictly enforces equality constraints by projecting the sample onto $\Sigma$ immediately after each Euclidean backward step. Let $\tilde{x}_k$ be the proposal from the Euclidean backward step. Then, the final state is obtained via $x_k = \mathcal{P}_\Sigma(\tilde{x}_k)$, where $\mathcal{P}_\Sigma$ finds the root of $h(y) = 0, g(y) \leq 0$ close to $\tilde{x}_k$ using the interior point method [56, 57]. We remark that our implementation uses log-barrier for $g(x) < 0$ and quadratic penalty for $g(x) \geq 0$.

3. **Lagrangian**: This method formulates the sampling step as a constrained optimization problem using the Augmented Lagrangian Method (ALM). At each timestep, the proposal $\tilde{x}_k$ is refined by minimizing an augmented Lagrangian objective:

$$\mathcal{L}(x, \lambda, \mu) = \lambda^T h(x) + \frac{\rho}{2}\|h(x)\|^2 + \frac{1}{2\rho}\left(\|\text{ReLU}(\mu + \rho g(x))\|^2 - \|\mu\|^2\right)$$

The inequality term follows the Powell-Hestenes-Rockafellar (PHR) formulation. This specific form is derived by introducing a non-negative slack variable $s \geq 0$ to convert the inequality constraint $g(x) \leq 0$ into an equality $g(x) + s = 0$. By constructing the standard augmented Lagrangian for this equality and analytically minimizing it with respect to $s$, the slack variable is eliminated, resulting in the closed-form $\text{ReLU}(\mu + \rho g(x))$ term. This ensures that penalties are applied correctly only when constraints are violated or multipliers are active. The multipliers $\lambda$ and $\mu$ are updated iteratively via dual ascent. We note that this approach is also introduced in Liang et al. [58].

4. **Guided**: This approach utilizes constraint guidance during sampling. The standard drift term of the backward process is modified by adding a guidance term derived from the gradient of a constraint violation energy potential. This potential is defined as $V(x) = \frac{1}{2}\|h(x)\|^2 + \frac{1}{2}\|\text{ReLU}(g(x))\|^2$, where the first term penalizes deviations from equality constraints and the second term penalizes violations of inequality constraints. Consequently, the backward update rule naturally incorporates a gradient descent step on this potential, which steers the generated trajectory towards the feasible set $\Sigma$ by actively minimizing the constraint violation at each step.

### E.2 EXPERIMENT SETTINGS AND DESCRIPTIONS

**Earth and Climate Science Datasets** $S^2$. This benchmark [59, 60, 22, 61, 62] evaluates the model's ability to learn geographical distributions on the Earth's surface, which is modeled as the 2-sphere, $S^2$. The datasets represent the locations of phenomena such as volcanoes, earthquakes, floods, and fires.

*Mathematical formulation.* A sample $x$ represents a point in 3D Euclidean space lying on the surface of a unit sphere. Thus, $x \in \mathbb{R}^d$ with $d = 3$. The manifold is defined by a single, simple equality constraint $h(x) = \|x\|_2 - 1 = 0$.

*Prior distribution.* As this is a compact manifold, the prior distribution $p_N$ is set to be the uniform distribution over the surface of the sphere $S^2$.

**3D Mesh Data on a Learned Manifolds.** The objective is to learn a probability distribution over the surface of a complex 3D shape, such as the Stanford Bunny [63] and Spot the Cow [64]. The manifold is implicitly defined as the zero-level set of a Signed Distance Function (SDF) that is itself represented by a pre-trained neural network $h_{NN}(x)$ as performed in Rozen et al. [23], Gropp et al. [65].

*Mathematical Formulation.* A sample $x$ represents a point in 3D Euclidean space, thus $x \in \mathbb{R}^d$ with $d = 3$. The manifold is defined by a single equality constraint requiring any valid point to lie on the zero-level set of $h_{NN}(x) = 0$.

*Prior distribution.* The prior distribution $p_N$ is chosen to be uniform distribution over the learned manifold surface $\Sigma$ due to its compactness.

**High-Dimensional Special Orthogonal Group ($SO(10)$).** This experiment tests the model's ability to learn a multimodal distribution on the high-dimensional Lie group $SO(10)$. This is a challenging task due to the high dimensionality and non-trivial geometric structure of the manifold.

*Mathematical Formulation.* A sample is a $10 \times 10$ matrix, which is vectorized into $x \in \mathbb{R}^{100}$. The constraints enforce the defining properties of a special orthogonal matrix. For the equality

constraints, we impose

$$h_{ij}(X) = (X^T X - I)_{ij} = 0 \quad \text{for } 1 \le i \le j \le 10$$

and the determinant condition $\det(X) = 1$ is handled by via rejection when it is violated.

*Prior distribution.* Similarly, the manifold is compact and we choose uniform distribution over $SO(10)$ as our prior distribution.

**Alanine Dipeptide** This task involves generating valid 3D conformations of Alanine dipeptide, a model system in biophysics. The goal is to learn the distribution of structures subject to constraints on specific internal coordinates, including a mixed equality and inequality setup. Following the same approach in Liu et al. [34], we generated the dataset by running a 1ns constrained molecular dynamics simulation of alanine dipeptide in water using GROMACS [66] with a 1 fs timestep. A harmonic bias was applied through the COLVARS module [67], where the chosen collective variable was dihedral angle $\phi$. The harmonic restraint was centered at $\phi = -70°$ with a force constant 5.0. Other simulation settings follow closely those reported in Lelièvre et al. [68]. In total, $10^4$ configurations were collected by saving a snapshot every 100 simulation steps. Hydrogen atoms were removed, leaving the coordinates of the 10 heavy atoms for further analysis.

*Mathematical Formulation.* The state $x$ consists of the 3D coordinates of the 10 non-hydrogen atoms, so $x \in \mathbb{R}^{30}$. The constraints are placed on two of the molecule's primary dihedral angles, $\phi$ and $\psi$. For the equality constraints, the dihedral angle $\phi$ is fixed to a specific value:

$$h(x) = \phi(x) - (-70°)_{\text{rad}} = 0$$

and we impose an inequality constraint so that another adjacent dihedral angle $\psi$ is constrained to lie within the range $[130°, 170°]$. This is formulated as a single inequality:

$$g(x) = \max \left\{ \psi(x) - 170°_{\text{rad}}, 130°_{\text{rad}} - \psi(x) \right\} \le 0.$$

*Prior distribution.* Instead of introducing a potential-based drift term to induce a specific unimodal prior as in Liu et al. [34], we employ an empirical prior strategy. We first generate a large set of forward trajectories using the corresponding constrained dynamics (OLLA/ULLA) by running them to approximate the terminal prior distribution $q_N$ on the feasible set. The terminal states $x_N$ of these trajectories are collected, and the backward sampling process is initiated by drawing starting points uniformly from this pre-computed set, serving as a discrete approximation of the prior. Furthermore, to ensure the generated conformations respect physical symmetries, the score network for this task is designed to be $SE(3)$-invariant as proposed in Liu et al. [34].

**7-DOF Robot Arm Trajectory** This experiment focuses on learning a complex, bimodal distribution of trajectories for a 7-DOF Franka Emika Panda robot arm. The model is trained on a dataset of 400 valid paths (200 for S-shaped, 200 for reverse S-shaped paths) generated by the Rapidly-exploring Random Tree (RRT) algorithm. The primary task is to generate trajectories that trace both S-shaped and reverse S-shaped paths between fixed start and end points. Throughout the motion, the generated trajectories must satisfy several critical constraints: the robot arm must navigate around two spherical obstacles, and its end-effector must maintain a constant height of $z = z_{\text{target}} = 0.1$.

*Mathematical Formulation.* The fundamental state of the robot arm is its configuration in joint space, represented by a vector of 7 joint angles, $\theta \in \mathbb{R}^7$. A trajectory is a time-discretized sequence of these configurations, $(\theta_l)_{l=1}^L$. To avoid the periodicity issue of raw angles, which poses challenges for neural networks, we represent each joint angle $\theta_{l,j}$ as a 2D vector on the unit circle $(\cos(\theta_{l,j}), \sin(\theta_{l,j}))$. Consequently, the state at a single time step $l$ is a vector $x_l \in \mathbb{R}^{14}$. The full trajectory is flattened into a single vector $x = [x_1, \ldots, x_L] \in \mathbb{R}^d$. For a trajectory with $L \in \{10, 20, 30, 40\}$ as in our setup, the ambient space dimension is $d \in \{140, 280, 420, 560\}$.

The constraints on the robot's behavior, such as end-effector position and obstacle avoidance, are defined in 3D Cartesian space. We bridge the joint space representation and the Cartesian space constraints using the forward kinematics function, $\text{FK} : \mathbb{R}^7 \to \mathbb{R}^{3 \times K}$, which maps a set of joint angles $\theta_l$ to the 3D positions of the $K = 7$ links of the robot arm. To handle the large number of resulting constraints efficiently, we employ a "summation trick" to combine multiple constraint violations into a single function for both equalities and inequalities. In particular, multiple geometric and kinematic conditions are aggregated into a single sum-of-squares function:

$$h(x) = \sum_{i=1}^m h_i(x)^2 = 0.$$

The individual components $h_i(x)$ enforce: (1) the validity of the joint representation, $h_{\text{rep}}(x_{l,j}) = \cos^2(\theta_{l,j}) + \sin^2(\theta_{l,j}) - 1 = 0$, for each joint $j$ at each time step $l$, (2) fixed start and end points for the trajectory

$$h_{\text{end}}(x_L) = \|\text{FK}(\theta_L)_{\text{end-effector}} - p_{\text{end}}\|^2 = 0, \quad h_{\text{start}}(x_1) = \|\text{FK}(\theta_0)_{\text{end-effector}} - p_{\text{start}}\|^2 = 0$$

with $p_{\text{start}}$ and $p_{\text{end}}$ being the target start and end positions, and (3) a fixed $z$-height for the end effector throughout the trajectory, $h_z(x_l) = [\text{FK}(\theta_l)_{\text{end-effector}}]_z - z_{\text{target}} = 0$.

For the inequality constraint, the robot arm must avoid two spherical obstacles. For each relevant robot link $k \in [K]$ and obstacle $o \in \{1, 2\} := N_{\text{obs}}$, the distance between them must exceed a safety margin. These conditions are combined into a single function by summing the rectified violations:

$$g(x) = \sum_{l=1}^{L} \sum_{k=1}^{K} \sum_{o=1}^{N_{\text{obs}}} \text{ReLU}\left((r_{\text{obs},o} + r_{\text{safety}}) - \|\text{FK}(\theta_l)_{\text{link},k} - p_{\text{obs},o}\|\right) \leq 0.$$

with $r_{\text{obs},o}, p_{\text{obs},o}$ being the radius and position of obstacles. This function is non-positive if and only if all links maintain the required minimum distance $r_{\text{safety}}$ from all obstacles throughout the entire trajectory.

*Prior distribution.* Similar to the Alanine Dipeptide task, we employ an empirical prior strategy. We first generate a large set of forward trajectories using the corresponding constrained dynamics (OLLA/ULLA) by running them to approximate the target prior distribution $q_N$ on the feasible set. The terminal states $x_N$ of these trajectories are collected, and the backward sampling process is initiated by drawing starting points uniformly from this pre-computed set, serving as a discrete approximation of the prior.

Table 6: **Summary of constrained feasible set dimensions and constraint specifications.** Below table represent the ambient dimension $d$, the intrinsic manifold dimension, and the number of equality ($m$) and inequality ($l$) constraints. For the Robot Arm task, $L$ denotes the number of time steps (e.g., $L \in \{10, \ldots, 40\}$), and the constraint counts $m$ and $l$ are reported before applying the summation trick.

| Dataset / Task | Ambient Dim. ($d$) | Intrinsic Dim. | Equality ($m$) | Inequality ($l$) |
|---|---|---|---|---|
| Earth & Climate ($S^2$) | 3 | 2 | 1 | 0 |
| 3D Mesh (Bunny / Spot) | 3 | 2 | 1 | 0 |
| Lie Group $SO(10)$ | 100 | 45 | 55 | 0 |
| Alanine Dipeptide | 30 | 29 | 1 | 2 |
| 7-DOF Robot Arm | $14L$ | $7L$ | $8L + 2$ | $14L$ |

Table 7: Detailed hyperparameters for all datasets, specified per algorithm.

| Dataset | Algorithm | $\gamma$ | $\sigma_{\min}$ | $\sigma_{\max}$ | $N$ | $T$ | $l_f$ | $N_{epoch}$ | $B$ | $N_{node}$ | $N_{layer}$ | $\alpha$ | $\epsilon$ |
|---|---|---|---|---|---|---|---|---|---|---|---|---|---|
| | OLLA | - | 0.01 | 1.0 | 100 | 4.0 | 1 | 20000 | 128 | 512 | 5 | 50 | - |
| Volcano | ULLA | 3 | 0.1 | 1.3 | 50 | 2.0 | 1 | 20000 | 128 | 512 | 5 | 50 | - |
| | ULLA-P | 3 | 0.1 | 1.3 | 50 | 2.0 | 1 | 20000 | 128 | 512 | 5 | 50 | - |
| | OLLA | - | 0.01 | 1.0 | 100 | 4.0 | 1 | 20000 | 512 | 512 | 5 | 50 | - |
| Earthquake | ULLA | 3 | 0.1 | 1.3 | 50 | 2.0 | 1 | 20000 | 512 | 512 | 5 | 50 | - |
| | ULLA-P | 3 | 0.1 | 1.3 | 50 | 2.0 | 1 | 20000 | 512 | 512 | 5 | 50 | - |
| | OLLA | - | 0.01 | 1.0 | 100 | 4.0 | 1 | 20000 | 512 | 512 | 5 | 50 | - |
| Flood | ULLA | 3 | 0.1 | 1.3 | 50 | 2.0 | 1 | 20000 | 512 | 512 | 5 | 50 | - |
| | ULLA-P | 3 | 0.1 | 1.3 | 50 | 2.0 | 1 | 20000 | 512 | 512 | 5 | 50 | - |
| | OLLA | - | 0.01 | 1.0 | 100 | 4.0 | 1 | 20000 | 512 | 512 | 5 | 50 | - |
| Fire | ULLA | 3 | 0.1 | 1.3 | 50 | 2.0 | 1 | 20000 | 512 | 512 | 5 | 50 | - |
| | ULLA-P | 3 | 0.1 | 1.3 | 50 | 2.0 | 1 | 20000 | 512 | 512 | 5 | 50 | - |
| Bunny $(k=50)$ | OLLA | - | 0.07 | 0.07 | 100 | 8.0 | 100 | 2000 | 2048 | 256 | 5 | 25 | - |
| | ULLA | 20 | 0.2 | 0.6 | 30 | 3.0 | 100 | 2000 | 2048 | 256 | 5 | 25 | - |
| | ULLA-P | 20 | 0.2 | 0.6 | 50 | 3.0 | 100 | 2000 | 2048 | 256 | 5 | 25 | - |
| Bunny $(k=100)$ | OLLA | - | 0.07 | 0.07 | 100 | 5.0 | 100 | 2000 | 2048 | 256 | 5 | 25 | - |
| | ULLA | 20 | 0.2 | 0.6 | 30 | 3.0 | 100 | 2000 | 2048 | 256 | 5 | 25 | - |
| | ULLA-P | 20 | 0.2 | 0.6 | 50 | 3.0 | 100 | 2000 | 2048 | 256 | 5 | 25 | - |
| Spot $(k=50)$ | OLLA | - | 0.1 | 0.1 | 100 | 5.0 | 100 | 2000 | 2048 | 256 | 5 | 25 | - |
| | ULLA | 20 | 0.2 | 0.5 | 30 | 3.0 | 100 | 2000 | 2048 | 256 | 5 | 25 | - |
| | ULLA-P | 20 | 0.2 | 0.5 | 50 | 3.0 | 100 | 2000 | 2048 | 256 | 5 | 25 | - |
| Spot $(k=100)$ | OLLA | - | 0.1 | 0.1 | 100 | 3.0 | 100 | 2000 | 2048 | 256 | 5 | 25 | - |
| | ULLA | 20 | 0.2 | 0.5 | 30 | 3.0 | 100 | 2000 | 2048 | 256 | 5 | 25 | - |
| | ULLA-P | 20 | 0.2 | 0.5 | 50 | 3.0 | 100 | 2000 | 2048 | 256 | 5 | 25 | - |
| SO(10) $(m=3)$ | OLLA | - | 0.2 | 2.0 | 100 | 1.0 | 100 | 2000 | 512 | 512 | 3 | 50 | - |
| | ULLA | 50 | 0.3 | 2.2 | 50 | 1.0 | 50 | 2000 | 512 | 512 | 3 | 5 | - |
| | ULLA-P | 50 | 0.3 | 2.2 | 50 | 1.0 | 50 | 2000 | 512 | 512 | 3 | 5 | - |
| SO(10) $(m=5)$ | OLLA | - | 0.2 | 2.0 | 100 | 1.0 | 100 | 2000 | 512 | 512 | 3 | 50 | - |
| | ULLA | 50 | 0.3 | 2.2 | 50 | 1.0 | 50 | 2000 | 512 | 512 | 3 | 5 | - |
| | ULLA-P | 50 | 0.3 | 2.2 | 50 | 1.0 | 50 | 2000 | 512 | 512 | 3 | 5 | - |
| Alanine dipeptide | ULLA-P | 10 | 0.5 | 2.0 | 100 | 0.2 | 10 | 2000 | 512 | 1024 | 5 | 50 | 0.05 |
| 7-DOF robot arm | ULLA-P | 1 | 0.1 | 2.0 | 100 | 0.2 | 50 | 10000 | 160 | 1024 | 5 | 200 | 0.001 |

### E.3 GENERATED SAMPLES FROM TASKS.

In this subsection, we demonstrate the effectiveness of our proposed methods by comparing the generated samples in various tasks to the baseline RDDPM [34].

**Earth and Climate Science Datasets $S^2$ –Volcano.**

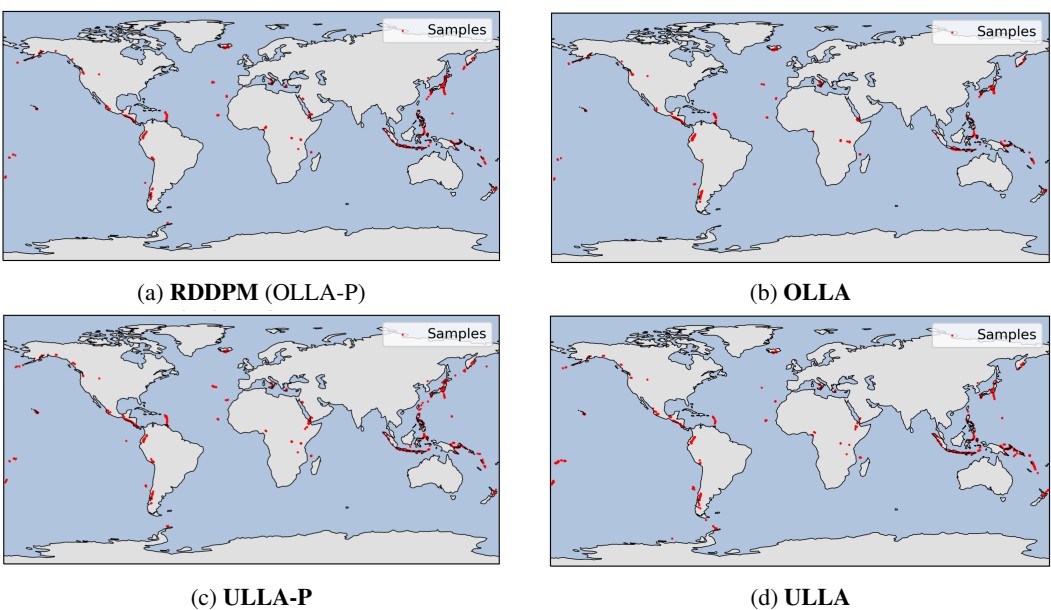

(a) **RDDPM** (OLLA-P)  (b) **OLLA**

(c) **ULLA-P**  (d) **ULLA**

Figure 5: Comparison of generated distributions across different algorithms-Volcano dataset

**3D Mesh data on learned manifold – Spot the Cow $(k = 100)$.**

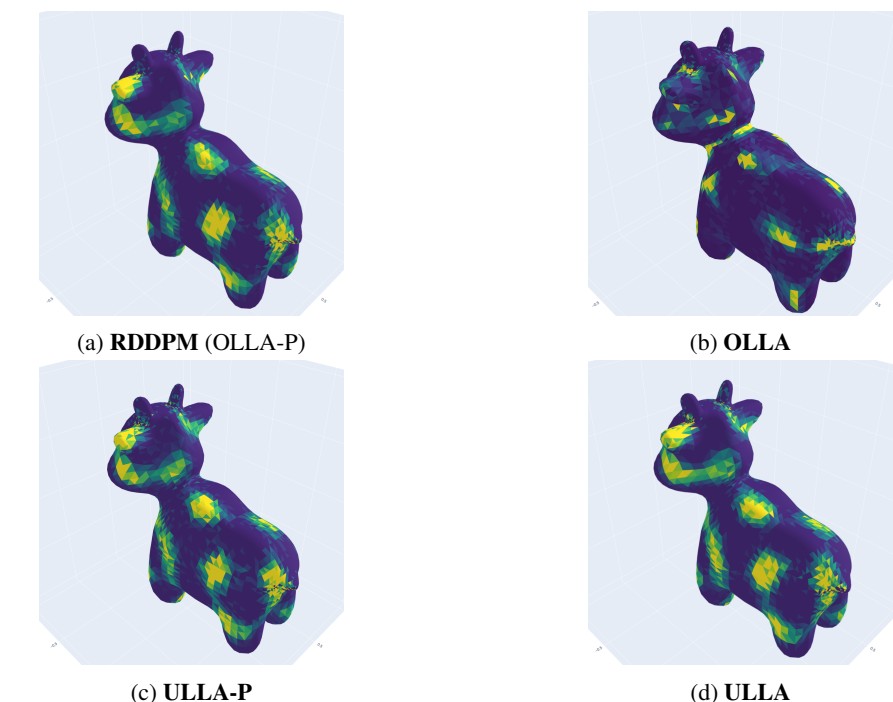

(a) **RDDPM** (OLLA-P)  (b) **OLLA**

(c) **ULLA-P**  (d) **ULLA**

Figure 6: Comparison of generated distributions across different algorithms- Spot the Cow $k = 100$

**SO(10) manifold with** $m = 3$**.**

(a) **RDDPM** (OLLA-P)

(b) **OLLA**

(c) **ULLA-P**

(d) **ULLA**

Figure 7: Comparison of generated distributions across different algorithms- $SO(10)$ with $m = 3$

### E.4 SUPPLEMENTARY RESULTS - EFFECT OF HYPERPARAMETERS $\alpha$ AND $\epsilon$.

Table 8: Effect of hyperparameters $\alpha$ (with fixed $\epsilon = 0.05$) and $\epsilon$ (with fixed $\alpha = 50$) on JSD metrics and constraint violations for Alanine Dipeptide using ULLA without last step projection.

| Parameter | JSD ($\psi$ angle) | JSD (RMSD) | $\mathbb{E}[|h(x)|]$ | $\mathbb{E}[g(x)^+]$ |
|---|---|---|---|---|
| *Effect of $\alpha$ (with $\epsilon = 0.05$)* | | | | |
| $\alpha = 0.1$ | $0.419_{\pm.000}$ | $0.045_{\pm.002}$ | $5.40 \times 10^{-3}$ | $3.79 \times 10^{-3}$ |
| $\alpha = 0.5$ | $0.247_{\pm.000}$ | $0.036_{\pm.001}$ | $5.39 \times 10^{-3}$ | $2.82 \times 10^{-3}$ |
| $\alpha = 1.0$ | $0.134_{\pm.004}$ | $0.034_{\pm.003}$ | $5.51 \times 10^{-3}$ | $1.60 \times 10^{-3}$ |
| $\alpha = 5.0$ | $0.060_{\pm.002}$ | $0.033_{\pm.001}$ | $2.97 \times 10^{-3}$ | $3.42 \times 10^{-4}$ |
| $\alpha = 10.0$ | $0.053_{\pm.004}$ | $0.034_{\pm.002}$ | $1.02 \times 10^{-3}$ | $1.46 \times 10^{-4}$ |
| $\alpha = 20.0$ | $0.043_{\pm.001}$ | $0.035_{\pm.002}$ | $2.09 \times 10^{-4}$ | $4.28 \times 10^{-7}$ |
| $\alpha = 50.0$ | $0.033_{\pm.002}$ | $0.034_{\pm.002}$ | $7.20 \times 10^{-5}$ | $7.95 \times 10^{-8}$ |
| $\alpha = 100.0$ | $0.051_{\pm.003}$ | $0.035_{\pm.002}$ | $3.70 \times 10^{-5}$ | $6.92 \times 10^{-7}$ |
| $\alpha = 200.0$ | $0.059_{\pm.008}$ | $0.183_{\pm.066}$ | $2.73 \times 10^{-3}$ | $4.19 \times 10^{-3}$ |
| $\alpha = 400.0$ | $0.235_{\pm.001}$ | NaN | $6.74 \times 10^{-1}$ | $9.95 \times 10^{-1}$ |
| *Effect of $\epsilon$ (with $\alpha = 50$)* | | | | |
| $\epsilon = 0.001$ | $0.084_{\pm.001}$ | $0.036_{\pm.002}$ | $7.00 \times 10^{-5}$ | $3.49 \times 10^{-6}$ |
| $\epsilon = 0.005$ | $0.054_{\pm.003}$ | $0.033_{\pm.002}$ | $7.30 \times 10^{-5}$ | $3.58 \times 10^{-5}$ |
| $\epsilon = 0.01$ | $0.048_{\pm.002}$ | $0.032_{\pm.002}$ | $7.00 \times 10^{-5}$ | $5.49 \times 10^{-7}$ |
| $\epsilon = 0.05$ | $0.033_{\pm.002}$ | $0.034_{\pm.002}$ | $7.20 \times 10^{-5}$ | $7.95 \times 10^{-8}$ |
| $\epsilon = 0.1$ | $0.052_{\pm.007}$ | $0.035_{\pm.002}$ | $2.57 \times 10^{-4}$ | $5.26 \times 10^{-4}$ |
| $\epsilon = 0.5$ | $0.081_{\pm.009}$ | $0.738_{\pm.012}$ | $2.37 \times 10^{-1}$ | $1.98 \times 10^{-1}$ |
| $\epsilon = 1.0$ | $0.107_{\pm.024}$ | $0.788_{\pm.001}$ | $3.01 \times 10^{-1}$ | $1.95 \times 10^{-1}$ |
| $\epsilon = 5.0$ | $0.134_{\pm.002}$ | $0.796_{\pm.002}$ | $7.34 \times 10^{-1}$ | $2.97 \times 10^{-1}$ |
| $\epsilon = 10.0$ | $0.188_{\pm.003}$ | $0.796_{\pm.002}$ | $1.05 \times 10^{0}$ | $1.08 \times 10^{0}$ |

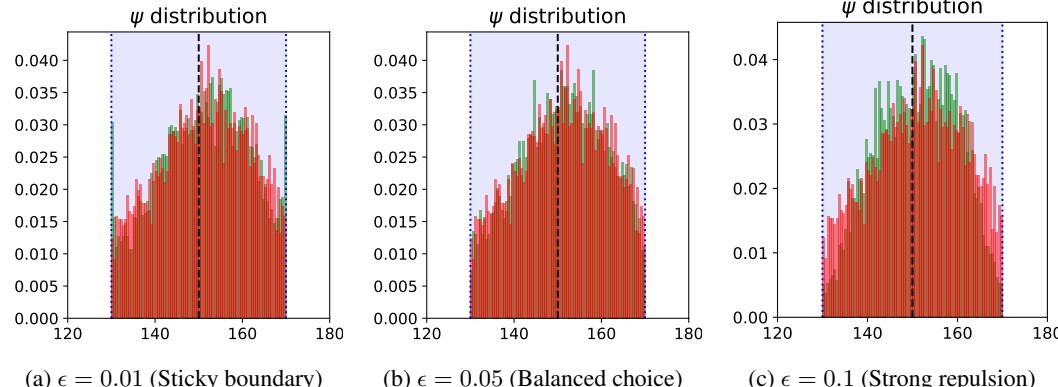

(a) $\epsilon = 0.01$ (Sticky boundary)     (b) $\epsilon = 0.05$ (Balanced choice)     (c) $\epsilon = 0.1$ (Strong repulsion)

Figure 8: **Effect of boundary repulsion rate $\epsilon$ on the generated distribution.** When $\epsilon$ is too small (a), trajectories tend to stick to the boundary. Conversely, an excessively large $\epsilon$ (c) aggressively pushes samples away from the boundary, distorting the distribution. A moderate choice (b) balances these effects, yielding the best sampling quality.

