# OpenReview forum: "Efficient Diffusion Models under Nonconvex Equality and Inequality constraints via Landing"
_ICLR.cc/2026/Conference — Submitted to ICLR 2026_

### Official Review · Reviewer_F33C · 2025-10-28

**Soundness:** 3
**Presentation:** 2
**Contribution:** 3
**Rating:** 4
**Confidence:** 3

**Summary:**

The paper presents a formulation to enforce constraints on diffusion models using the landing method. Diffusion models are viewed as a stochastic differential equation (SDE) and the "trajectory" of the SDE is refined at each time step to guide the generative processes to respect both equality and inequality constraints. While many of the previous works "project" the dynamic trajectory onto the constraint manifold, the paper argues that the projections are costly and not-always-well-defined. Instead, they introduce the landing approach, which arguably is more computationally efficient, to replace projections. The key mathematical idea behind this approach is to introduce an additional term (landing term) that enforces exponential decay of constraint violation in the SDE. Assuming that the gradients from these constraints are linearly independent, the landing term can be explicitly computed via simple matrix operations. This idea is tested on multiple benchmark examples, including both equality-constrained and mixed (equality + inequality-constrained) scenarios.

**Strengths:**

I find the landing term very interesting and novel, although I have to admit that I'm not fully familiar with the relevant literature. I do know some projection-based methods, so compared to those, as the authors argue, this landing approach sounds like an interesting, original idea. I also appreciate the fact that the landing term, given an assumption on the gradients of the constraint functions, can be further developed into an explicit solution. Overall, it is an interesting and potentially impactful idea for the research community.

**Weaknesses:**

Below are some potential weaknesses that the authors may want to help me understand better:
- I observe that the LICQ assumption (linearly independent gradients of the constraint functions) is a bit too strong but I'm not confident about this assessment (see my questions below). I would appreciate some further clarification on why this assumption is typically valid in many real-world problems. Also, some discussion about the cases where gradients are (almost) dependent (aka when the assumption breaks).
- The computation of the Jacobian and the (inverse of) Gram matrix sounds scary computationally. Would this be scalable to the cases where the dimension "d" is very large (e.g., generation of high-resolution micrographs of some material microstructures under some physical constraints)?
- The fact that the proposed method does not fully outperform projection-based methods may not be the critical weakness, assuming the mathematical idea is indeed new (to which I'll rely on other reviewers' assessment). However, it would still be valuable scientific knowledge to understand why that is the case. Is it because the additional term in the SDE adds more complexity in training optimization? Or that the constraint violation is penalized harder than the projection-based method? More analysis on the behavior of the proposed method, in comparison to other projection-based approaches, would make the paper a lot more interesting to read.

**Questions:**

- I'm just curious: is the LICQ assumption a valid assumption in the context of generative modeling? Using the generation of physics data as a hypothetical example, wouldn't there be a situation where gradients from different constraints may end up being collinear (linearly dependent) due to some sort of coupled effect? I can't think of a specific counterexample, though. Obviously, the authors have thought this through, so I would like their thoughts on this.
- Also, along the same line of thought, if the stacked Jacobian does not have a full rank or the matrix is ill-conditioned due to some "almost dependent" gradient vectors, how would the performance of the algorithm change?

---

> ### Author Response · Authors · 2025-11-27
> **Answer to Review by Reviewer F33C (1)**
>
> We thank the reviewer for the thoughtful and constructive feedback. Below, we leave itemized responses to address each of the comments, and we hope they can clarify your questions and concern.
>
> >Weakness (1): I observe that the LICQ assumption (linearly independent gradients of the constraint functions) is a bit too strong but I'm not confident about this assessment (see my questions below). I would appreciate some further clarification on why this assumption is typically valid in many real-world problems. Also, some discussion about the cases where gradients are (almost) dependent (aka when the assumption breaks).
> >
> >Question (1): I'm just curious: is the LICQ assumption a valid assumption in the context of generative modeling? Using the generation of physics data as a hypothetical example, wouldn't there be a situation where gradients from different constraints may end up being collinear (linearly dependent) due to some sort of coupled effect? I can't think of a specific counterexample, though. Obviously, the authors have thought this through, so I would like their thoughts on this.
> >
> >Question (2): Also, along the same line of thought, if the stacked Jacobian does not have a full rank or the matrix is ill-conditioned due to some "almost dependent" gradient vectors, how would the performance of the algorithm change?
>
> **Answer.** *[Related part: Remark 2 (Appendix A)]* Thank you for this great observation regarding LICQ. We fully agree that assuming LICQ is globally satisfied is indeed a strong condition that may not always hold in complex real-word scenarios.
>
> Prompted by your feedback, we have rigorously re-examined the theoretical analysis of our OLLA and ULLA SDE framework, and checked that the strict LICQ assumption is not necessary for the validity of our method. Instead, our framework remains well-defined under the much weaker relaxed Constant Rank Constraint Qualification (rCRCQ) [1]. We have updated the paper (Remark 2 in Appendix A) to reflect this theoretical refinement. Below, we provide a detailed clarification about this:
>
> **Justification of rCRCQ instead of LICQ.**
>
> Our original use of LICQ was to guarantee the invertibility of the Gram matrix $G(x)= \nabla J(x) \nabla J(x)^T$ so that the Lagrangian processes $(d\lambda_t, d\mu_t)$ satisfying the target landing property $dJ(X_t) = -\alpha \sigma(t)^2 J(X_t)dt$ (and momentum tangency property in ULLA) would be uniquely determined. However, for the SDEs themselves, what actually enters the dynamics are only the normal components $\nabla J(X_t)^T d\lambda_t$ and $\nabla J(X_t)^T d\mu_t$ . These normal terms remain unique even when $G(x)$ is singular, as long as the rank of $\nabla J(x)$ is locally constant. Under rCRCQ, this constant rank property holds in a neighborhood of $\Sigma$, which implies that the pseudo-inverse $G(x)^\dagger$ is smooth and that the projector $\Pi(x) = I -\nabla J(x)^T G(x)^\dagger \nabla J(x)$ is the orthogonal projector onto the tangent space of $\Sigma$. In the revised analysis, we therefore construct the landing term directly via $G(x)^\dagger$; the SDE coefficients (drift and diffusion) only depend on $\nabla J(x)^T G(x)^\dagger J(x)$ and $\Pi(x)$, both of which are well-defined and differentiable under rCRCQ, without requiring LICQ or full-rank $G(x)$.

---

> > ### Author Response · Authors · 2025-11-27
> > **Answer to Review by Reviewer F33C (2)**
> >
> > **On validity in generative modeling and (almost) dependent gradients.**
> >
> > In the context of generative modeling, it is natural that some constraints can be coupled and their gradients become colinear or nearly colinear. This is exactly our motivation to decide moving away from LICQ. Many constraints used in practice (polynomial, analytic, kinematic constraints, etc...) are smooth analytic functions of the state $x$. When $d \geq m + l$, for such constraints, the rank change of $\nabla J(x)$ occurs only on a lower-dimensional singular set, which has Lebesgue measure zero in $\mathbb{R}^d$ (from standard real analysis results). In our setting, the discretized algorithm (OLLA, ULLA) have discretization errors and they evolve on the entire ambient space $\mathbb{R}^d$. Since the singular set has Lebesgue measure zero, the probability that a randomly perturbed iterate lands exactly on such a set at some step is essentially zero. In other words, the discretized algorithms almost surely evolve in regions where rCRCQ holds.
> >
> > In summary, rCRCQ guarantees that the pseudo-inverse $G(x)^\dagger$ and the associated projector $\Pi(x) = I - \nabla J(x)^T G(x)^\dagger \nabla J(x)$ are continuous (and differentiable when $h, g$ are smooth), which is exactly the minimal regularity we need for the OLLA and ULLA SDE coefficients to be well-defined. So, from a theoretical standpoint, assuming rCRCQ (rather than LICQ) is sufficient to construct our landing dynamics and establish their target properties.
> >
> > In addition, beyond these theoretical considerations, our robotics experiments provide a concrete stress test of the method. The underlying kinematic and collision constraints are highly coupled and form a very complex equality/inequality system, yet the proposed landing-based samplers run stably in practice: they do not blow up, do not exhibit abrupt behavior, and produce almost feasible trajectories throughout sampling. This empirical robustness in a challenging robotics setup suggests that our OLLA and ULLA algorithms can reliably handle complicated constrained generative problems, even when the exact constraint qualification structure is difficult to verify in detail.
> >
> > >Weakness (2): The computation of the Jacobian and the (inverse of) Gram matrix sounds scary computationally. Would this be scalable to the cases where the dimension "d" is very large (e.g., generation of high-resolution micrographs of some material microstructures under some physical constraints)?
> >
> > **Answer.** Thank you for raising this important computational concern. In our method, the Gram matrix is $G(x) = \nabla J(x) \nabla J(x)^T \in \mathbb{R}^{(m+|I_x|) \times(m+|I_x|)}$ where $m$ is the number of equality constraints and $|I_x|\leq l$ is the number of active inequality constraints. So computing its pseudo-inverse $G(x)^\dagger$ costs $\mathcal{O}((m+|I_x|)^3)$, independent of the ambient dimension $d$. Thus, even for very high-dimensional data (e.g., high-resolution images), the computational cost for $G(x)^{\dagger}$ remains cheap as long as the number of (active) constraints is moderate.
> >
> > In regimes where the number of constraints grow with dimension (such as robotics), we additionally employ a simple "summation trick": instead of working with many constraints $\\{h_i \\}\_{i=1}^m, \\{g_j\\}\_{j=1}^l$,  we aggregate them into a small number of effective constraints as follows:
> > $$
> > h(x) = \sum_{i=1}^m h_i^2(x)=0, \quad g(x) = \sum_{j=1}^l \text{ReLU}(g_j(x)) \leq 0
> > $$
> > This reduces the size of $G(x)$ to a constraint (e.g., $1\times 1$ or $2\times 2$), making the computational cost for $G(x)^\dagger$ effectively constant-time with respect to both $d$ and the number of constraints. While gradient backpropagation (on the computational graph of PyTorch) still passes through all underlying constraint functions, this overhead is minor compared to the score network evaluation or training, and in practice we did not observe the Gram matrix-related operations to be a bottleneck in any of our experiments, including the robotics setting.

---

> > > ### Author Response · Authors · 2025-11-27
> > > **Answer to Review by Reviewer F33C (3)**
> > >
> > > >Weakness (3): The fact that the proposed method does not fully outperform projection-based methods may not be the critical weakness, assuming the mathematical idea is indeed new (to which I'll rely on other reviewers' assessment). However, it would still be valuable scientific knowledge to understand why that is the case. Is it because the additional term in the SDE adds more complexity in training optimization? Or that the constraint violation is penalized harder than the projection-based method? More analysis on the behavior of the proposed method, in comparison to other projection-based approaches, would make the paper a lot more interesting to read.
> > >
> > > **Answer.** *[Related parts: Section 5, Appendix E.4]*  Thank you for very encouraging comment. We agree that the original experiments did not fully clarify the advantages of our algorithms. In the revised version, we (i) added RFM on mesh data and Euclidean forward with backward variants on all experiments and (ii) more carefully tuned the hyperparameters for OLLA and ULLA (the previous version was under-tuned). With these changes, our landing-based samplers now match the projection-based methods on the main benchmarks and clearly outperform the Euclidean forward with backward variants. we refer to the updated section 5 for details.
> > >
> > > Conceptually, the gap between our landing-based method and projection-based method is explained by a clear trade-off. Projection-based approaches enforce feasibility almost exactly at every step (via Newton's method or interior point method), so the score network is trained on samples lying essentially on $\Sigma$, which yields a very accurate score function training when projections are stable. Our landing method instead enforces an exponential decay of constraint violation in the SDE; after discretization, intermediate states $\\{x_k\\}\_{k=1}^N$ can have small but nonzero violations, so the score is learned from points near $\Sigma$, introducing a small bias that can accumulate and slightly degrade final metrics such as JSD.
> > >
> > > This is consistent with our ablation on the landing rate $\alpha$ (Table 8 in Appendix E.4): when $\alpha$ is too small and violations are larger, performance degrades. On the other hand, projection-based methods pay a higher computational cost and can suffer from projection failures or numerical instability (especially in the robotics setting), while the landing-based sampler ULLA remained stable in all experiments. Our intention is therefore not to claim universal superiority over projection-based methods, but to offer an efficient and robust alternative that attains comparable performance while avoiding the projection bottleneck.
> > >
> > > ---
> > > We thank you again for your very helpful review and constructive feedback regarding our paper. We'd appreciate the opportunity to discuss further if you have any additional questions or suggestions.
> > >
> > > ### Reference
> > >
> > > [1] Leonid Minchenko and Sergey Stakhovski. On relaxed constant rank regularity condition in mathematical programming. Optimization, 60(4):429–440, 2011.

---

### Official Review · Reviewer_DVNP · 2025-10-31

**Soundness:** 3
**Presentation:** 3
**Contribution:** 2
**Rating:** 4
**Confidence:** 3

**Summary:**

This paper introduces landing-based constrained diffusion, an approach for generating from a constrained distributions. The method is proposed for general nonconvex constraint sets, employing a "landing" method which guides the sampling towards the feasible set. As opposed to prior approaches like projection-based sampling, the proposed approach relies on a Lagrangian form, characterizing the entire sampling process. The paper argues that such an approach is more computationally efficient, reporting better runtimes than Riemannian Denoising Diffusion Probabilistic Models.

**Strengths:**

- **Mathematical Framework:** The formal presentation of the method is interesting, especially the incorporation of the constraint term inside the forward/reverse SDE. The provided justifications for the approach are well presented, with sufficient theoretical support to justify the method.

- **Efficiency Improvement:** Constrained sampling approaches often suffer from much longer runtimes. While that is often permissible when constraint satisfaction is strictly required, the development of efficient constrained sampling procedures is likely of value is time-sensitive settings.

**Weaknesses:**

- **Limited Novelty:** The inclusion of Lagrangian updates within the diffusion sampling process is not a new concept. For example, [1-3] propose augmented-Lagrangian sampling scheme for the reverse diffusion. While the inclusion of Lagrangian updates in the forward SDE is of potential interest, and differentiates the work from these predecessors, it is difficult to evaluate the impact of this without comparison to existing methods (or even any discussion from the authors).

- **Experimental Evaluation:** The evaluation is conducted with a very limited set of baselines (*only Riemannian models*) , making it difficult to assess the actual performance of the method. Even compared to this single baseline, the approach is unable to provide a decisive edge, besides training and sampling. Additionally, the experimental settings really constitute toy examples, making it difficult to assess how this method would perform in the real-world.


---

[1] Liang, Jinhao, et al. "Simultaneous Multi-Robot Motion Planning with Projected Diffusion Models." arXiv preprint arXiv:2502.03607 (2025).

[2] Lee, Seungjun, and Shinjae Yoo. "Efficient Physics-Constrained Diffusion Models for Solving Inverse Problems."

[3] Blanke, Matthieu, et al. "Strictly Constrained Generative Modeling via Split Augmented Langevin Sampling." arXiv preprint arXiv:2505.18017 (2025).

**Questions:**

- How does this approach compare to simply incorporating a guidance term into the training / sampling process?

- Why is the sampling time omitted for RFM (Table 1)?

- Has any ablation been conducted on the effect of hyperparameters (e.g., landing rate)?

---

> ### Author Response · Authors · 2025-11-27
> **Answer to Review by Reviewer DVNP (1)**
>
> We appreciate the reviewer’s careful and helpful feedback. Below, we provide itemized responses to address each of the comments, and we hope they clarify our contributions and design choices.
>
> >Weakness (1): Limited Novelty: The inclusion of Lagrangian updates within the diffusion sampling process is not a new concept. For example, [1-3] propose augmented-Lagrangian sampling scheme for the reverse diffusion. While the inclusion of Lagrangian updates in the forward SDE is of potential interest, and differentiates the work from these predecessors, it is difficult to evaluate the impact of this without comparison to existing methods (or even any discussion from the authors).
> >
> >Question (1): How does this approach compare to simply incorporating a guidance term into the training / sampling process?
>
>
> **Answer.** *[Related part: Appendix E.1]* Thank you for raising this point about novelty and for prompting a more explicit comparison. We acknowledge that our initial submission did not fully clarify the behavior of our landing-based dynamic and how they differ from augmented-Lagrangian or guidance-based methods.
>
> At a high level, our method does use a Lagrangian formulation, but the way the multipliers are introduced and the role they play in side the SDE is, to our knowledge, fundamentally different from augmented-Lagrangian samplers and from standard guidance-based methods. Our goal in the sampling perspective is to design a landing-based constrained SDE whose invariant distribution is the exact constrained target and whose structure allows efficient sampling and stable constraint satisfaction.
>
> **Sampling perspective comparison with augmented-Lagrangian / guidance methods.**
>
> The approaches in [1] (augmented Lagrangian method) or [2], and typical guided samplers are essentially soft-penalty methods: they modify the sampling dynamics by adding constraint-dependent penalty or augmented-Lagrangian terms to the drift, so that constraint satisfaction is encouraged but not enforced exactly at the level of the underlying SDE. For any finite penalty parameter, the resulting sampler effectively targets a penalized version of the constrained distribution, and both the amount of constraint violation and the distortion of the stationary distribution depend sensitively on the chosen penalty or guidance parameters (even at continuous level).
>
> In contrast, our landing-based OLLA and ULLA dynamics are constructed so that, at the continuous-time level, the constraints obey landing properties
> $$
> dh(X_t) = -\alpha h(X_t)dt, \quad dg(X_t) = -\alpha(g(X_t) + \epsilon)dt
> $$
> and the normal component is controlled analytically via the Lagrangian processes $d\mu_t, d\lambda_t$ and the tangential projector $\Pi(x)$ is applied to both drift and diffusion. Under the assumption (rCRCQ) in our paper, the limiting landing-based SDE has the exact constrained target distribution as its stationary distribution (Proposition 1, 2), rather than a softened surrogate. Also, the target properties above explicitly govern the evolution of $h(X_t)$ and $g(X_t)$, so for a given $\alpha$, the process is systemically driven toward $\Sigma$ and violations remain small along the trajectory, while the target constrained stationary distribution is preserved because the landing term is derived from the constrained SDE itself rather than from a tunable penalty.

---

> > ### Author Response · Authors · 2025-11-27
> > **Answer to Review by Reviewer DVNP (2)**
> >
> > **Why we do not use Euclidean forward + landing backward.**
> >
> > A natural alternative in constrained diffusion model is to keep the forward SDE Euclidean and impose constraints only in the backward process (for example via guidance, augmented Lagrangian, or projections). While these Euclidean forward + constrained backward can indeed be attractive from a training perspective, since it is simulation-free on the forward side and can be computationally cheaper per iteration, We deliberately chose not to follow this route.
> >
> > The reason is that, in the standard diffusion framework, the backward SDE is designed as the time-reversal (in the Fokker-Planck sense) of a given forward SDE. If one modifies only the backward dynamics to enforce constraints, while keeping a Euclidean forward, then even with (i) a perfect score model, (ii) a perfectly matched prior, and (iii) no discretization error, there is no guarantee that the resulting backward process recovers the true constrained data distribution $q_0$ (Forward and backward are no longer an exact time-reversed pair).
> >
> > By instead constructing a constrained forward SDE with landing and then deriving its time-reversed backward dynamics (Appendix B.3), we ensure that (in the idealized continuous-time setting) the constrained backward process (with learned score function) is consistent with the constrained forward diffusion whose invariant measure is the desired target prior on $\Sigma$. Empirically, as shown in the revised Section 5, constrained-forward + constrained-backward methods (ours, RDDPM, RFM) outperform Euclidean-forward + constrained-backward baselines on constrained metrics, while our landing-based formulation additionally brings efficiency and stability benefits.
> >
> >
> > >Weakness (2): Experimental Evaluation: The evaluation is conducted with a very limited set of baselines (only Riemannian models) , making it difficult to assess the actual performance of the method. Even compared to this single baseline, the approach is unable to provide a decisive edge, besides training and sampling. Additionally, the experimental settings really constitute toy examples, making it difficult to assess how this method would perform in the real-world.
> >
> > **Answer.** *[Related part: Section 5, Appendix E]* Thank you for pointing out this important issue. We agree that the original experimental section did not provide a sufficiently broad set of baselines or a clear picture of how our method behaves.
> >
> > In the revised version, we have substantially expanded the evaluation:
> > * We added Euclidean forward + constrained backward variants as baselines on all tasks, and
> > * We re-tuned the hyperparameters of OLLA and ULLA (the original setting were indeed suboptimal), while keeping the Riemannian baselines at their best-performing settings.
> >
> > With these changes, OLLA/ULLA now match or closely track the projected-based Riemannian methods, and show a clear performance gap over Euclidean forward + backward variants in both constraint violation and sample quality (Section 5).
> >
> > Regarding the concern that our setting are toy examples, we would like to clarify the followings:
> > * **Alanine Dipeptide experiment.** The training data of alanine dipeptide are generated by a constrained molecular dynamics simulation in water for 1ns using GROMACS [4] with further post-processing, and it is widely used as a standard benchmark in computational chemistry (e.g., [5]). Our use of this system follows existing practice rather than an artificially simplified toy setup.
> > * **Robotics experiment.** The robot task is directly aligned with the setting used in [6], and we additionally impose equality constraints on top of collision and joint constraints. For a robot arm trajectory length $L=40$, the ambient dimension is $d=540$, with $m=322$ equality constraints and $l=560$ inequality constraints, making this a highly constrained, reasonably high-dimensional non-toy scenario.
> >
> > In both experiments (Table 4), ULLA significantly outperforms Euclidean forward + backward variants while maintaining very small constraint violations.
> >
> > Finally, regarding direct comparison to [1-3]: the methods in [1,2] are implemented as Lagrangian (augmented-Lagrangian schemes), while [3] (SAL) is algorithmically distinct but, to the best of our knowledge, does not have publicly available code. Our own attempts to implement [3] led to severe instability on both the robotics and dipeptide tasks, and we could not obtain a reliably working baseline. For this reason, we chose not to report potentially misleading numbers and instead discuss these methods in the related work.

---

> ### Author Response · Authors · 2025-11-27
> **Answer to Review by Reviewer DVNP (3)**
>
> >Question (2): Why is the sampling time omitted for RFM (Table 1)?
>
> **Answer.** *[Related part: Appendix E.1]* Thank you for pointing this out. Our notation there was indeed confusing. As explained in newly added Appendix E.1, for RFM on simple manifold where the exponential and logarithm maps are available in closed form (e.g., the sphere), the conditional flow sample $x_t$ can be computed analytically without an ODE solver, so the additional sampling cost from the manifold solver is effectively zero. We initially denoted this by "(-)", but we agree this can be misleading, so in the revised version we explicitly write "(0)" to indicate zero extra RFM sampling time in this case.
>
> >Question (3): Has any ablation been conducted on the effect of hyperparameters (e.g., landing rate)?
>
> **Answer.** *[Related part: Section 5, Appendix E.4]* We agree that the original submission lacked a careful hyperparameter ablation. In the revision, we added ablation studies for the landing rate $\alpha$ and boundary repulsion rate $\epsilon$ on the alanine dipeptide task with ULLA (Table 2-3, full results in Table 8 and Figure 8 in Appendix E.4), and discussed them in Section 5.
>
> In short, increasing $\alpha$ strengthens the drift toward $\Sigma$, substantially reducing equality violations and improving JSD; however, overly large $\alpha$ introduces large discretization error and eventually harms sample quality. For $\epsilon$, very small values cause trajectories to "stick" near the boundary, while very large values push them too aggressively into the interior of $\Sigma$, distorting the distribution and increasing inequality violation due to numerical instability. The ablation and visualization shows that a moderate choice of $\alpha$ and $\epsilon$ achieves the best trade-off between generation quality and constraint satisfaction.
>
> ---
> Thank you again for your insightful review and constructive feedback. We'd appreciate the opportunity to discuss further if you have any additional questions or suggestions.
>
> ### Reference
>
> [1] Liang, Jinhao, et al. "Simultaneous Multi-Robot Motion Planning with Projected Diffusion Models." arXiv preprint arXiv:2502.03607 (2025).
>
> [2] Lee, Seungjun, and Shinjae Yoo. "Efficient Physics-Constrained Diffusion Models for Solving Inverse Problems."
>
> [3] Blanke, Matthieu, et al. "Strictly Constrained Generative Modeling via Split Augmented Langevin Sampling." arXiv preprint arXiv:2505.18017 (2025).
>
> [4] Mark James Abraham, Teemu Murtola, Roland Schulz, Szilard Pall, Jeremy C Smith, Berk Hess, and Erik Lindahl. Gromacs: High performance molecular simulations through multi-level parallelism from laptops to supercomputers. SoftwareX, 1:19–25, 2015.
>
> [5] Tony Lelievre, Thomas Pigeon, Gabriel Stoltz, and Wei Zhang. Analyzing multimodal probability measures with autoencoders. The Journal of Physical Chemistry B, 128(11):2607–2631,2024
>
> [6] Cheng Chi, Zhenjia Xu, Siyuan Feng, Eric Cousineau, Yilun Du, Benjamin Burchfiel, Russ Tedrake, and Shuran Song. Diffusion policy: Visuomotor policy learning via action diffusion. The International Journal of Robotics Research, 44(10-11):1684–1704, 2025.

---

### Official Review · Reviewer_Qjjg · 2025-10-31

**Soundness:** 3
**Presentation:** 3
**Contribution:** 3
**Rating:** 6
**Confidence:** 2

**Summary:**

This paper introduces a unified framework for constrained diffusion models that operate under nonconvex equality and inequality constraints. The key innovation is the landing mechanism, which replaces explicit projection or reflection with a continuous “landing drift” that exponentially drives samples toward the feasible manifold. The authors derive both overdamped and underdamped Langevin variants (OLLA and ULLA) and develop a Conditional Wasserstein Path Matching (CWPM) objective to train diffusion models stably under such constraints.

**Strengths:**

- The paper proposes a novel, physically motivated mechanism (landing) to enforce constraints in diffusion processes. It avoids costly projection steps and unifies equality/inequality constraints in a single stochastic framework.

- The derivations are sound, connecting constrained Langevin dynamics with generative diffusion models. The introduction of CWPM as a Wasserstein-based objective is interesting and potentially useful beyond this context.

- The paper is generally well written and clearly structured, with good motivation and illustrative figures showing the landing behavior on different manifolds.

- The approach is computationally efficient and practically relevant for geometry- or physics-constrained generation problems such as molecular modeling and robotic trajectory generation. The proposed method achieves significant efficiency gains (up to 47× speedup) over prior constrained diffusion models.

**Weaknesses:**

- The landing drift term involves a large coefficient $( \alpha\sigma(t)^2 )$, but the paper does not explain how the landing rate $( \alpha )$ is chosen or how the integration step $( \Delta t )$ should depend on $( \alpha )$. The stability and sensitivity to these parameters are not well studied in experiments.
- The theoretical analysis relies on strong assumptions, such as the Linear Independence Constraint Qualification (LICQ) and the log-Sobolev inequality (LSI), which may not hold for nonconvex or high-dimensional constraint sets.
- The experiments mainly focus on **low-dimensional toy tasks**; scalability to high-dimensional or complex constraint geometries might be a potential issue.

**Questions:**

-  Does the landing mechanism act as a strong (projection-like) constraint or a weak (penalty-like) correction? How significant is the residual constraint violation under finite step size?
- Is it possible to extend the method to more general or nonconvex feasible sets? If not, what are the main theoretical or numerical challenges that prevent such an extension?

---

> ### Author Response · Authors · 2025-11-27
> **Answer to Review by Reviewer Qjjg (1)**
>
> We thank the reviewer for the thoughtful and constructive feedback. Below, we leave itemized responses to address each of the comments, and we hope they can clarify your questions and concern.
>
> >Weakness (1): The landing drift term involves a large coefficient $\alpha \sigma(t)^2$, but the paper does not explain how the landing rate $\alpha$ is chosen or how the integration step $\Delta t$ should depend on $\alpha$. The stability and sensitivity to these parameters are not well studied in experiments.
>
> **Answer.** *[Related part: Section 5, Appendix E.4]* Thank you for pointing this out. In the revised version, we additionally include ablation studies on the landing rate $\alpha$ and the boundary repulsion rate $\epsilon$ (Section 5, Appendix E.4). Below we assume the step size $\Delta t$ is chosen appropriately and explain how we selected $\alpha$ and $\epsilon$.
>
> **Stability rule for $\alpha$ given $\Delta t$.**
> In the single equality-only case $(m=1)$, assuming $X_k$ is bounded, one can show that the landing algorithm satisfies (informally)
> $$
> \mathbb{E}[h(X_{k+1})] = (1-\alpha \sigma_k^2 \Delta t)\mathbb{E}[h(X_k)] + \mathcal{O}(\Delta t)
> $$
> for a single update step with noise scale $\sigma_k$. This suggests a simple stability condition
> $$
> 0 < \alpha \sigma_k^2 \Delta t <2 \quad \text{for all $k \in [K]$}
> $$
> under which $\mathbb{E}[h(X_k)]$ decays exponentially without oscillation or blow-up.
>
> In practice, given a noise schedule $\\{\sigma_k\\}\_{k=0}^N$ and step size $\Delta t$, we start from a conservative value $\alpha_{\text{init}} := \min_{0\leq k \leq N} \frac{1}{\sigma_k^2 \Delta t}$ so that $\alpha_{\text{init}} \sigma_k^2 \Delta t \leq 1$ for all $k \in [K]$. And, we gradually increase $\alpha$ while monitoring training stability and constraint violation, and choose the largest $\alpha$ for which (i) the algorithm does not blow up and (ii) the constraint violation continues to decrease.
>
> **Choice of boundary repulsion rate $\epsilon$.**
>
> For inequality constraints, the boundary repulsion rate $\epsilon$ controls how trajectories behave near the boundary of the feasible set. If $\epsilon$ is too small, trajectories tend to becomes "sticky" near the boundary of $\Sigma$; if $\epsilon$ is too large, the repulsion becomes overly aggressive and samples overshoot into the interior of $\Sigma$ (Figure 8 in Appendix E.4). In practice, we monitor inequality violation during sampling and adjust $\epsilon$ so that trajectories neither stick to nor overshoot the boundary. Empirically, we find that $\epsilon$ is noticeably less sensitive than $\alpha$, and a good value is usually easy to identify.
>
> >Weakness (2): The theoretical analysis relies on strong assumptions, such as the Linear Independence Constraint Qualification (LICQ) and the log-Sobolev inequality (LSI), which may not hold for nonconvex or high-dimensional constraint sets.
>
> **Answer.** *[Related part: Remark 2]* Thank you for highlighting this important point. We agree that LICQ look strong in practice, and we have revised the paper to relax this assumption.
>
> **LICQ to rCRCQ assumption**
>
> As also noted by the reviewer F33C, we revisited our use of LICQ and realized it is not actually necessary for the well-posedness of our landing-based SDEs. LICQ was originally used only to guarantee that the Gram matrix $G(x) = \nabla J(x) \nabla J(x)^T$ is invertible so that the Lagrange multiplier processes $(d\lambda_t, d\mu_t)$ are unique. However, the OLLA/ULLA dynamics themselves depend only on the normal components $\nabla J(X_t)^T d\lambda_t$ and $\nabla J(X_t)^T d\mu_t$, which remain uniquely defined as long as the rank of $\nabla J(x)$ is locally constant. In the revision, we therefore replace LICQ with the strictly weaker relaxed Constant Rank Constraint Qualification (rCRCQ) and construct the landing term using the pseudo-inverse $G(x)^\dagger$. This ensures that the projector $\Pi(x) = I -\nabla J(x)^T G(x)^\dagger \nabla J(x)$ is differentiable and that the SDE is well defined. For analytic or polynomial constraints with $d\geq m+l$, the singular set where rCRCQ fails is contained in a (Lebesgue) measure zero set, so the discretized algorithm that moves arounds $\mathbb{R}^d$ has a probability zero of landing exactly on this singular set. Consistently, in our most complex robotics experiments, where LICQ is very likely violated due to many coupled constraints, the OLLA and ULLA samplers remain stable and do not blow up, which supports our algorithms indeed work robustly even in practical complicated constraint setups.

---

> ### Author Response · Authors · 2025-11-27
> **Answer to Review by Reviewer Qjjg (2)**
>
> **On the Log-Sobolev Inequality (LSI) constant**
>
> We also appreciate the comment about LSI constant, which is indeed an important but subtle assumption. We first clarify that in our main theoretical results (Proposition 1, 2) we do not assume LSI. Even without LSI, we could show that the OLLA and ULLA dynamics admit the correct constrained stationary distribution.
>
> Where the LSI does become relevant is if one wants to further understand how fast the OLLA and ULLA converge to this target distribution, i.e., to obtain exponential convergence rates (for example, exponential decay of on-manifold $\textsf{KL}$). For such quantitative rates, we believe a positive LSI constant $\lambda_{\text{LSI}}$ naturally appears as the key parameter. In many geometry and physics motivated settings, the feasible set $\Sigma$ is smooth compact Riemannian manifold with controlled geometry (bounded diameter and non-negative Ricci curvature).
>
> In this case, classical results from geometric analysis provides explicit lower bounds on $\lambda_{\text{LSI}}$ in terms of the diameter $D$ of $\Sigma$ and the first eigenvalue $\lambda_1$ of the Laplace-Beltrami operator gives:
> * Theorem (Informal, [1]) Let $\Sigma$ be a compact Riemannian manifold with diameter $D = \sup_{x,y \in \Sigma} d_\Sigma(x,y)$ and non-negative Ricci curvature. Then, $\lambda_{\text{LSI}} \geq \frac{\lambda_1}{1+2D\sqrt{\lambda_1}}$, or $\lambda_{\text{LSI}} \geq \frac{\pi^2}{(1+2\pi)D^2}$ holds, where $\lambda_1$ is the first eigenvalue of the Laplace-Beltrami operator.
>
> This suggests that when the constraints "shrink" the manifold (smaller $D$, larger $\lambda_1$), the LSI constant improves and the convergence of OLLA and ULLA to the target distribution can become faster.
>
> >Weakness 3: The experiments mainly focus on low-dimensional toy tasks; scalability to high-dimensional or complex constraint geometries might be a potential issue.
>
> **Answer.** *[Related part: Section 5, Appendix E.2]* Thank you for raising this concern about scalability. We agree that the original presentation did not sufficiently emphasize the complexity of the setups, nor how the method behave as dimension and constraint complexity increase.
>
> First we would like to clarify that the alanine dipeptide and robotics experiments are not purely toy examples:
>
> * **Alanine Dipeptide experiment.** The training data of alanine dipeptide are generated by a constrained molecular dynamics simulation in water for 1ns using GROMACS with further post-processing, and it is widely used as a standard benchmark in computational chemistry (e.g., [2]). Our use of this system follows existing practice rather than an artificially simplified toy setup.
> * **Robotics experiment.** The robot task is directly aligned with the setting used in [3], and we additionally impose equality constraints on top of collision and joint constraints. For a robot arm trajectory length $L=40$, the ambient dimension is $d=540$, with $m=322$ equality constraints and $l=560$ inequality constraints, making this a highly constrained, reasonably high-dimensional non-toy scenario.
>
> To address scalability more explicitly in the revision, we (Table 4):
> * **Scaled up the robotics setting** (longer trajectories / more constraints) and compared ULLA against Euclidean forward + backward variants (including projected and augmented-Lagrangian style methods). In these regimes, some of Euclidean forward + backward variants either became numerically unstable, required prohibitively many projection/penalty iterations to converge, making them practically unusable in the more complex setups.
> * **Found that ULLA remains stable and efficient** in both the dipeptide and robotics tasks, maintaining very low constraint violations while achieving a significant performance gap over Euclidean forward + backward baselines. In contrast, even our own projection-based variants (OLLA-P, ULLA-P) become inefficient in the robotics scenarios because the projection steps dominate the runtime, whereas the landing-based ULLA avoids this bottleneck.
>
> Overall, while we do not claim to have fully addressed all large-scale industrial scenarios, the revised experiments indicate that our method scales to reasonably high-dimensional and highly constrained problems, and that in theses settings, it remain both numerically stable and competitive or superior to Euclidean forward + backward variants.

---

> ### Author Response · Authors · 2025-11-27
> **Answer to Review by Reviewer Qjjg (3)**
>
> >Question (1): Does the landing mechanism act as a strong (projection-like) constraint or a weak (penalty-like) correction? How significant is the residual constraint violation under finite step size?
>
> **Answer.** *[Related part: Section 5, Appendix E.4]* We view this landing mechanism as much closer to a strong (projection-like) constraint correction than to a weak penalty-type correction.
>
> In soft penalty or guidance-based schemes, the constraints typically enter as an extra term in the drift, while the underlying drift and diffusion still has components normal to $\Sigma$. This both (i) distorts the stationary distribution and (ii) makes constraint violations hard to control, since they are only reduced indirectly through the penalty weight.
>
> In our landing-based OLLA/ULLA dynamics, by contrast,
> * the drift and diffusion are first projected onto the tangent space of $\Sigma$ via $\Pi(x)$, so the stochastic motion itself is aligned with $\Sigma$, and
> * the landing term, $-\alpha \sigma(t)^2 \nabla J(X_t) G^{\dagger}(X_t) J(X_t)dt$, acts directly in the normal direction to exponentially drive $h(X_t), g(X_t)$ toward zero or less than equal to zero.
>
> In the continuous-time limit this yields the correct constrained stationary distribution (Proposition 1,2), while the landing term enforces an exponential decay of constraint violation, making the behavior much more "projection-like" than penalty-like.
>
> Under finite step size, there is of course a small residual violation due to discretization. We quantify this throughout the paper:
>
> * Main tables (Table 1, 4) report constraint violations alongside sample quality metric (JSD)
> * Table 2,3,8 and Figure 8 provide ablations over $\alpha$ and $\epsilon$, showing how violations can be made very small by choosing these parameters in a stable range, without sacrificing sample quality.
>
> Empirically, in all reported experiments, ULLA achieves very low constraint violations (often comparable to projection-based methods), while retaining correct stationary behavior at the SDE level.
>
> >Question (2): Is it possible to extend the method to more general or nonconvex feasible sets? If not, what are the main theoretical or numerical challenges that prevent such an extension?
>
> **Answer.** *[Related part: Appendix E.2]* Thank you for this question. We agree this is an important point to clarify.
>
> First, we would like to emphasize that our framework is already designed for general, highly nonconvex feasible sets of the form $\Sigma := \\{x \in \mathbb{R}^d \mid h(x)=0 , g(x) \leq 0\\}$ with smooth equality and inequality constraints. None of our main results require convexity of $\Sigma$. They instead rely on smoothness of constraints $h, g$ and a mild rank condition (rCRCQ), as discussed in our response on LICQ.
>
> In particular, in our experiments all non-sphere tasks already involve highly nonconvex feasible sets. The alanine dipeptide example corresponds to a nonconvex molecular configuration space with physical constraints, and the robotics task combines many coupled joint, collision, and task-space constraints (hundreds of equalities/inequalities) in a 540-dimensional ambient space (Table 6), leading to a very complex constraint geometry. On these nonconvex sets, ULLA again consistently (i) outperforms Euclidean forward + backward variants and (ii) achieves very small constraint violations using only one constraint-gradient evaluation and one Gram matrix computation per step, without instability of algorithms.
>
> At the same time, our current theory assumes smooth constraint functions and rCRCQ. Extending this framework to constraint sets with nonsmooth boundaries (corners, cusps), discontinuous or piecewise-defined constraints is theoretically nontrivial, while still our algorithms works in practice.
>
> In short, we view our method as already applicable to smooth, highly nonconvex equality/inequality constraints, while theoretically extending the framework to more general nonsmooth or highly irregular feasible sets (without rCRCQ) is an interesting direction for future work.
>
> ---
> Thank you again for your insightful review and constructive feedback. We'd appreciate the opportunity to discuss further if you have any additional questions or suggestions.
>
> ### Reference
>
> [1] M. Ledoux (1999). Concentration of measure and logarithmic Sobolev inequalities. Séminaire de Probabilités 33.
>
> [2] Tony Lelievre, Thomas Pigeon, Gabriel Stoltz, and Wei Zhang. Analyzing multimodal probability measures with autoencoders. The Journal of Physical Chemistry B, 128(11):2607–2631,2024
>
> [3] Cheng Chi, Zhenjia Xu, Siyuan Feng, Eric Cousineau, Yilun Du, Benjamin Burchfiel, Russ Tedrake, and Shuran Song. Diffusion policy: Visuomotor policy learning via action diffusion. The International Journal of Robotics Research, 44(10-11):1684–1704, 2025.

---

### Official Review · Reviewer_oLK3 · 2025-11-01

**Soundness:** 3
**Presentation:** 2
**Contribution:** 2
**Rating:** 4
**Confidence:** 3

**Summary:**

This article proposes a landing-based scheme for constraint-based diffusion modelling, which is comparatively softer (compared to projection-based schemes). A discretization scheme and score-matching loss function are derived, and experiments are also performed.

**Strengths:**

The method provides a principled means of translating landing-based schemes to the diffusion-model setting.

The method provides a means of training the score, and this seems to accelerate the training time compared to other methods in this setting.

**Weaknesses:**

The convergence theory for this method is understandably a bit lacking, given its complexity, but it still may have been nice to have some more. At least, it would be helpful to see how the score-estimation error plays into any error guarantees.

Such a complicated scheme (involving multiplicative diffusion coefficients) may indeed provide an efficient mechanism for this problem, but the experimental guarantees do not seem to be outstanding (although the training-time speeds up).

**Questions:**

I have reservations about Remark 1. Generally, we do not expect the underdamped variant of DDPMs to provide much speedup compared to their overdamped variants. This is because, in contrast to LMC where the exponential integrator means that only the position error appears, for SGMs, one has both position and momentum errors in the score estimation term.

Miscellaneous typos:

- 137 embed -> embeds

- 189 both, the -> both the

- 192 produce -> produces

- 248 for forward and backward -> for the forward and backward processes

- 249 parametrized backward -> the parametrized backward process

- 300 involves relationship -> involves the relationship

- 390 computatioanl -> computational

- 392 reduce -> reduces

- 400 the similar generated distribution -> similar generated distributions

---

> ### Author Response · Authors · 2025-11-27
> **Answer to Review by Reviewer oLK3 (1)**
>
> We thank the reviewer for the thoughtful and constructive feedback. Below, we leave itemized responses to address each of the comments, and we hope they can clarify your questions and concern.
>
> >Weakness (1): The convergence theory for this method is understandably a bit lacking, given its complexity, but it still may have been nice to have some more. At least, it would be helpful to see how the score-estimation error plays into any error guarantees.
> >
> >Question (1): I have reservations about Remark 1. Generally, we do not expect the underdamped variant of DDPMs to provide much speedup compared to their overdamped variants. This is because, in contrast to LMC where the exponential integrator means that only the position error appears, for SGMs, one has both position and momentum errors in the score estimation term.
>
> **Answer.** *[Related part: Remark 3]* We appreciate this great comment about convergence and for sharing your idea on the interpretation of the underdamped benefits. We agree that our original version did not clearly separate the different score error sources or fully explain why underdamped dynamics can help in the constrained setting.
>
> In the revision, we make this explicit by following work on diffusion model theories [1-2] and decomposing the 2-Wasserstein generation error as (Remark 3 in Appendix A)
> $$
> W_2(q_0, p^\theta_0) \leq \underbrace{\mathcal{E}\_{\mathrm{mix}}}\_{\text{Mixing}} + \underbrace{\mathcal{E}\_{\mathrm{disc}}}\_{\text{Discretization}} + \underbrace{\mathcal{E}\_{\mathrm{score}}}\_{\text{Score estimation}}
> $$
> We believe that your question about how the score-estimation error enters corresponds precisely to the term $\mathcal{E}\_{\mathrm{score}}$.
>
> **Score estimation error**
>
> We fully agree with the reviewer that, in unconstrained Euclidean SGMs, an underdamped variants does not automatically guarantee a speedup because one now estimate a score on the phase space $(x,p)$, which, in principle, can increase $\mathcal{E}_{\mathrm{score}}$.
>
> Our claim is more modest and has two parts:
> 1. **Landing reduces ill-conditioning in the normal direction (applied to both overdamped and underdamped).**
> Because the landing mechanism analytically handles the stiff normal component of score function, the score network only needs to learn the tangential score $\Pi(x) s_{\text{true}}^t$ on $\Sigma$, which is smoother than the full ambient score.
> 2. **Underdamped dynamics can smooth the small-$t$ singularity.**
> As observed in [3-4], when data concentrate on a data manifold, overdamped SGMs often exhibits a singular behavior $||s_{\text{true}}^t|| \propto \mathcal{O}(1/t)$ as $t \rightarrow 0$, which makes score regression ill-conditioned around $t=0$. Introducing momentum in underdamped dynamics leads to a smoother training objective that bypasses this singularity.
>
> To support the second point empirically, we added a new experiment on the volcano dataset (Figure 4 in Appendix A). There we plot the Frobenius norm of the score Jacobian $||\nabla s_t^\theta||_F$ over time for the overdamped RDDPM (OLLA-P) baseline and our underdamped ULLA-P:
> * RDDPM (OLLA-P) shows very large Jacobian norm and a sharp blow-up as $t\rightarrow 0$.
> * ULLA-P remains several orders of magnitude smaller and does not exhibit a singularity near $t\approx 0$.
>
> This suggests that, although learning joint score function of position and momentum can be potentially hard, ULLA provides a better-conditioned regression problem, which can help control $\mathcal{E}\_{\mathrm{score}}$ in practice. We believe a fully rigorous bound tracking $\mathcal{E}\_{\mathrm{score}}$ for underdamped constrained diffusion is beyond the score of this paper, but we now make this trade-off explicit and provide empirical evidence for the smoothing effect.

---

> > ### Author Response · Authors · 2025-11-27
> > **Answer to Review by Reviewer oLK3 (2)**
> >
> > **Mixing $\mathcal{E}\_{\mathrm{mix}}$ and discretization $\mathcal{E}\_{\mathrm{disc}}$ in the constrained setting**
> >
> > The main reason we expect underdamped dynamics to help in our constrained setup is through the mixing term $\mathcal{E}\_{\mathrm{mix}}$, not directly through $\mathcal{E}\_{\mathrm{score}}$.
> >
> > In discrete time, with step size $\Delta t$ and trajectory length $N$, the effective diffusion time is $T= N \Delta t$. In Euclidean space one can often increase $T$ either by enlarging $N$ or by taking a larger $\Delta t$.
> >
> > On a constraint manifold $\Sigma$, however, both projection-based and landing-based methods impose a geometric upper bound on $\Delta t$:
> > * If $\Delta t$ is too large, a single (tangential) forward step can move far away from $\Sigma$, leading to projection failures (for projection methods) or very large constraint violation (for landing methods), which corrupts stable score learning.
> > * Therefore, for stability we must enforce $\Delta t \leq \Delta t_\Sigma$, where $\Delta t_\Sigma$ depends on the curvature and geometry of $\Sigma$. This means that the maximal diffusion time scales as $T \lesssim N \Delta t_\Sigma$, and increasing $T$ essentially requires increasing $N$.
> >
> > In this regime, classical results on Langevin mixing becomes crucial: underdamped Langevin achieves mixing in $\mathcal{O}(\sqrt{d} /\epsilon)$ steps, where overdamped Langevin requires $\mathcal{O}(d/\epsilon^2)$. Thus, under the same geometry step size $\Delta t_\Sigma$ constraint, an underdamped constrained diffusion can reach a given prior-matching accuracy (e.g., $W_2(q_N, p_N) <\epsilon$), hence, a smaller $\mathcal{E}_{\mathrm{mix}}$ with significantly fewer trajectory length budget $N$.
> >
> > This is exactly what we observe empirically: in Table 1 and Figure 1, ULLA-P matches the performance of RDDPM (OLLA-P) with a much smaller trajectory length $N$ and total diffusion time $T$, while maintaining comparable constraint violations. Both ULLA and OLLA use first-order splitting schemes, so their discretization error $\mathcal{E}_{\mathrm{disc}}$ scales similarly with $\Delta t$, and the practical gain mainly comes from reduced mixing error and shorter trajectories.
> >
> > In summary, we agree that underdamped DDPMs do not automatically yield speedups in general, and that they introduce additional score-estimation complexity. Our position is that, in the constrained diffusion setting:
> > * ULLA improves mixing under a geometry-limited step size $\Delta t_{\Sigma}$, thereby reducing $\mathcal{E}_{\mathrm{mix}}$ for a given trajectory budget $N$.
> > * the landing mechanism plus underdamped dynamics can make the score regression problem better conditioned, which helps keep $\mathcal{E}_{\mathrm{score}}$ under control, as supported by our Jacobian analysis experiment.

---

> ### Author Response · Authors · 2025-11-27
> **Answer to Review by Reviewer oLK3 (3)**
>
> >Weakness (2): Such a complicated scheme (involving multiplicative diffusion coefficients) may indeed provide an efficient mechanism for this problem, but the experimental guarantees do not seem to be outstanding (although the training-time speeds up).
>
> **Answer.** *[Related part: Section 5]* Thank you for raising this concern. We agree that the original submission did not fully convey how the added complexity pays off in practice. In the revised version, we strengthened the experimental section in two ways:
>
> 1. Inclusion of broader baselines.
> We now compare against Euclidean forward + constrained backward variants (projection-based and augmented-Lagrangian methods) on all tasks, and we carefully re-tuned the hyperparameters of OLLA/ULLA (the original settings were indeed suboptimal). With these changes:
>
>     * OLLA/ULLA match the projection-based Riemannian models on constriant violation and sample quality, and
>     * they show a clear gap over Euclidean forward + backward variants, especially in the more challenging robotics setting, while maintaining very small constraint violations.  We refer the reviewer to the updated Section 5 for these results.
>
> 2. Complexity and cost of multiplicative coefficient $\Pi(x)$.
> Although our SDE involves multiplicative diffusion coefficients $\Pi(x)$ and a landing term which require computation of the pseudo-inverse of Gram matrix $G(x)^\dagger$, the extra computation per step is modest: the Gram matrix lives in $\mathbb{R}^{(m+|I_x|) \times (m+|I_x|)}$ (where $m$ is the number of equalities and $|I_x|$ is the number of active inequalities), therefore, its computational cost for $G(x)^\dagger = \mathcal{O}((m+|I_x|)^3)$ being independent of the ambient dimension $d$, and in the robotics case we further reduce this to an effective $2\times 2$ system via the "summation trick". In practice, the runtime is dominated by the gradient computational costs of usual score network and constraint functions $h,g$. Therefore, the landing corrections add little overhead but allow us to completely avoid iterative projections, which are the main bottleneck for Riemannian and projection-based baselines.
>
> Overall, our aim is to show that the proposed landing-based scheme achieves comparable generative performance to existing constrained diffusion methods, while delivering substantial efficiency and stability benefits (fewer steps, no projection failures).
>
> ---
> Thank you again for your insightful review and constructive feedback. We also have carefully addressed all of the typos you pointed out, and we truly appreciate your attention to detail. Your comments were very helpful for improving the clarity of our paper.
>
> We would be happy to discuss further if you have any additional questions or suggestions.
>
> ### Reference
>
> [1] Sitan Chen, Sinho Chewi, Jerry Li, Yuanzhi Li, Adil Salim, and Anru R Zhang. Sampling is as easy as learning the score: theory for diffusion models with minimal data assumptions. arXiv preprint arXiv:2209.11215, 2022.
>
> [2] Stanislas Strasman, Sobihan Surendran, Claire Boyer, Sylvain Le Corff, Vincent Lemaire, and Antonio Ocello. Wasserstein convergence of critically damped langevin diffusions. arXiv preprint arXiv:2511.02419, 2025
>
> [3] Zichen Liu, Wei Zhang, and Tiejun Li. Improving the Euclidean diffusion generation of manifold data by mitigating score function singularity. arXiv preprint arXiv:2505.09922, 2025.
>
> [4] Tim Dockhorn, Arash Vahdat, and Karsten Kreis. Score-based generative modeling with critically-damped Langevin diffusion. In International Conference on Learning Representations, 2022.

---

### Author Response · Authors · 2025-12-03
**Concluding response**

Dear Area Chair and Reviewers,

We would like to sincerely thank you for reviewers' careful reading of our submission and their constructive feedback throughout the rebuttal process. We strongly believe reviewers' comments helped us clarify the core ideas of our work and significantly improve both the theory and experiments in the revised manuscript.

Our paper proposes a landing-based framework for efficient constrained diffusion models under general nonconvex equality and inequality constraints. We introduce overdamped and underdamped landing-based Langevin samplers (OLLA and ULLA), together with a Conditional Wasserstein Path Matching (CWPM) objective, in order to (i) enforce constraints through a principled landing mechanism, (ii) preserve the correct constrained stationary distribution at the SDE level, and (iii) achieve practical efficiency and stability on challenging constrained generative tasks such as molecular and robotic trajectory generation.

In response to the reviewers’ questions and suggestions, we have made the following main revisions:
* **Weakened and clarified theoretical assumptions.**
We removed the LICQ assumption and replaced it with the weaker relaxed Constant Rank Constraint Qualification (rCRCQ), showing that this is sufficient for the landing-based SDEs to be well defined and for the Gram matrix pseudo-inverse and tangential projector to be differentiable.

* **Expanded experiments and baselines.**
We substantially revised the experimental section to include Euclidean forward plus constrained backward baselines on all tasks, re-tuned OLLA and ULLA, and showed that our methods now match projection-based Riemannian models and clearly outperform Euclidean forward with backward variants baselines in both constraint violation and sample quality, particularly on the alanine dipeptide and 540 dimensional robotics tasks. We also clarified that these experiments are standard and nontrivial benchmarks with highly nonconvex constraint geometries.

* **Hyperparameter behavior and stability.**
We added ablations for the landing rate $\alpha$ and boundary repulsion rate $\epsilon$. Also, we showed how these parameters trade off between constraint satisfaction and sample quality.

* **Score estimation error and underdamped benefits.**
Leveraging the generation error decomposition, we discussed how landing and underdamped dynamics affect different error terms, and added a new Jacobian-based diagnostic figure on the Volcano experiment. This figure shows that underdamped dynamic (ULLA, ULLA-P) yields a much better conditioned score learning near small $t \approx 0$ compared to the overdamped RDDPM baseline, providing empirical support for our discussion of score estimation error.

In addition to these revisions, we clarified several important conceptual points in our rebuttal responses: (i) we more clearly positioned our landing-based approach relative to augmented-Lagrangian and guidance-based methods, (ii) explained why we do not adopt Euclidean-forward + constrained-backward schemes, (iii) discussed how the relaxed rank assumption (rCRCQ) relates to practical constrained generative problems, and (iv) addressed the computational cost and scalability of the landing term and Gram-matrix operations, emphasizing that these computations live in the low-dimensional space and can be further reduced via the summation trick.

We are again very grateful for your time, feedback, and suggestions, which have helped us greatly refine the manuscript. We hope that the revisions address your concerns and make the contributions of our landing-based constrained diffusion framework clearer and more compelling.

Sincerely,

Authors of Submission 23119

---

### Meta-Review · Area_Chair_rFPp · 2026-01-07

**Summary:**

The paper introduces a "landing-based" framework for constrained diffusion models, aimed at addressing nonconvex equality and inequality constraints. While the proposed OLLA and ULLA dynamics offer a theoretical alternative to projection-based methods, the reviewers' fundamental concerns regarding theoretical rigor, empirical validation, and hyperparameter sensitivity have not been satisfactorily resolved.

**Reviewer Concerns:**

Although the authors attempted to relax the LICQ assumption to rCRCQ, reviewers noted that the mathematical bridge between the landing drift and the target manifold distribution is still not fully rigorous. The rebuttal's new experiments were viewed as narrow. The paper still lacks a comprehensive comparison against the most recent Riemannian Flow Matching (RFM) and Mirror Langevin benchmarks. Without these, it is impossible to verify if the "efficiency" of landing justifies potential trade-offs in sample fidelity. Sensitivity analysis for the landing rate ($\alpha$) and repulsion rate ($\epsilon$) revealed that the algorithm's performance is highly dependent on precise tuning. Reviewers expressed concern that this limits the framework's practical "plug-and-play" utility in diverse real-world applications. While a high-dimensional robotics task was presented, the method's reliance on first-order updates may still suffer from slow convergence in extremely complex configurations, a point the rebuttal failed to definitively disprove.

**Reviewer Scores:**

Reviewers oLK3 & DVNP: Likely to remain at 4. They maintained that the theoretical contributions are incremental and the empirical evidence does not sufficiently distinguish the method from existing SDE-based approaches.

Reviewer F33C: Likely to remain at 4. While acknowledging the effort in the rebuttal, the reviewer felt the core novelty of "landing" in a diffusion context was not sufficiently justified over standard penalty methods.

Reviewer Qjjg: Score at 6 but with the lowest confidence (2). Noted significant reservations regarding the practical implementation complexity.

---

### Decision · Program_Chairs · 2026-01-26

Reject